# ALIX- and ESCRT-III–dependent sorting of tetraspanins to exosomes

Jorge Larios, Vincent Mercier, Aurélien Roux*, and Jean Gruenberg*

**The intraluminal vesicles (ILVs) of endosomes mediate the delivery of activated signaling receptors and other proteins to lysosomes for degradation, but they also modulate intercellular communication when secreted as exosomes. The formation of ILVs requires four complexes, ESCRT-0, -I, -II, and -III, with ESCRT-0, -I, and -II presumably involved in cargo sorting and ESCRT-III in membrane deformation and fission. Here, we report that an active form of the ESCRT-associated protein ALIX efficiently recruits ESCRT-III proteins to endosomes. This recruitment occurs independently of other ESCRTs but requires lysobisphosphatidic acid (LBPA) in vivo, and can be reconstituted on supported bilayers in vitro. Our data indicate that this ALIX- and ESCRT-III–dependent pathway promotes the sorting and delivery of tetraspanins to exosomes. We conclude that ALIX provides an additional pathway of ILV formation, secondary to the canonical pathway, and that this pathway controls the targeting of exosomal proteins.**

## Introduction

Upon endocytosis, signaling receptors and other cell surface proteins are delivered to early endosomes, from where they are recycled to the plasma membrane, transported to the trans-Golgi network, or targeted to lysosomes for degradation (Scott et al., 2014). In the latter case, endocytosed cargoes are selectively incorporated into intraluminal vesicles (ILVs), which form within multivesicular regions of endosomes. ILVs are then transported toward late endosomes and lysosomes, where they are degraded together with their cargo (Scott et al., 2014). ILVs, however, may meet a different fate and escape degradation. They may undergo backfusion with the limiting membrane, in particular when hijacked by pathogenic agents (Bissig and Gruenberg, 2014; Gruenberg and van der Goot, 2006; Nour and Modis, 2014; van Niel et al., 2018). They may also harbor major histocompatibility complex class II molecules loaded with peptides for presentation at the plasma membrane (Kleijmeer et al., 2001; Peters et al., 1991; Zwart et al., 2005) or contribute to the biogenesis of melanosomes in melanocytes (Berson et al., 2001; Hurbain et al., 2008) and other lysosome-related organelles in specialized cell types (Delevoye et al., 2019; Marks et al., 2013). Finally, ILVs can also be secreted into the extracellular milieu as exosomes (Kowal et al., 2014), which serve as key modulators of intercellular communication in many physiological and pathological processes (Kalra et al., 2012; McGough and Vincent, 2016; Simons and Raposo, 2009). Essentially nothing is known about the mechanisms that control the alternative fates of ILVs to degradation or secretion, or the corresponding

targeting of ILV cargoes to the lysosomes or the extracellular milieu as physiological mediators.

In contrast, much progress has been made in unraveling how proteins are incorporated into ILVs destined for the lysosomes. Sorting is mediated by the addition of an ubiquitin signal (Hicke and Riezman, 1996; Katzmann et al., 2001; Kölling and Hollenberg, 1994), which is recognized by the endosomal sorting complexes required for transport (ESCRTs; Babst et al., 2002a,b; Katzmann et al., 2003; Saksena et al., 2009; Teis et al., 2008). ESCRTs are organized in four complexes, ESCRT-0, -I, -II, and -III (Williams and Urbé, 2007), with ESCRT-0, -I, and -II having multiple ubiquitin-binding domains (Shields and Piper, 2011). ESCRT-III is nucleated at the membrane by ESCRT-II (Babst et al., 2002b; Teis et al., 2010) and exhibits membrane remodeling activity proposed to be involved in ILV formation. The main component of ESCRT-III, charged multivesicular body protein 4 (CHMP4, SNF7 in yeast), forms spiral-shaped structures that act as molecular springs (Chiaruttini et al., 2015; Wollert et al., 2009). These can store mechanical energy that is proposed to play a role in all membrane remodeling functions of ESCRT-III (Chiaruttini and Roux, 2017; Elia et al., 2011; Guizetti et al., 2011; Shen et al., 2014). Consistently, ESCRT-III is proposed to act as a general fission machinery away from the cytoplasm, as it is required for cytokinesis (Carlton and Martin-Serrano, 2007; Mierzwa et al., 2017; Morita et al., 2007), virus budding (Garrus et al., 2001; Martin-Serrano et al., 2001; Strack et al., 2003), nuclear envelope reassembly following mitosis (Gu et al.,

---

Department of Biochemistry, Université de Genève, Geneva, Switzerland.

*A. Roux and J. Gruenberg contributed equally to this study; Correspondence to Jean Gruenberg: jean.gruenberg@unige.ch; Aurélien Roux: aurelien.roux@unige.ch.

2017; Olmos et al., 2015, 2016; Vietri et al., 2015), exosome biogenesis (Colombo et al., 2013), and autophagy (Filimonenko et al., 2007; Lee et al., 2007). ESCRT-III functions may also depend on the turnover of individual subunits via the triple A ATPase vacuolar protein sorting-associated protein (VPS4; Adell et al., 2014, 2017; Mierzwa et al., 2017). Moreover, in a process perhaps similar to its fission activity, ESCRT-III also mediates plasma membrane (Jimenez et al., 2014; Scheffer et al., 2014), endosome (Radulovic et al., 2018; Skowyra et al., 2018; López-Jiménez et al., 2018), and nuclear envelope (Denais et al., 2016; Raab et al., 2016) repair. ESCRT-independent mechanisms have also been proposed to regulate the biogenesis of intralumenal membranes in specialized cell types, including CD63 in melanocytes (Theos et al., 2006; van Niel et al., 2011, 2015) and perhaps other cell types (Edgar et al., 2014), and ceramides in oligodendrocytes (Trajkovic et al., 2008).

In addition to the classic ESCRT-0, -I, -II pathway, other mechanisms for ESCRT-III recruitment have been suggested (Christ et al., 2016; Gu et al., 2017; Jimenez et al., 2014; Scheffer et al., 2014; Tang et al., 2016; Vietri et al., 2015). In yeast, BRO-domain containing protein 1 (BRO1; Kim et al., 2005), together with ESCRT-0, nucleates SNF7 polymers on membranes independently of ESCRT-I and -II (Tang et al., 2016). In mammals, there are three proteins containing a BRO1 domain, which all interact with SNF7/CHMP4: His domain-containing protein tyrosine phosphatase (HD-PTP), ALG-2 interacting protein X (ALIX), and BRO1 domain-containing protein (BROX; Ichioka et al., 2007, 2008; Katoh et al., 2003; Strack et al., 2003). By linking ESCRT-0 and -III, HD-PTP may participate in the ILV sorting of EGF receptor (EGFR) independently of ESCRT-II (Ali et al., 2013; Doyotte et al., 2008). Similarly, ALIX is believed to mediate the ubiquitin-independent sorting of the G protein–coupled receptors PAR1 and P2Y1 into ILVs for degradation (Dores et al., 2012, 2016), and yet ALIX binds ubiquitin (Dowlatshahi et al., 2012; Joshi et al., 2008; Keren-Kaplan et al., 2013). ALIX is also required for other ESCRT-dependent processes such as human immunodeficiency virus (HIV) budding (Strack et al., 2003), cytokinetic abscission (Carlton and Martin-Serrano, 2007; Christ et al., 2016; Morita et al., 2007), autophagy (Murrow and Debnath, 2015), exosome biogenesis (Abrami et al., 2013; Baietti et al., 2012), and plasma membrane (Jimenez et al., 2014) and endosome repair (Radulovic et al., 2018; Skowyra et al., 2018).

ALIX has an N-terminal BRO1 domain, a central V-shaped domain, and a flexible C-terminal proline-rich region (PRR; Fisher et al., 2007). ALIX BRO1 domain interacts with CHMP4 (Katoh et al., 2003; Strack et al., 2003), while the PRR binds tumor susceptibility gene 101 protein (TSG101) in ESCRT-I (Strack et al., 2003) and interacts with the BRO1 domain (Strack et al., 2003; Zhai et al., 2011; Zhou et al., 2008, 2009, 2010). The V-domain binds ubiquitin (Dowlatshahi et al., 2012) and the YPX$_3$L motif in the cytosolic region of the G protein–coupled receptors PAR1 and P2Y1 (Dores et al., 2012, 2016). We previously reported that ALIX interacts via an exposed loop in the BRO1 domain with lysobisphosphatidic acid (LBPA), also termed bis(monoacylglycero)phosphate (Bissig et al., 2013), an unconventional phospholipid found only in late endosomes and not

detected in other cellular membranes (Kobayashi et al., 1998). However, the role of ALIX in endolysosomal trafficking and biogenesis remains unclear. Here, we report that ALIX mediates the highly selective recruitment of ESCRT-III to late endosomes, in a process that depends on ALIX–LBPA interactions, and that this pathway may regulate the endosomal sorting of tetraspanins and their secretion via exosomes.

## Results

### Activated ALIX is primarily recruited to late endosomes containing LBPA

Previous studies suggested that overexpression of full-length ALIX can be detrimental because of ALIX proapoptotic activity (Trioulier et al., 2004; Wu et al., 2002), due to interactions between the PRR and the protein ALG-2 (Missotten et al., 1999; Vito et al., 1999), which is necessary for cell death (Vito et al., 1996). The PRR of ALIX also functions as an autoinhibitory domain by interacting with the BRO1 domain, keeping ALIX in a closed, autoinhibited conformation (Fig. 1 A; Strack et al., 2003; Zhai et al., 2011; Zhou et al., 2008, 2009, 2010). When expressed in HeLa cells, both full-length ALIX and ALIX without the PRR (ALIXΔPRR), tagged with mCherry, showed a cytosolic and punctate distribution (Fig. 1 B). However, the punctate staining was stronger with ALIXΔPRR than with WT ALIX, presumably because deletion of the autoinhibitory domain in ALIXΔPRR facilitates membrane association. Also, ALIXΔPRR retains the capacity to undergo dimerization, a process presumably required for membrane association (Bissig et al., 2013; Pires et al., 2009). ALIXΔPRR-mCherry colocalized to a very large extent with the late endosomal lipid LBPA, and to a much lesser extent with the early endosomal marker early endosome antigen 1 (EEA1; Fig. 1 C; quantification in Fig. 1 D), consistent with the fact that ALIX membrane association is primarily LBPA dependent (Bissig et al., 2013; Matsuo et al., 2004).

### ALIX recruits ESCRT-III on late endosomes

Since ALIX binds ESCRT-III through its BRO1 domain (Katoh et al., 2003), we investigated whether ALIXΔPRR-mCherry was able to interact with ESCRT-III in vivo. To this end, we overexpressed ALIXΔPRR-mCherry in cells stably expressing GFP-CHMP4B at low, endogenous levels (Poser et al., 2008; Fig. S1, A and B). While GFP-CHMP4B was primarily cytosolic and nucleoplasmic in nontransfected cells (Fig. S1 A), the intensity and number of CHMP4B-labeled puncta increased severalfold upon overexpression of ALIXΔPRR-mCherry, and both proteins colocalized on the same structures (Fig. 1 E; quantification in Fig. 1 F). Expression levels of CHMP4B and GFP-CHMP4B were not affected by ALIXΔPRR expression (Fig. 1 G), showing that the intense puncta were caused by GFP-CHMP4B relocalization. Consistent with light microscopy observations (Fig. 1, E and F), both CHMP4B and GFP-CHMP4B also increased several fold in light membrane fractions containing endosomes after fractionation of cells expressing ALIXΔPRR-mCherry (Fig. 1 G; quantification in Fig. 1 H) or full-length ALIX (Fig. S1, C and D). Membrane recruitment was more efficient with ALIXΔPRR, presumably because of the auto-inhibitory function of PRR

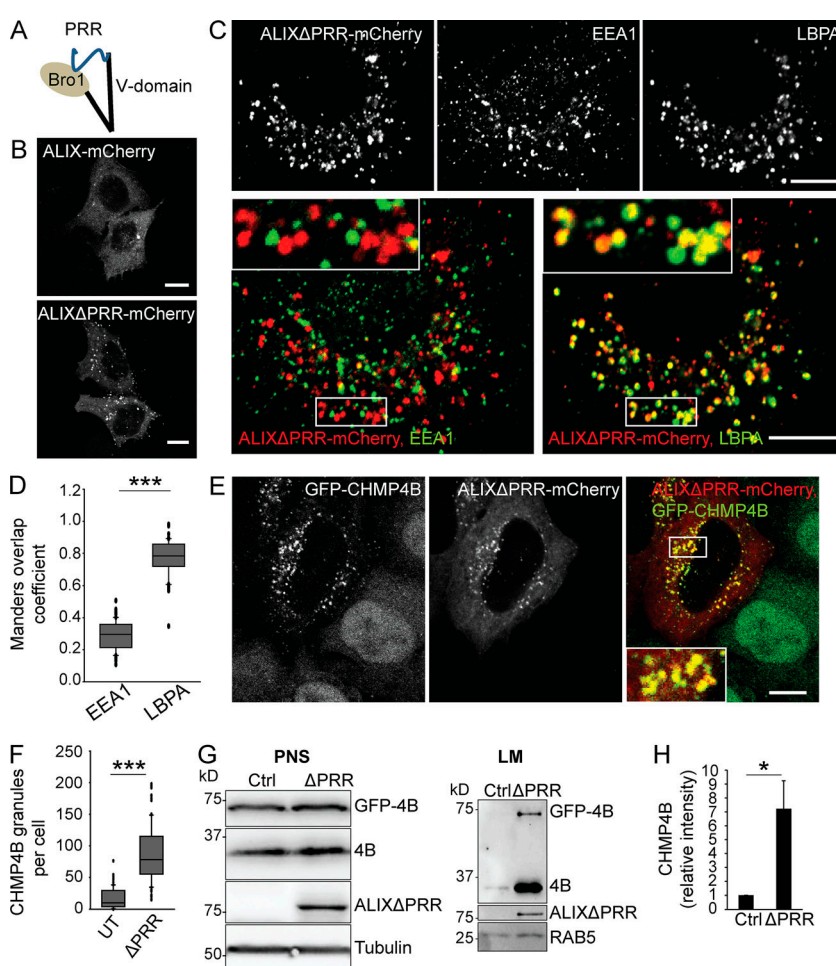

Figure 1. **ALIXΔPRR recruits the ESCRT-III protein CHMP4B. (A)** Outline of ALIX organization into three structural domains: the BRO1 domain, the V-domain, and the autoinhibitory PRR. Experiments were performed using ALIXΔPRR, lacking PRR. **(B)** HeLa-MZ cells were transfected with ALIX-mCherry or ALIXΔPRR-mCherry and imaged by confocal microscopy. Scale bar: 10 μm. **(C and D)** As in B, but cells were labeled with antibodies against LBPA and EEA1 (C). The boxed areas in the merged images are shown in higher magnification. The fraction of endosomes containing ALIXΔPRR-mCherry and EEA1 or LBPA was quantified by calculating the Manders overlap coefficients (D). Box plot, median (box central line); 25% and 75% percentiles (box edges); 10% and 90% percentiles (whiskers); outliers (black circles). Mann–Whitney $U$ test; ***, P < 0.001; n = 50 cells from three independent experiments; scale bar: 10 μm. **(E and F)** HeLa GFP-CHMP4B cells transfected with ALIXΔPRR-mCherry were imaged (E) by confocal microscopy. The number of CHMP4B granules was quantified (F) in cells expressing ALIXΔPRR-mCherry (ΔPRR) and untransfected cells (UT). Box plot as in D; n = 90 cells from three independent experiments. Scale bar: 10 μm. Pearson correlation coefficient for ALIX and CHMP4B in E: 0.72, n = 50 cells. **(G and H)** HeLa GFP-CHMP4B cells transfected for 18 h with ALIXΔPRR (ΔPRR) or an empty vector as control (ctrl) were fractionated by flotation in a sucrose density gradient. The postnuclear supernatant (PNS) and the light membranes (LM) were analyzed by Western blotting (G) using antibodies against CHMP4B, ALIX, and tubulin and RAB5 (equal loading controls). CHMP4B quantification in LM fractions by densitometry (H), relative to RAB5. Boxes, mean; error bars, ±SD (n = 3, from three independent experiments); t test, ctrl versus ΔPRR; *, P < 0.05.

(Fig. S1, C and D). GFP-CHMP4B then colocalized with ALIXΔPRR-mCherry and the late endocytic marker lysosome-associated membrane glycoprotein 1 (LAMP1; Fig. 2 A). Moreover, overexpression of the Ras-related protein 7 (RAB7) effector RILP (RAB-interacting lysosomal protein), which clusters late endosomes at the microtubule-organizing center (Cantalupo et al., 2001), also clustered ALIXΔPRR in perinuclear structures (Fig. 2 B) that contained RILP itself (Fig. 2 B), GFP-CHMP4B, and LAMP1 (Fig. 2 C). Our results thus show that ALIXΔPRR relocalizes CHMP4B to late endosomal membranes.

The association of GFP-CHMP4B and ALIXΔPRR-mCherry to endosomal membranes was dynamic, and thus functional, since the fluorescence of GFP-CHMP4B and ALIXΔPRR-mCherry on endosomes was partially recovered after photobleaching (Fig. 2 D; quantification in Fig. 2 E). These observations are consistent with the notion that ESCRT-III polymerization and membrane remodeling activity involve VPS4-dependent turnover of individual subunits (Adell et al., 2014, 2017; Mierzwa et al., 2017).

We then investigated whether ALIXΔPRR also recruited other ESCRT-III subunits onto endosomes. CHMP1A-V5 and CHMP1B-Flag exhibited primarily a cytosolic distribution without ALIXΔPRR (Fig. 3, A and B). Much like CHMP4B, both proteins were efficiently recruited onto endosomes containing ALIXΔPRR-mCherry (Fig. 3, C and D). Similarly, both CHMP4A and CHMP3 were enriched in light membrane fractions prepared

from cells expressing ALIXΔPRR (Fig. 3, E and F; antibodies against CHMP4A and CHMP3 did not work by immunofluorescence). Additionally, VPS22, an ESCRT-II component, and VPS4B were also increased in light membranes upon ALIXΔPRR expression. CHMP6 (VPS20 in yeast) has been proposed to nucleate CHMP4/SNF7 on endosomal membranes (Saksena et al., 2009; Tang et al., 2015; Teis et al., 2008). However, a Western blot analysis of light membranes did not show increased CHMP6 levels upon ALIXΔPRR expression, compared with controls (Fig. 3, E and F), suggesting that ALIX-dependent ESCRT-III recruitment is CHMP6 independent, and thus that ALIX and CHMP6 may function along parallel pathways. Similarly, hepatocyte growth factor-regulated tyrosine kinase substrate (HRS) and TSG101, subunits of ESCRT-0 and -I, respectively, were not significantly enriched in the endosomal fraction upon ALIXΔPRR expression. Altogether, these data demonstrate that ALIXΔPRR causes the constitutive and specific recruitment of ESCRT-III subunits on the late endosomes.

## ALIX-dependent membrane association of ESCRT-III depends on LBPA

The I212D mutation in ALIX BRO1 domain disrupts ALIX/ESCRT-III interactions (Fisher et al., 2007; McCullough et al., 2008). Consistently, the ALIXΔPRR-I212D mutant, in contrast to ALIXΔPRR, was unable to recruit GFP-CHMP4B onto endosomal

Figure 2. **ALIXΔPRR recruits ESCRT-III onto late endosomes. (A)** HeLa GFP-CHMP4B cells transfected with ALIXΔPRR-mCherry were stained with anti-LAMP1 antibodies (boxed area: higher-magnification views). Cells were permeabilized with saponin before fixation to reduce the cytosolic staining. Arrows point at endosomes containing ALIXΔPRR, CHMP4B, and LAMP1. **(B)** HeLa-MZ cells were transfected with ALIXΔPRR-mCherry and GFP-RILP (boxed area: higher-magnification view). **(C)** HeLa GFP-CHMP4B cells transfected with RILP and ALIXΔPRR-mCherry were labeled with antibodies against LAMP1. **(D and E)** HeLa GFP-CHMP4B cells transfected with ALIXΔPRR-mCherry were treated with 10 µM nocodazole for 2 h (to limit endosome movement), and endosomes containing both markers were analyzed by FRAP. White circles, photobleached regions (D); time 0, before photobleaching. Fluorescence recovery calculated for endosomes containing GFP-CHMP4B and ALIXΔPRR-mCherry (E). Dots, mean; shaded area, ±SD (n = 28 endosomes from three independent experiments). Scale bars (A–D): 10 µm.

membranes, as monitored by fluorescence microscopy (Fig. 4 A; quantification in Fig. 4 B) and subcellular fractionation (Fig. 4, C and D; quantification in Fig. 4 E). In addition, ALIXΔPRR-I212D had a strictly cytosolic distribution in cells expressing GFP-CHMP4B (Fig. 4 A). These observations indicate not only that ALIX recruits ESCRT-III highly specifically, but also that interactions with ESCRT-III and the polymerization of ESCRT-III filaments are necessary to stabilize ALIX onto endosomal membranes. Without ESCRT-III, ALIX is released and no longer remains membrane associated.

Similarly, we previously showed that mutation of the BRO1 domain hydrophobic residues L104 and F105 to glutamines

(ALIXΔPRR-QQ), which mediates ALIX interaction with LBPA, abolishes membrane association (Bissig et al., 2013). Much like ALIXΔPRR-I212D, ALIXΔPRR-QQ remained strictly cytosolic in cells expressing GFP-CHMP4B (Fig. 4 A) and was unable to recruit GFP-CHMP4B (Fig. 4, A–E). Hence, interactions with both LBPA and ESCRT-III are required for the ALIX-dependent recruitment of ESCRT-III onto late endosomal membranes in vivo.

### Both LBPA and ALIX are necessary to support ESCRT-III binding in vitro

Since ALIX binds late endosomes via LBPA and recruits ESCRT-III in vivo, we then investigated whether the process could be

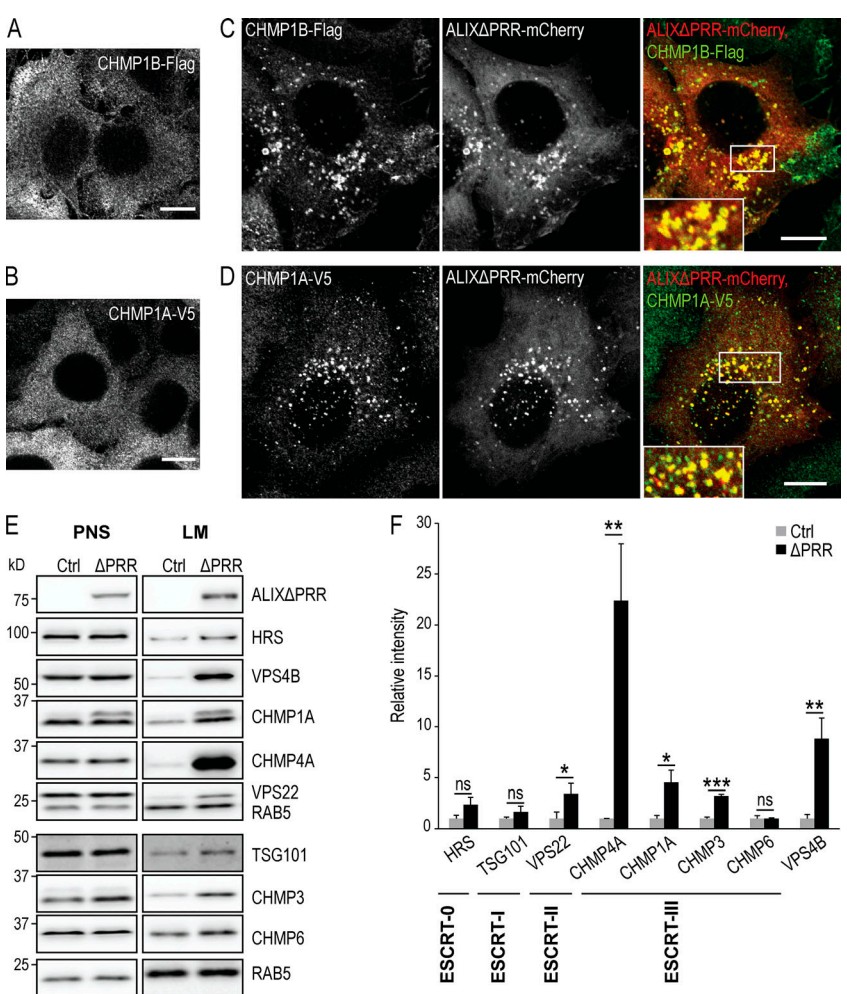

Figure 3. **ALIXΔPRR recruits VPS22 and ESCRT-III proteins to endosomes. (A–D)** HeLa cells stably expressing CHMP1B-Flag (A and C) or CHMP1A-V5 (B and D) were transfected (C and D) or not (A and B) with ALIXΔPRR-mCherry. Cells labeled with antibodies against Flag (A and C) and V5 (B and D) epitopes were analyzed by confocal microscopy (boxed areas: higher-magnification views). Scale bar: 10 μm. **(E and F)** HeLa GFP-CHMP4B cells transfected with ALIXΔPRR for 18 h were fractionated as in Fig. 1 G. The postnuclear supernatant (PNS) and light membranes (LM) fractions were analyzed by Western blotting with antibodies against the indicated proteins; RAB5, equal loading control. Quantification of each protein in LM fractions by densitometry (F), as in Fig. 1 H. Boxes, mean; error bars, ±SD ($n = 3$, from three independent experiments); $t$ test; *, $P < 0.05$; **, $P < 0.01$; ***, $P < 0.001$; ns, not significant.

reconstituted in vitro. Giant unilamellar vesicles (GUVs) were prepared with the phospholipid composition of late endosomes enriched in LBPA (phosphatidylcholine [DOPC]:phosphatidyl ethanolamine [DOPE]:PI:LBPA, 5:2:1:2 molar ratio; Kobayashi et al., 1998). GUVs were labeled with trace amounts of N-rhodamine PE. We then prepared supported bilayers using these GUVs, as previously established (Chiaruttini et al., 2015; Mierzwa et al., 2017). Supported bilayers were sequentially incubated with purified proteins, recombinant ALIX BRO1 domain, and Alexa Fluor 488 CHMP4B (Fig. S2).

In the absence of LBPA, no CHMP4B was bound to supported bilayers, whether ALIX-BRO1 was present or not (Fig. 4, F and H). Similarly, hardly detectable levels of CHMP4B could be recruited in the presence of ALIX-BRO1, when LBPA was replaced by PS, another negatively charged lipid, at the same molar ratio (DOPC:DOPE:PI:phosphatidylserine [DOPS], 5:2:1:2). By contrast, CHMP4B was massively recruited onto LBPA-containing bilayers, but only when ALIX-BRO1 was present (Fig. 4 F; quantification in Fig. 4 H). CHMP4B recruitment occurred with relatively rapid kinetics in vitro, as the apparent $t_{1/2}$ was ≈5 min (Fig. 4 G; boxed area in Fig. 4 F). Finally, much like in vivo (Fig. 4, A and E), CHMP4B failed to be recruited onto LBPA-containing bilayers in the presence of ALIX-BRO1-I212D, defective in CHMP4B binding, or ALIX-BRO1-QQ, defective in LBPA binding

(Fig. 4 I; quantification in Fig. 4 J). Altogether, these observations demonstrate that CHMP4B recruitment onto the bilayer can be fully recapitulated in vitro, provided that LBPA and ALIX-BRO1 are present.

**Other ESCRTs or ESCRT-related proteins are dispensable for ALIX-dependent membrane association of CHMP4B in vivo**
Our results show that CHMP4B recruitment onto late endosomal membranes depends on LBPA and ALIX. We then investigated whether other ESCRT-III binding partners also play a role in ALIX-dependent CHMP4B membrane association in vivo, including the ESCRT-0 subunits signal transducing adapter molecule (STAM) 1/2 and the ESCRT-I subunit TSG101, which binds ALIX. We also tested the possible role of the ESCRT-II subunit VPS22 and the ESCRT-III proteins CHMP3 and CHMP6. Finally, we tested the possible involvement of other BRO1 domain-containing proteins HD-PTP and BROX (Fig. 5). Each protein was depleted using a pool of siRNA target sequences to limit off-target effects, except ALIX, which was depleted using a well-characterized single siRNA (Bissig et al., 2013). After knockdown (KD), each protein level was reduced 5–10 times, except STAM1, which was reduced ~3 times (Fig. 5 C).

ALIXΔPRR was fully capable of redistributing GFP-CHMP4B from cytosol to endosomal membranes upon depletion of any protein we tested (Fig. 5 A). Indeed, much like in mock-treated

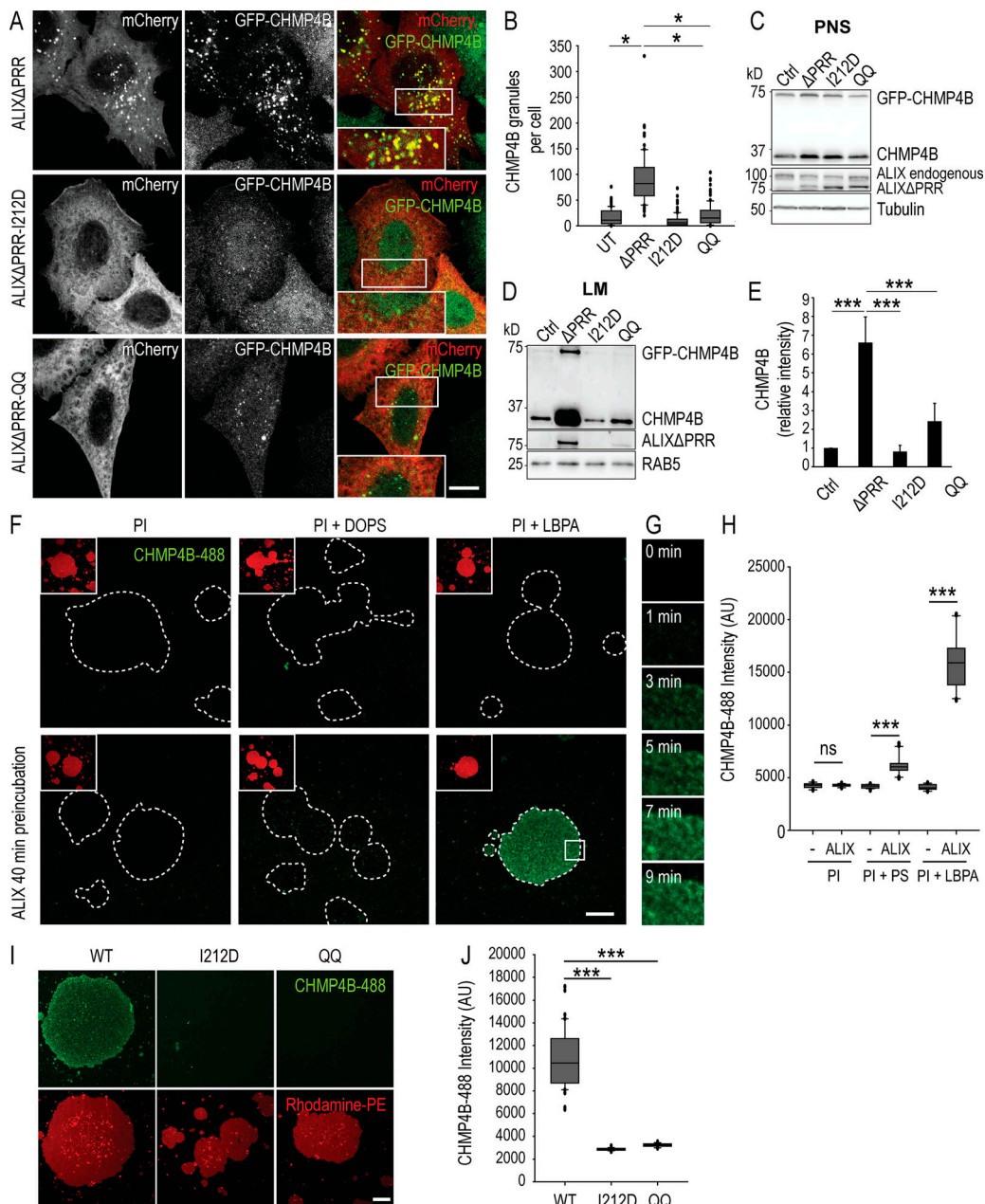

**Figure 4. CHMP4B recruitment to endosomes depends on the CHMP4B and LBPA binding sites in ALIX BRO1 domain and can be recapitulated in vitro. (A and B)** HeLa GFP-CHMP4B cells were transfected with ALIXΔPRR-mCherry (ΔPRR), ALIXΔPRR-I212D-mCherry (I212D), and ALIXΔPRR-QQ-mCherry (QQ); boxed areas: higher-magnification views. Scale bar: 10 μm. The number of CHMP4B granules was quantified and analyzed (B) as in Fig. 1 F. Mann–Whitney U test; *, P < 0.05; n = 90 cells from three independent experiments; UT, untransfected. **(C–E)** HeLa GFP-CHMP4B transfected with ALIXΔPRR (ΔPRR), ALIXΔPRR-I212D (I212D), ALIXΔPRR-QQ (QQ), or an empty vector (Ctrl) were as in Fig. 1 G. The PNS (C) and LM (D) fractions were analyzed by Western blotting with antibodies against CHMP4B and ALIX; tubulin and RAB5, loading controls. Quantification of each protein in LM fractions by densitometry (E), as in Fig. 1 H. Boxes, mean; error bars, ± SD (n = 3, from three independent experiments); t test; ***, P < 0.001. **(F–H)** Supported bilayers prepared with the lipid composition: PI (DOPC:DOPE:PI, 6.99:2:1 mol), PI + PS (DOPC:DOPE:PI:DOPS, 4.99:2:1:2 mol), and PI + LBPA (DOPC:DOPE:PI:LBPA, 4.99:2:1:2 mol) and labeled with N-Rhodamine PE (red) were preincubated (F, bottom row) or not (F, top row) with ALIX BRO1 domain for 40 min, and then for 25 min with CHMP4B-488 (green), and analyzed by time-lapse double-channel confocal microscopy. Each panel in F corresponds to a view in the green (CHMP4B-488) channel after 25 min; boxed areas, smaller view of the same field in the red channel (N-Rhodamine PE); dashed lines, areas covered with lipids. The time course of CHMP4B-488 association to LBPA-containing bilayers preincubated with ALIX is shown in G, corresponding to a magnified view of the boxed area in F. The intensity of CHMP4B-488 fluorescence was quantified on supported bilayers after 25 min (H). Box plot as in Fig. 1 D; Mann–Whitney U test; ***, P < 0.001; n = 20 lipid patches from two independent experiments; ns, not significant; scale bar: 30 μm. **(I and J)** The experiments were as in F using supported bilayers with the PI + LBPA composition only, except that I212D and QQ mutants in the ALIX BRO1 domain were used (I) instead of ALIX BRO1 domain. **(J)** Box plot as in Fig. 1 D; Mann–Whitney U test; ***, P < 0.001; n = 40 lipid patches from two independent experiments; scale bar: 30 μm.

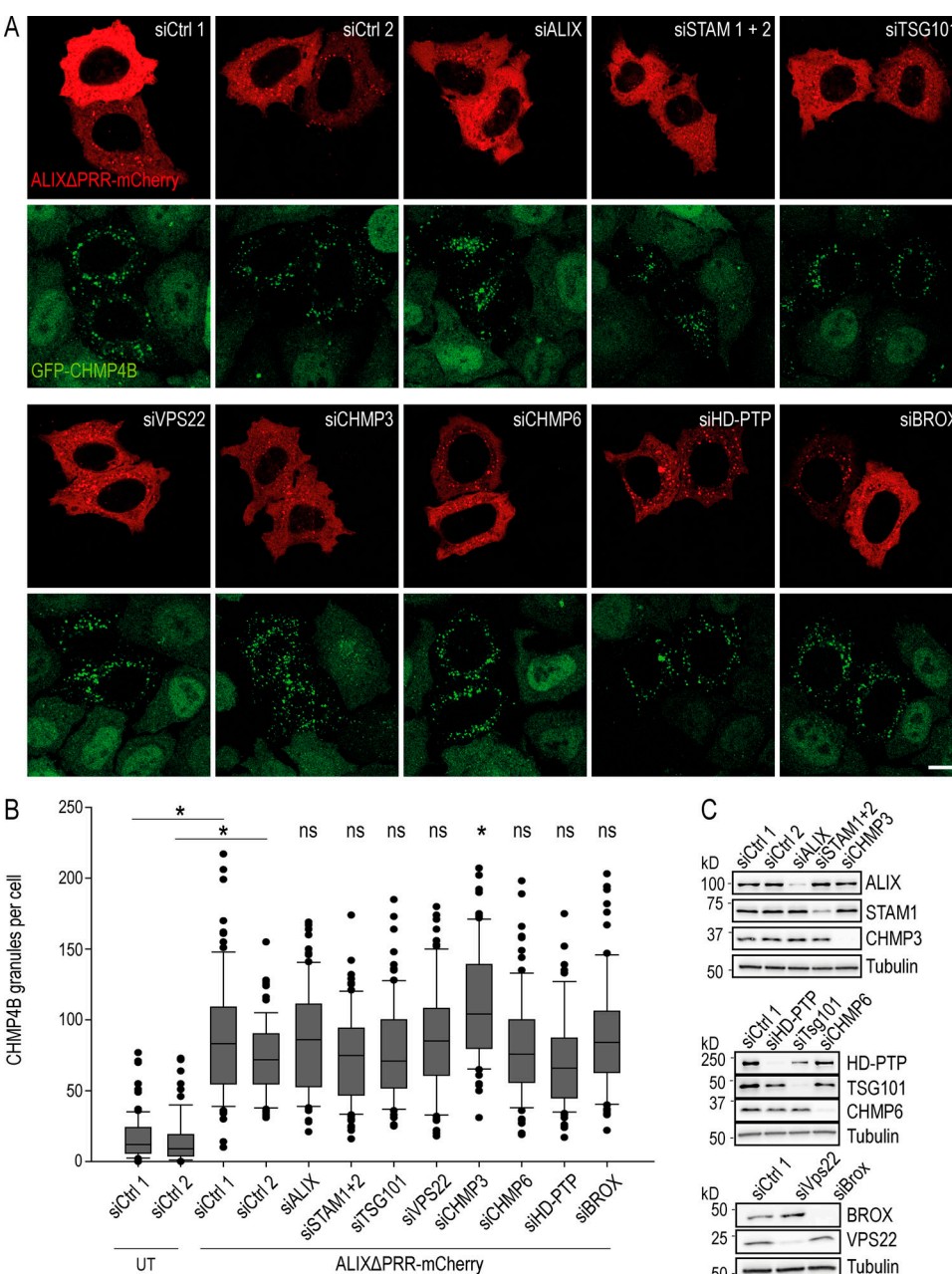

Figure 5. **CHMP4B recruitment by ALIXΔPRR is independent of other ESCRT proteins. (A and B)** HeLa GFP-CHMP4B (green) cells were transfected with an siRNA-resistant mutant of ALIXΔPRR-mCherry (red), and treated with siRNAs against the indicated proteins; siCtrl 1, siRNA control pool used for siRNA pools against all targets except ALIX; siCtrl 2, single siRNA sequence used as control for the single siRNA against ALIX. Scale bar: 10 μm. The number of CHMP4B granules was quantified (B) as in Fig. 1 F. Box plot as in Fig. 1 D; Mann–Whitney U test; *, P < 0.05; n = 83 cells from three independent experiments; ns, not significant; UT, untransfected. **(C)** The KD efficiency of the different targets in A and B analyzed by Western blotting using antibodies against the indicated proteins. Tubulin was used as an equal loading control.

controls, CHMP4B-GFP exhibited after each KD the same characteristic punctate pattern as in Fig. 1, including after ALIX KD in the presence of ectopically expressed ALIXΔPRR (Fig. 5 A). The quantification confirmed that none of the KDs had significant effect on GFP-CHMP4B membrane association (Fig. 5 B). Only a slight increase in GFP-CHMP4B recruitment was observed in CHMP3 KD cells, perhaps because of CHMP3 functions in ESCRT-III disassembly (Babst et al., 2002a; Lata et al., 2008). Altogether, our in vitro and in vivo data indicate that LBPA and

ALIX are necessary for the recruitment of ESCRT-III onto late endosomal membranes. We conclude that, in addition to the canonical mechanism dependent on ESCRT-0, ESCRT-I, and ESCRT-II, ALIX functions as an alternative mechanism to recruit ESCRT-III selectively onto late endosomes containing LBPA.

### Role of ALIX and ESCRT-III in cargo sorting
Protein sorting into ILVs depends mostly on ubiquitination (Hicke and Riezman, 1996; Katzmann et al., 2001; Kölling and

Hollenberg, 1994), and ubiquitinated proteins accumulate in endosomes after depletion or overexpression of the ESCRT proteins HRS, TSG101, and VPS28 (Bishop et al., 2002). Strikingly, an analysis by immunofluorescence revealed that conjugated ubiquitin accumulated in LAMP1-positive late endosomes containing ALIXΔPRR and GFP-CHMP4B (Fig. 6, A and B), while conjugated ubiquitin was hardly detected, if at all, in the endosomes of cells that expressed ALIXΔPRR-I212D and ALIXΔPRR-QQ or untransfected cells (Fig. 6 C; quantification in Fig. 6 D; note that cells were prepermeabilized with saponin before fixation to reduce the cytosolic staining). In addition, cell fractionation showed accumulation of ubiquitinated proteins in the light membranes of cells expressing ALIXΔPRR (Fig. 6 E; quantification in Fig. 6 F; transfection efficiency ≈70%). Altogether, these data indicate that ALIXΔPRR expression causes the accumulation of some ubiquitinated cargo proteins in endosomes.

Next, to evaluate the possible role of ALIXΔPRR in this pathway, we followed EGFR, the canonical ubiquitinated cargo targeted by ESCRTs to lysosomes for degradation (Raiborg and Stenmark, 2009). The total levels of EGFR in light membranes were not affected by ALIXΔPRR expression (Fig. 7 A; quantification in Fig. 7 B). Similarly, the kinetics of EGF receptor degradation in cells treated with EGF were essentially identical whether ALIXΔPRR was present or not (Fig. 7 C; quantification in Fig. 7 D). This finding agrees with previous studies showing that EGFR lysosome targeting is ALIX independent (Bowers et al., 2006; Doyotte et al., 2008; Luyet et al., 2008), although this notion has been challenged (Schmidt et al., 2004). Consistent with these observations, EGFR distribution and degradation after EGF addition was not affected in cells expressing ALIXΔPRR-mCherry, when analyzed by automated fluorescence microscopy (Fig. 7 E; quantification in Fig. 7 F). Finally, the levels of mature cathepsin D heavy chain were not affected by ALIXΔPRR expression (Fig. 7 G; quantification in Fig. 7 H); neither was the pH of endosomes and lysosomes when compared with untransfected cells using a pH-sensitive probe (Fig. 7 I; quantification in Fig. 7 J). Altogether, these data indicate that both traffic to the lysosomes and the degradation capacity of lysosomes were not affected by ALIXΔPRR, and thus suggest that the expression of ALIXΔPRR does not cause a general traffic jam in the late endosomal pathway, but rather results in the selective retention of a subset of ubiquitinated cargoes.

## Exosomal protein secretion depends on ALIX

In addition to its intracellular functions, ALIX is also one of the best-established exosome markers (Kalra et al., 2012; Théry et al., 2009; van Niel et al., 2018). ALIX, together with ESCRT proteins, also plays a role in exosome biogenesis (Abrami et al., 2013; Baietti et al., 2012), consistent with the hypothesis that exosomes correspond to a subpopulation of ILVs released into the medium upon endosome fusion with the plasma membrane. We thus wondered if ALIX overexpression, by causing the specific accumulation of proteins in the endosomes, could have an impact on the biogenesis of exosomes.

Exosomes were prepared from HeLa cells expressing ALIXΔPRR or GFP, as a control, using a well-established protocol of differential centrifugations (Théry et al., 2006), and their

protein composition was analyzed and compared by Western blotting. As expected (Kalra et al., 2012; Théry et al., 2009; van Niel et al., 2018), both endogenous WT ALIX (full length) and ALIXΔPRR were recovered in the exosomal fractions (Fig. 8 A). Strikingly, however, ALIXΔPRR was sorted into exosomes far more efficiently than WT ALIX. Indeed, the yield of exosomal ALIXΔPRR, corresponding to ≈0.4% of the total cell lysate, was sixfold higher compared with the endogenous full-length protein (Fig. 8 A). By contrast, the Golgi-resident protein GM130 was not detected in the exosomal fractions, whether ALIXΔPRR was expressed or not (Fig. 8 A); neither was free GFP, when using GFP-expressing cells or CHMP4B, and only background levels of the abundant ER protein calnexin (Fig. S3 A), demonstrating the efficacy of the fractionation protocol.

## ALIX stimulates the secretion of exosomes containing CD9, CD63, and CD81

In addition to ALIX, the tetraspanins CD9, CD81, and CD63 are also among the best-established markers of exosomes (Kalra et al., 2012; Théry et al., 2009; van Niel et al., 2018), and consistently these proteins were detected in exosomal fractions (Fig. 8 A). Notably, we found that the secretion of CD9, CD81, and CD63 was significantly increased after ALIXΔPRR expression, compared with control (Fig. 8 A; quantification in Fig. 8 B). Conversely, CD9, CD63, and to some extent, CD81 did not accumulate in exosomes upon overexpression of the ALIXΔPRR-QQ mutant, defective in LBPA binding (Fig. S3, B–D), consistent with observations that this mutant was unable to recruit GFP-CHMP4B onto endosomes in vivo (Fig. 4, A–E) and onto supported bilayers in vitro (Fig. 4, I and J). By contrast, ALIXΔPRR-QQ had little if any effect on the presence of EGFR in exosomes (Fig. S3, B–D). Moreover, ALIX depletion with siRNAs specifically decreased the exosomal secretion of CD9, CD81, and CD63 (Fig. 8 C; quantification in Fig. 8 D). Similarly, CD9, CD81, and CD63 secretion was also decreased after depletion of the ESCRT-III nucleator CHMP6 (Fig. 8 E; quantification in Fig. 8 F), supporting the view that both CHMP6- and ALIX-dependent ESCRT-III nucleation mechanisms are involved in exosome biogenesis.

Given our observations that ALIXΔPRR expression caused the accumulation of ubiquitinated proteins in endosomes (Fig. 6) and stimulated the production of exosomes containing CD9, CD81, and CD63 (Fig. 8, A and B), we wondered whether protein sorting along the exosomal pathway involved protein ubiquitination. To this end, we made use of the fact that CD9 cytoplasmic regions only contain three Lys residues to generate a mutant that cannot be ubiquitinated by replacing these Lys residues with Arg (CD9/3R). Strikingly, the levels of a CD9/3R were significantly reduced in exosomes compared with WT CD9 (Fig. 8 G; quantification in Fig. 8 H). This defective incorporation could be partially rescued by ALIXΔPRR overexpression. The overexpression of ALIXΔPRR might have triggered the release of CD9/3R-containing microvesicles from the plasma membrane. However, this is quite unlikely, given that ALIXΔPRR is not detected on the plasma membrane and that ALIXΔPRR membrane association requires an intact binding site for the late endosome lipid LBPA. Alternatively, the incorporation of the

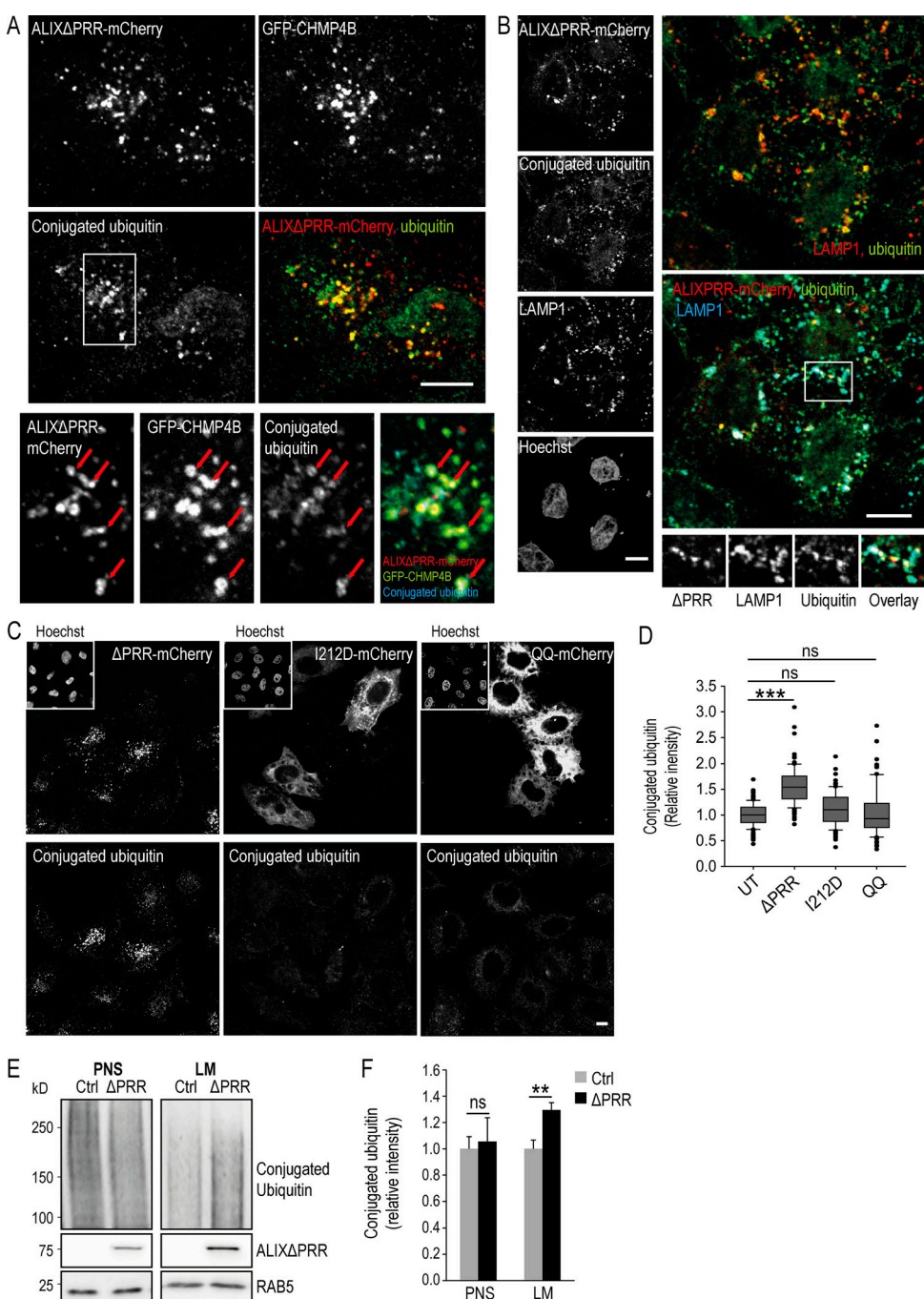

Figure 6. **ALIXΔPRR induces the endosomal accumulation of ubiquitinated proteins. (A)** HeLa GFP-CHMP4B cells were transfected with ALIXΔPRR-mCherry, permeabilized with saponin 0.01% in PBS before fixation, and labeled with an antibody against conjugated ubiquitin (boxed area: higher-magnification view). Arrows point at endosomes containing ALIXΔPRR, CHMP4B, and conjugated ubiquitin. Scale bar: 10 μm. **(B)** HeLa-MZ cells processed as in A were labeled with antibodies against conjugated ubiquitin and LAMP1, and nuclei were stained with Hoechst. Scale bar: 10 μm. **(C and D)** The nuclei of HeLa cells treated as in A were stained with Hoechst to illustrate the accumulation of ubiquitinated proteins in cells expressing ALIXΔPRR-mCherry, compared with cells expressing ALIXΔPRR-I212D or ALIXΔPRR-QQ (C). Scale bar: 10 μm. The ubiquitin intensity per cell was quantified (D) in cells expressing ALIXΔPRR-mCherry (ΔPRR) and compared with either mutant. Box plot as in Fig. 1 D; Mann–Whitney $U$ test; $n$ = 135 cells from three independent experiments. ns, not significant. ***, P < 0.001. **(E and F)** HeLa GFP-CHMP4B cells transfected with ALIXΔPRR (ΔPRR) or an empty vector (Ctrl) were fractionated as in Fig. 1 G. The postnuclear supernatant (PNS) and light membranes (LM) fractions were analyzed by Western blotting (E) with antibodies against conjugated ubiquitin and ALIX; RAB5, equal loading control. The relative amounts of conjugated ubiquitin in PNS and LM were quantified by densitometry (F), using RAB5 intensity to normalize the signal. Boxes, mean; error bars, ±SD ($n$ = 3, from three independent experiments); $t$ test; **, P < 0.01.

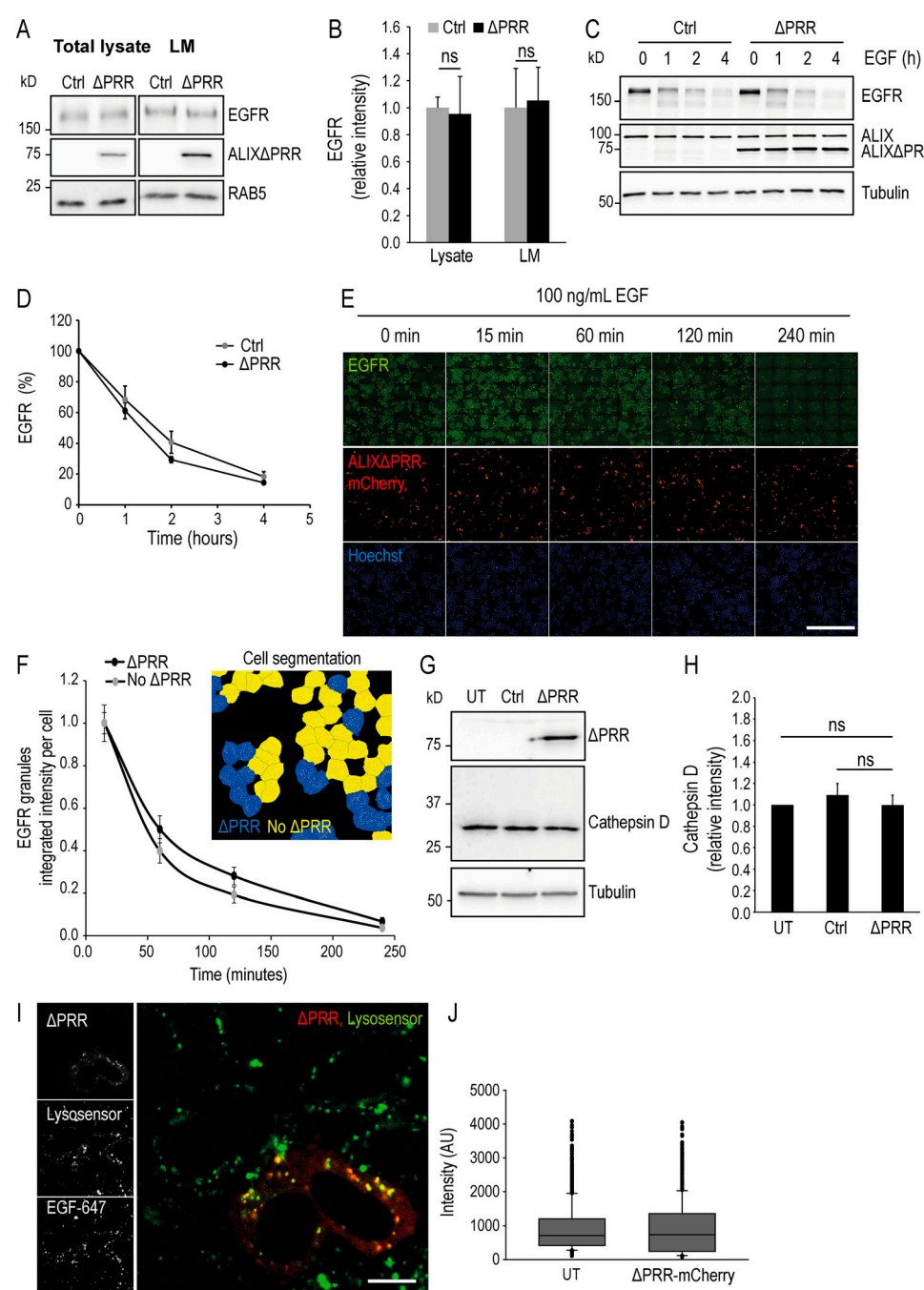

Figure 7. **ALIXΔPRR does not affect EGFR transport to lysosome and degradation, lysosomal cathepsin D maturation, and endolysosomal pH.** **(A and B)** HeLa GFP-CHMP4B cells transfected for 18 h with ALIXΔPRR (ΔPRR) or an empty vector (Ctrl) were fractionated as in Fig. 1 G (≈70% cells were transfected). The total cell lysate and LM (A) were analyzed by Western blotting with antibodies against EGFR and ALIX; RAB5, equal loading control. The RAB5 and ALIXΔPRR panels (A) are the same as in Fig. 6 E, because the same membrane was used for blots against conjugated ubiquitin (Fig. 6 E) and EGFR (A). The relative amounts of EGFR in total cell lysate and light membranes (LM) was quantified by densitometry (B), using RAB5 intensity to normalize the EGFR signal. Boxes, mean; error bars, ±SD (*n* = 3, from three independent experiments); *t* test; ns, not significant. **(C and D)** HeLa GFP-CHMP4B cells were transfected for 18 h with ALIXΔPRR (ΔPRR) or an empty vector (Ctrl; ≈70% cells were transfected). Cells were sequentially incubated at 37°C for 3 h in serum-free medium, for 1 h with 10 µg/ml cycloheximide, and finally for the indicated time with 100 ng/ml EGF. Samples were collected and analyzed by Western blotting with antibodies against EGFR and ALIX; tubulin, equal loading control (C). The relative amount of EGFR was quantified by densitometry (D), using tubulin intensity to normalize the EGFR signal. Circles, mean; error bars, ±SD (*n* = 3, from three independent experiments). *t* test. There is no significant difference between Ctrl and ΔPRR for the different time points. **(E and F)** The experiments were as in C and D, except that cells were transfected with ALIXΔPRR-mCherry and analyzed by automated triple-channel fluorescence microscopy. In contrast to A–D, 30–40% cells were transfected to facilitate the compared analysis of transfected versus untransfected cells. Each row represents the imaged area obtained from one 96-well plate (E). Scale bar: 1 mm. For each time point, >12,000 cells were imaged, and ALIXΔPRR-mCherry positive (ΔPRR) or negative (no ΔPRR) cells were segmented using mCherry signal (insert in F). The endosomal EGFR was also segmented (EGFR granules), and the average EGFR granule intensity per cell was plotted for each time point (F). Because at time 0 min, EGFR is mostly at the plasma membrane, 15 min was used as first time point. Circles, mean; error bars, ± SD; *t* test. There is no significant difference between untransfected and

ALIXΔPRR-mCherry transfected cells. **(G and H)** HeLa GFP-CHMP4B were untransfected (UT) or transfected for 18 h with an empty vector (Ctrl) or ALIXΔPRR (ΔPRR). Total cell lysates were analyzed by Western blotting with antibodies against ALIX or cathepsin D (G); tubulin, equal loading control. The relative amounts of cathepsin D were quantified by densitometry (H), using tubulin intensity to normalize the signal. Boxes, mean; error bars, ±SD ($n = 3$, from three independent experiments); $t$ test; ns, not significant. **(I and J)** Cells transfected with ALIXΔPRR (ΔPRR) as in E and F were treated with EGF-647 for 45 min at 37°C to label endosomes, and then with Lysosensor to label acidic compartments. Cells were then analyzed by automated microscopy as in E and F, and the intensity of the lysotracker signal per cell (J) was quantified. Box plot as in Fig. 1 D; $n = 1,000$ endosomes from four independent experiments. AU, arbitrary units. Scale bar: 10 μm.

CD9/3R mutant into exosomes was facilitated by interactions of CD9/3R with WT CD9 and other tetraspanins (Berditchevski and Odintsova, 2007; Perez-Hernandez et al., 2013; van Deventer et al., 2017) upon stimulation of exosome production (Fig. 8 A).

The enrichment of tetraspanins in exosomes driven by ALIXΔPRR expression was highly specific, as the levels of integrins α6 and β3, which form multiprotein complexes with tetraspanins at the plasma membrane (Berditchevski et al., 1996; Yu et al., 2017), did not depend on expression of ALIXΔPRR (Fig. 8, A and B) or ALIXΔPRR-QQ (Fig. S3, B–D). Similarly, EGFR could be detected in exosomes by others (Higginbotham et al., 2016) and us (Fig. 8 A), but the amounts in exosomes were unchanged upon ALIXΔPRR expression and only marginally affected by the ALIXΔPRR-QQ mutant (Fig. 8, A and B; and Fig. S3, B–D). We then investigated to what extent ALIX also regulated the incorporation of other exosomal proteins into exosomes. The secretion of syntenin, which interacts with ALIX during exosome biogenesis (Baietti et al., 2012), although unaffected by ALIXΔPRR (Fig. 8, A and B), was reduced by overexpression of the ALIXΔPRR-QQ mutant (Fig. S3, B–D) and by depletion of ALIX by RNAi (Fig. 8, C and D), as previously shown (Baietti et al., 2012). By contrast, flotillin-1 was not sensitive to ALIX overexpression or depletion (Fig. 8, A–D). These data are consistent with the notion that protein sorting into exosomes is mediated by more than one mechanism, and/or that cells produce more than one population of exosomes (Tkach and Théry, 2016; van Niel et al., 2018).

### Tetraspanin intracellular distribution

Since the exosomal secretion of tetraspanins was increased by ALIXΔPRR expression, we wondered whether the endosomal contents were affected. To this end, we quantified CD81 and CD63 in late endosomes labeled with the late endosomal marker LAMP1, by high-throughput automated microscopy; we were not able to monitor CD9 adequately with available antibodies. In the untransfected cells, CD63 was abundant in endolysosomes containing LAMP1, as expected under steady-state conditions (Kobayashi et al., 2002; Lebrand et al., 2002; Fig. 9 A; in these experiments, the transfection rate was ≈30% so that untransfected and transfected cells could be easily captured in the same field). Strikingly, the levels of CD63 in LAMP1-containing endolysosomes were significantly reduced in cells expressing ALIXΔPRR-mCherry (Fig. 9 A), as better visualized by displaying CD63 staining within LAMP1-positive endolysosomes only (the mask used in automated microscopy), after subtraction of the staining present elsewhere (Fig. 9 A, quantification in Fig. 9 B). By contrast, CD81 was found mostly at the plasma membrane in control cells, with some present in LAMP1-positive endolysosomes (Fig. 9 A), as expected (Brankatschk et al., 2011; Levy et al.,

1998). Much like CD63, CD81 was reduced in the LAMP1-containing endolysosomes of cells expressing ALIXΔPRR-mCherry, albeit not to the same extent (Fig. 9, A and B). The difference between CD63 and CD81 after ALIXΔPRR-mCherry expression likely reflects the difference in the steady-state distribution of these proteins. In any case, these data agree very nicely with our observations that tetraspanin exosomal secretion is stimulated by ALIXΔPRR (Fig. 8, A and B).

The distribution of EGFR, syntenin, and flotillin-1 was affected marginally, if at all, after ALIXΔPRR expression (Fig. S4; quantification in Fig. 9 B), much like their incorporation in exosomes (Fig. 8, A and B). The ER protein calnexin and the Golgi protein GM130 were not detected in LAMP1-containing endolysosomes, as expected, and this (absence of) distribution was not affected by ALIXΔPRR-mCherry expression (Fig. 9, A and B). Finally, as a control, we also analyzed the endosomal distribution of CD63 and CD81 after ALIX KD. Consistent with our observations that ALIX KD prevented the exosomal secretion of tetraspanins (Fig. 8, C and D), no difference was observed in their endosomal distribution (Fig. S5; quantification in Fig. 9 C). Similarly, ALIX KD had no effect on EGFR, syntenin, calnexin, and GM130 distribution, and perhaps some marginal effects on flotillin-1 (Fig. S5, A and B; quantification in Fig. 9 C; note that, like GM130 or calnexin, flotillin-1 is essentially absent from LAMP1 endosomes with or without ALIX).

We conclude that, concomitant with the observed increase in the exosomal secretion of tetraspanins (Fig. 8, A and B), ALIXΔPRR expression causes a highly selective decrease in the endosome levels of tetraspanins (Fig. 9, A and B). We also conclude that ALIX recruits ESCRT-III onto late endosomes containing LBPA and triggers the formation of ILVs containing CD9, CD81, and CD63, which are destined to be released as exosomes.

## Discussion

### ALIX recruits the ESCRT proteins to late endosomes

Our previous studies showed that the ALIX BRO1 domain contains a lipid-binding loop that interacts with LBPA, and that this interaction is important for ALIX binding to endosomes (Bissig et al., 2013). We now find that ALIXΔPRR binds more efficiently to late endosomes containing LBPA than the full-length protein and induces the endosomal recruitment of ESCRT-III proteins. ALIXΔPRR presumably corresponds to an "active" form of the protein, in which the BRO1 domain is no longer blocked by the flexible PRR and is more accessible to interact with the ESCRT-III protein CHMP4 (Zhou et al., 2009, 2010). Likewise, ALIX V-domain interaction with the HIV-1 p6 Gag late domain is also increased in ALIXΔPRR (Strack et al., 2003; Zhou et al., 2008). These observations and other structural analyses have led to a

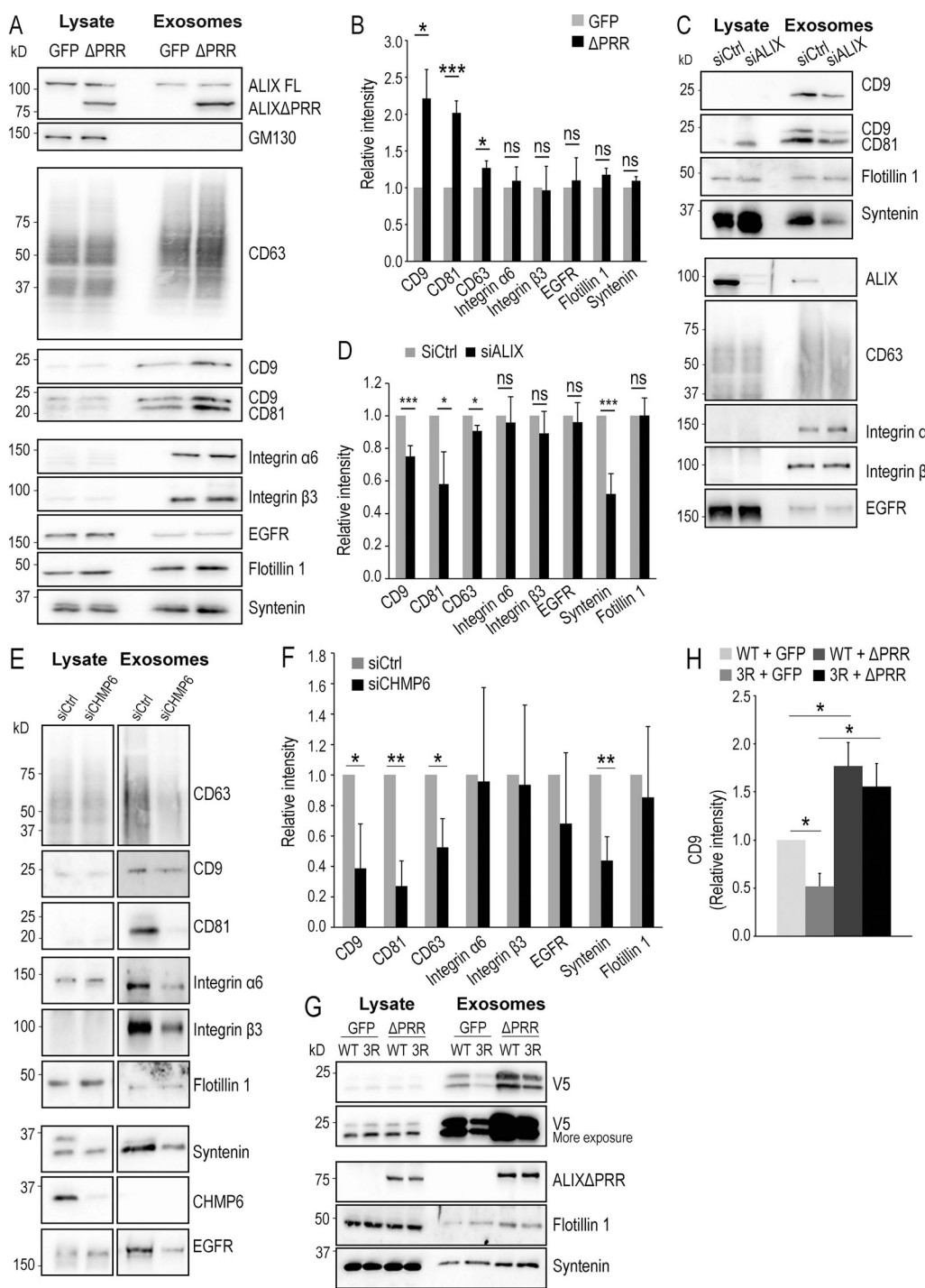

Figure 8. **ALIXΔPRR stimulates the release of exosomes containing tetraspanins. (A and B)** HeLa GFP-CHMP4B cells transfected for 18 h with ALIXΔPRR (ΔPRR) or GFP as a control and incubated with exosome-free medium for 24 h. The cell medium was collected, and exosomes were isolated by differential centrifugation. The cell lysate and the exosome samples were analyzed by Western blotting using antibodies against the indicated proteins. The same amount of protein was loaded for each condition (A). The amounts of protein were quantified by densitometry and are expressed relative to free GFP. In B, boxes, mean; error bars, ±SD ($n$ = 3, from three independent experiments); $t$ test; *, $P < 0.05$, ***, $P < 0.001$; ns, not significant. **(C and D)** HeLa GFP-CHMP4B cells were treated with siRNA against ALIX or nontarget siRNAs (siCtrl) and incubated with exosome-free medium for 24 h. Exosomes were isolated as in A and B, and then cell lysates and exosomes (C) were analyzed by Western blotting as in A and B. The relative amount of protein in exosome fractions was quantified by densitometry (D). Boxes, mean; error bars, ±SD ($n$ = 3, from three independent experiments); $t$ test; *, $P < 0.05$, ***, $P < 0.001$; ns, not significant. **(E and F)** The experiment was as in C and D except that siRNAs against CHMP6 and not against ALIX were used. The upper band in the syntenin blot of siCtrl is CHMP6, because the same membrane was reused for incubation with both antibodies. The relative amounts of protein were then quantified by densitometry (F). Boxes, mean; error bars, ±SD ($n$ = 3, from three independent experiments); $t$ test; *, $P < 0.05$, **, $P < 0.01$. **(G and H)** HeLa GFP-CHMP4B cells were transfected with WT CD9 (WT) tagged with V5 or with the V5-CD9/3R mutant (3R). Total cell lysates and exosomes prepared as in A and B were analyzed by Western blotting using antibodies against V5, ALIX, flotillin-1, and syntenin (G). The V5 blot is shown after short and long exposure, so that WT V5-CD9 and V5-CD9/3R can be better compared. The relative amounts of CD9 were quantified by densitometry (H). Boxes, mean; error bars, ±SD ($n$ = 3, from three independent experiments); $t$ test; *, $P < 0.05$.

**Larios et al.**
ALIX-dependent sorting to exosomes

**Journal of Cell Biology**   12 of 22

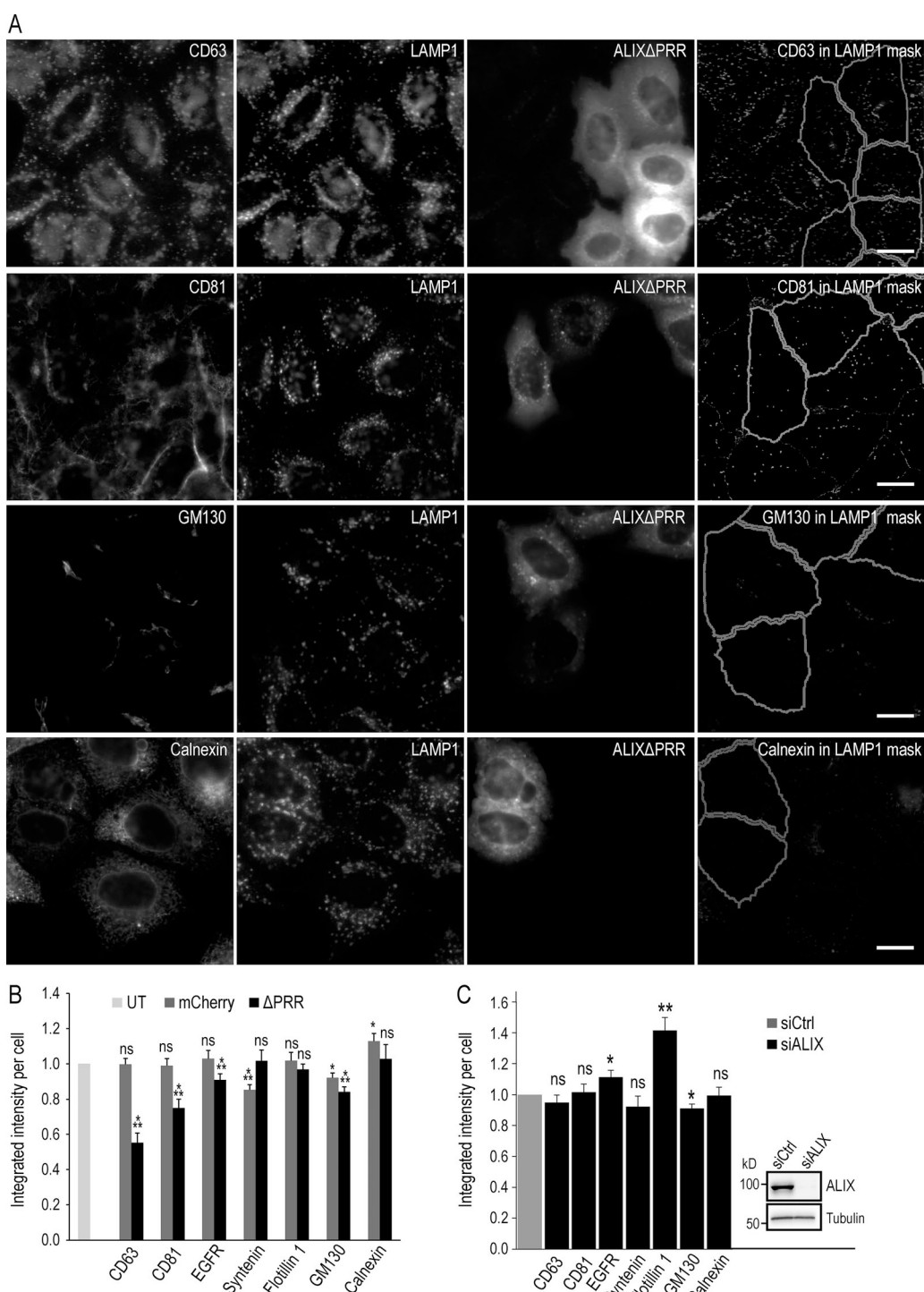

Figure 9. **ALIX controls tetraspanin levels in endosomes and exosomes. (A and B)** HeLa-MZ cells were transfected for 18 h with ALIXΔPRR-mCherry (A) or mCherry (not shown) as control: only 30–40% cells were transfected to facilitate the analysis of transfected versus untransfected cells. After fixation, cells were labeled with antibodies against LAMP1 as well as antibodies against either CD81 or CD63, GM130, and calnexin (A and B), as well as EGFR, syntenin, or flotillin-1 (Fig. S4). Samples were analyzed by automated triple-channel fluorescence microscopy (A). Cells were segmented using the ALIXΔPRR-mCherry or the mCherry signal to identify transfected and untransfected cells. To quantify the distribution of each marker within LAMP1-positive endolysosomes, a mask of the LAMP1 staining pattern was created by segmentation, and this mask was applied onto the staining of the given marker (CD81, CD63, EGFR, syntenin, flotillin-1, calnexin, and GM130), so that the integrated intensity per cell of the marker present in LAMP1-positive endolysosomes could be quantified in the mask and plotted (B). To illustrate the image analysis process, the presence of each marker in the LAMP1 mask only was visualized on the computer screen (reflecting exactly what was being quantified), and the corresponding image of the mask was captured. The righthand panels in A show these screen captures illustrating the staining of each marker within LAMP1-endosomes only (LAMP1 mask). UT, untransfected control cells. Boxes, mean; error bars, ±SD (n = >5,000 cells); $t$ test; *, $P < 0.05$; **, $P < 0.01$; ***, $P < 0.001$; ns, not significant. Scale bar: 10 µm. **(C)** HeLa-MZ cells were treated with siRNA against ALIX or nontarget siRNAs (siCtrl), as in Fig. 8 (C and D) and were processed for automated fluorescence microscopy as in A and B. ALIX KD efficiency was analyzed by Western

blotting (insert in C). The corresponding micrographs are shown in Fig. S5 (A and B). The integrated intensity of each marker per cell was quantified and normalized to the corresponding control showing cells treated with nontarget siRNAs (C). Boxes, mean; error bars, ±SD (n = >5,000 cells); t test; *, P < 0.05, **, P < 0.01; ns, not significant.

model for ALIX "activation," in which the PRR loses its interaction with the BRO1 domain and the V-domain elongates, allowing ALIX dimerization (Pires et al., 2009). ALIXΔPRR, however, inhibits HIV budding (Fisher et al., 2007; Usami et al., 2007), perhaps because its dimerization capacity is no longer regulated, since an ALIXΔPRR dimerization mutant does not inhibit viral budding (Pires et al., 2009). Yet, the same dimerization mutant inhibits ALIX functions at the endosome, clearly suggesting that dimerization is required for membrane association (Bissig et al., 2013; Pires et al., 2009), but also that some factors necessary for HIV release, but not ILV formation, interact with the PRR. Interestingly, ALIXΔPRR also inhibits cytokinesis because the PRR contains CEP55 binding site (Carlton et al., 2008). CEP55 recruits ALIX at the midbody, which in turn recruits CHMP4B and other ESCRTs. Similarly, ALIXΔPRR may support exosome biogenesis, because it contains intact binding sites in the BRO1 domain for both LBPA at the membrane and ESCRT-III.

Thus, one may hypothesize that the BRO1 domain of ALIXΔPRR is also more exposed for interactions with LBPA, or that the global protein conformation allows stronger interactions with the lipid membrane, resulting in an increase in endosomal ALIX. However, beyond efficient recruitment, our FRAP observations show that CHMP4B remains dynamic on ALIXΔPRR-containing endosomes, indicating that individual CHMP4B subunits turn over between endosomal ESCRT-III polymers and cytoplasmic pools. This is consistent with the view that the AAA-ATPase VPS4, which regulates ESCRT-III polymer disassembly and remodeling, plays a crucial role during ESCRT-dependent membrane deformation and fission (Adell et al., 2014, 2017; Mierzwa et al., 2017). Finally, ALIXΔPRR also promotes the endosomal recruitment of VPS4B, further strengthening the notion that ESCRT-III–recruited proteins are functional and could participate in endosomal membrane deformation.

### Mechanism for ALIX-dependent ESCRT-III endosomal recruitment

ALIX has been shown to participate in the recruitment of ESCRT-III proteins during virus budding, cytokinesis, and plasma membrane repair (Carlton and Martin-Serrano, 2007; Jimenez et al., 2014; Morita et al., 2007; Strack et al., 2003) and is presumably involved in an alternative pathway for ESCRT-III nucleation and polymerization on membranes (Christ et al., 2016; Jimenez et al., 2014; McCullough et al., 2008; Meng et al., 2015; Pires et al., 2009). Furthermore, the yeast ALIX homologue, BRO1, induces the activation and membrane polymerization of the ESCRT-III protein SNF7 during cargo sorting into ILVs.

Two parallel pathways were proposed for ESCRT-III endosomal recruitment in yeast: one dependent on ESCRT-I/II (canonical pathway) and a second one dependent on ESCRT-0/ BRO1 (Tang et al., 2016). Our results show that ALIXΔPRR-

dependent CHMP4B recruitment to endosomes is coordinated by the dual capacity of ALIX BRO1 domain to interact with CHMP4 and LBPA; consistently, CHMP4 and LBPA interact with opposite regions of the BRO1 domain (Bissig et al., 2013). These observations agree well with the recent findings that ALIX can nucleate ESCRT-III on endosomal membranes (Skowyra et al., 2018; Radulovic et al., 2018). Moreover, we also find that the well-established CHMP4/SNF7 nucleation factor CHMP6/VPS20 (Saksena et al., 2009; Tang et al., 2015; Teis et al., 2008) is not recruited to endosomes upon ALIXΔPRR expression and is not required for ALIX-dependent CHMP4B membrane association. Yet, much like ALIX, CHMP6 is involved in exosome biogenesis. Altogether, our observations fit nicely with the view that ESCRT-III recruitment and polymerization on endosomal membranes can be mediated alternatively via parallel pathways, dependent on ALIX and CHMP6, respectively. In addition, our data show that other ESCRT-0, -I, -II, or -III proteins (CHMP3 and CHMP6) play no role in ALIXΔPRR-dependent CHMP4B recruitment. Finally, our observations with supported bilayers demonstrate that LBPA- and ALIX-dependent CHMP4B membrane recruitment can be fully recapitulated in vitro using purified components. We conclude that ALIX directly recruits CHMP4 onto LBPA-containing endosomes, thus providing an alternative pathway to the canonical ESCRT-0, -I, and -II-dependent mechanism for ESCRT-III recruitment, as observed during cytokinesis (Christ et al., 2016).

### ALIX- and ESCRT-III–dependent biogenesis of exosomes

The precise functions of the ALIX-dependent pathway into late endosomes remain to be elucidated. Clearly, this alternative pathway may function as a rescue mechanism in late endosomes to retrieve ILV-destined cargoes that may have escaped sorting by canonical ESCRTs earlier in the pathway. However, some further speculations based on our observations are also possible. Our data argue for the discriminating role of the ALIX-dependent pathway in the endosomal sorting of the tetraspanins CD9, CD63, and CD81 (well-established exosomal markers) but not EGFR, a canonical ESCRT-dependent cargo. CD63, in turn, may directly contribute to ILV formation as in melanocytes (Theos et al., 2006; van Niel et al., 2011, 2015) and perhaps other cell types (Edgar et al., 2014). In addition, CD63, CD9, and CD81 may also facilitate cargo sorting via the formation of a molecular web of interactions among tetraspanins and with other partners (Berditchevski and Odintsova, 2007; Perez-Hernandez et al., 2013; van Deventer et al., 2017). Yeast cells, which lack canonical tetraspanins, express Cos proteins with four transmembrane domains, which are involved in cargo sorting and may thus act as functional homologues of mammalian tetraspanins (MacDonald et al., 2015).

We also find that expression of ALIXΔPRR leads to its efficient and selective secretion in exosomes, together with tetraspanins, and that the secretion of this protein subset is accompanied by

decreased endosomal levels. It should be noted that the reported proapoptotic activity of ALIX (Trioulier et al., 2004; Wu et al., 2002) is due to interactions of the PRR of ALIX with the protein ALG-2 (Missotten et al., 1999; Vito et al., 1999), necessary for cell death (Vito et al., 1996), and thus ALIXΔPRR is very unlikely to cause the release of cell fragments and apoptotic bodies.

We speculate that the ALIX-dependent, alternative mechanism for ESCRT-III recruitment on late endosomal membranes controls the sorting of tetraspanins, hence of at least one type of cargoes destined for secretion in exosomes. While there is no doubt that ILVs can have different fates, an outstanding question remains how protein targeting is modulated along these pathways. Previous studies showed that exosomes contain ubiquitinated proteins, but these proteins were mostly of cytosolic origin (Buschow et al., 2005; Moreno-Gonzalo et al., 2018), and the role of ubiquitination in tetraspanin sorting is not known (Guix et al., 2017; Odintsova et al., 2013; Shi et al., 2017; van Niel et al., 2011; Wang et al., 2012). Here, we show that CD9 sorting into exosomes depends on an intact ubiquitination site, strongly suggesting that tetraspanin sorting is regulated by ubiquitination, much like the ESCRT-dependent sorting of other proteins into multivesicular endosomes (Piper et al., 2014). ALIX, however, exhibits both an ubiquitin-dependent (Dowlatshahi et al., 2012; Joshi et al., 2008; Pashkova et al., 2013) and an ubiquitin-independent (Dores et al., 2012, 2016) binding capacity. Since ALIX is cosecreted with tetraspanins in exosomes, it seems reasonable to believe that ALIX interactions with tetraspanins are of a more stable nature than ubiquitin-dependent interactions, which are controlled by deubiquitinating enzymes (Clague et al., 2019). Hence, sorting along the pathways leading to degradation or secretion may ultimately depend on the interactions of ALIX and perhaps other ESCRTs with cargo proteins.

## Materials and methods

### Cells, antibodies, and reagents

We obtained HeLa-MZ cells from Lucas Pelkmans (University of Zurich, Zurich, Switzerland), HeLa Kyoto cells stably expressing GFP-CHMP4B from Anthony Hyman (MPI-CBG, Dresden, Germany; Poser et al., 2008), and HeLa Kyoto cells stably expressing CHMP1A-V5 or CHMP1B-FLAG from Harald Stenmark (The Norwegian Radium Hospital, Oslo, Norway; Christ et al., 2016). Our HeLa cell lines are not on the list of commonly misidentified cell lines maintained by the International Cell Line Authentication Committee; they were authenticated by Microsynth, which revealed 100% identity to the DNA profile of the cell line HeLa (ATCC: CCL-2) and 100% identity over all 15 autosomal short tandem repeats to the Microsynth's reference DNA profile of HeLa. Cells are mycoplasma negative as tested by GATC Biotech (Konstanz, Germany). All cells were grown in MEM (Sigma-Aldrich) supplemented with 10% FBS (Thermo Fisher Scientific), 1% MEM nonessential amino acids (Thermo Fisher Scientific), 2 mM L-glutamine (Thermo Fisher Scientific), 100 µg/ml penicillin, and 100 units/ml streptomycin (Thermo Fisher Scientific) in a 37°C, 5% $CO_2$ incubator. HeLa Kyoto cells stably expressing GFP-CHMP4B were additionally supplemented

with 0.5 mg/ml Geneticin (Millipore), and cells expressing CHMP1A-V5 or CHMP1B-FLAG were maintained with 0.5 µg/ml puromycin (Thermo Fisher Scientific).

The anti-LBPA monoclonal antibody (6C4) has been described (Kobayashi et al., 1998), and the anti-RAB5 monoclonal antibody was a gift from Reinhard Jahn (Max Planck Institute for Biophysical Chemistry, Göttingen, Germany). The antibodies against STAM1 (12434-1-AP), HD-PTP (10472-1-AP), TSG101 (14497-1-AP), CHMP6 (16278-1-AP), VPS4B (17673-1-AP), CHMP1A (15761-1-AP), and CHMP3 (15472-1-AP) were from Proteintech; against FLAG (F3165) and tubulin (T9026) from Sigma-Aldrich; against CHMP4A (H-52) and VPS22 (EAP30 C-11) from Santa Cruz Biotechnology; against CHMP4B (ab105767), BROX (ab193008), flotillin-1 (ab41927), CD9 (ab2215), CD81 (ab79559), CD63 (ab59479), syntenin (ab19903), calnexin (ab13504), LAMP1 (ab25630), integrin α6 (ab181551), and integrin β3 (ab119992) from Abcam; against HRS (GTX101718) from Genetex; against EEA1 (ALX-210-239-C100) and conjugated ubiquitin (FK2) from Enzo Life Sciences; against LAMP1 (D2D11) from Cell Signaling Technology; against GFP (11814460001) from Roche; against V5 (R960-25) from Thermo Fisher Scientific; against EGFR (20-ES04, for Western blot) from Fitzgerald; against EGFR (555996, for immunofluorescence), cathepsin D (610801), and GM130 (610822) from BD Biosciences; and against ALIX (pab0204) from Covalab. EGF (E9644) was from Sigma-Aldrich. The Cy2-, Cy3-, and Cy5-conjugated fluorescent antibodies were from Jackson ImmunoResearch, and the peroxidase-conjugated secondary antibodies from Bio-Rad Laboratories. EGF Alexa Fluor 647 (E35351), Alexa Fluor 488 tetra-fluorophenyl ester, Ni-nitrilotriacetic acid (NTA) agarose, 7-kD molecular weight cutoff (MWCO) Zeba Spin Desalting Columns, and Hoechst 33342 were from Thermo Fisher Scientific. MBPTrap HD 5-ml columns, Glutathione Sepharose 4B, and Dextrin Sepharose High Performance were from GE Healthcare. The cOmplete Protease Inhibitor Cocktail was from Roche. Restriction enzymes were obtained from New England Biolabs.

LBPA ((S,S)bisoleoyl-lysobisphosphatidic acid) was from Echelon Biosciences. DOPC, DOPE, DOPS, liver PI, and lissamine rhodamine B sulfonyl (18:1 Rhod PE) were from Avanti Polar Lipids. Other reagents and chemicals were obtained from Sigma-Aldrich.

### Plasmids, RNA interference, and transfection

RILP plasmids were obtained from Cecilia Bucci (Università del Salento, Lecce, Italy). pmCherry vector was obtained from Clontech. Myc-ALIX plasmid was generated by cloning ALIX cDNA (GenBank: AJ005073.1) into a pCMV-Tag3C vector (Agilent Technologies) using XhoI site. The same procedure was used to generate Myc-ALIXΔPRR (ALIX BRO1 domain and V-domain, which corresponds to the first 702 aa). These plasmids were used to produce Myc-ALIX-mCherry and Myc-ALIXΔPRR-mCherry plasmids. The mCherry cDNA was cloned in ALIX C-terminus using ApaI site. A mutagenesis in ALIX stop codon was generated in the plasmids containing mCherry. The following primers were used: forward, 5′-CTATCCACAGCAGTT ACCTCGAGGGGGGGCCC-3′, and reverse, 5′-GGGCCCCCCCTC GAGGTAACTGCTGTGGATAG-3′. The ALIX BRO1 domain plasmid

for recombinant protein production was generated by cloning the BRO1 domain (1–359 aa) into a pGEX-6P-2 vector. The following primers were used to generate the I212D, QQ (mutation LF 104/105 to QQ) and the siRNA resistant ALIX mutants: I212D forward, 5′-GATAAGATGAAAGATGCCGACATAGCTAAGCTGGCAAATC-3′, and reverse, 5′-GATTTGCCAGCTTAGCTATGTCGGCATCTTTCATCTTATC-3′. LF104/105QQ forward, 5′-GCTTTTGATAAAGGTTCCCAGCAAGGAGGGTCTGTAAAATTGG-3′, and reverse, 5′-CCAATTTTACAGACCCTCCTTGCTGGGAACCTTTATCAAAAGC-3′. siRNA resistant forward, 5′-GCCAAGCCGCTCGTCAAATTCATCCAGCAGACGTAC-3′, and reverse, 5′-GTACGTCTGCTGGATGAATTTGACGAGCGGCTTGGC-3′. The CHMP4B expression plasmid for recombinant protein purification was generated by replacing SNF7 from pMBP-HIS2-SNF7 plasmid (Addgene 21492) for the CHMP4B cDNA, using BamHI and NotI sites. Before cloning CHMP4B into the plasmid, a silent mutation was generated in the CHMP4B sequence to remove an internal BamHI cutting site. The primers used for the mutation were forward, 5′-AACTGGGCTGGGTCCATGTAACCAGCTTTCTTG-3′, and reverse, 5′-CAAGAAAGCTGGTTACATGGACCCAGCCCAGTT-3′. The V5-CD9 plasmid was generated by PCR amplification of human CD9 (Addgene 55013) and cloning in a pDONOR 221 plasmid using the Gateway cloning system from Thermo Fisher Scientific. Then, CD9 was transferred by recombination to a pcDNA6.2/V5-DEST vector. The following primers were used to generate the V5-CD9/3R mutant (mutations K4R, K8R and K11R): forward, 5′-GGCTCCATGCCGGTCAGAGGAGGCACCAGGTGCATCAGATACCTGCTGTTCGG-3′, and reverse, 5′-CCGAACAGCAGGTATCTGATGCACCTGGTGCCTCCTCTGACCGGCATGGAGCC-3′.

The siRNA sequences against ALIX, STAM1, STAM2, TSG101, VPS22, CHMP3, CHMP6, HD-PTP, and BROX were from siTOOLs Biotech. For each gene, a pool of 30 different siRNA sequences was designed. A pool of 30 siRNA sequences that do not interact with human genes was used as a negative control. Each pool was used at a low 3-nM concentration to reduce the danger of off-target effects in KD experiments. The single siRNA sequence against ALIX (5′-AAGCCGCTGGTGAAGTTCATC-3′) was previously characterized, and its effects are fully rescued by RNAi-resistant ALIX (Bissig et al., 2013); the negative control siRNA (AllStars) was from Qiagen. Single siRNAs were used at 20 nM.

DNA and siRNA were transfected in cells according to the manufacturer's instructions using FuGENE HD (Promega Corp.) and Lipofectamine RNAiMAX (Thermo Fisher Scientific), respectively. Unless indicated otherwise, experiments were performed after transfection with DNA for 7 h and siRNA for 72 h.

## Immunofluorescence
Immunofluorescence was performed after fixing cells grown on glass coverslips, or directly in 96-well dish plates for high-throughput microscopy, for 20 min with 3% PFA in PBS. All steps of the immunofluorescence procedure were performed at room temperature. When indicated, cells were permeabilized for 5 min with 0.01% saponin in PBS before PFA fixation. After fixation, cells were incubated for 45 min in 1% fish skin gelatin and 0.1% saponin in PBS, followed by 30-min incubation with the primary antibody in 1% fish gelatin in PBS. After washing the primary antibody with PBS, the cells were incubated for 30 min with the secondary antibody (Cy2-, Cy3-, or Cy5-conjugated fluorescent antibodies) in 1% fish gelatin in PBS, followed by PBS washes. The cells were mounted in Mowiol 40-88 medium containing 10 µg/ml Hoechst and imaged with a Zeiss 700 confocal microscope (Carl Zeiss) using a 63× 1.4-NA oil differential interference contrast (DIC) Plan-Apochromat objective (Nikon) and Zen imaging software (Carl Zeiss). For high-throughput microscopy, 96-well plates were imaged with an ImageXpress Micro XLS Widefield High-content microscope (Molecular Devices) or an ImageXpress Micro Confocal High-content microscope (Molecular Devices; used in the widefield mode) using a 40× 0.95-NA objective (Nikon).

## GUV electroformation
GUVs were prepared by electroformation (Chiaruttini et al., 2015). Briefly, lipids were mixed in chloroform at a final concentration of 1 mg/ml. Then, 20 µl lipids were spread on two indium tin oxide–coated glass slides. After 1-h incubation in a vacuum oven at 30°C, a rubber ring was placed between the two slides, and the space between the glasses was filled with 500 mM sucrose solution (500 mOsmol). GUVs were formed by applying 1 V AC current (10 Hz sinusoidal) for 1 h at 55°C.

## Live imaging of protein binding to supported lipid bilayers
Supported bilayers labeled with trace (0.01-mol) amounts of N-Rhodamine PE were prepared with the following composition: DOPC:DOPE:PI (6.99:2:1 mol), DOPC:DOPE:PI:DOPS (4.99:2:1:2 mol), and DOPC:DOPE:PI:LBPA (4.99:2:1:2 mol) in a glass-bottom flow chamber sticky-slide VI 0.4 from Ibidi. The chamber was filled with 150 µl of 20 mM Hepes, pH 8, 250 mM NaCl (500 mOsmol), and 10 µl of GUVs. After mixing, to obtain a lipid bilayer attached to the glass, the solution was incubated for 10 min followed by three washes with 120 µl ALIX buffer: 25 mM Hepes, pH 7.4, 0.3 mM BAPTA, 0.3 mM NTA, 0.3 mM hydroxyethyl EDTA, and 686 µM $CaCl_2$ (20 µM free calcium, calculated using WEBMAXC Standard software). The chamber was incubated for 10 min with a mix of 4 mg/ml casein in 20 mM Hepes, pH 8.0, and ALIX buffer (volume ratio, 1:1), followed by 15 washes with ALIX buffer. When ALIX BRO1 domain, ALIX BRO1-I212D, or ALIX BRO1-QQ were added, the proteins were incubated for 40 min in ALIX buffer at a final concentration of 3.7 µM, and the unbound protein was removed with ALIX buffer. The chamber was placed in an inverted microscope assembled by Intelligent Imaging Innovation and Nikon (Eclipse C1, Nikon) for imaging. One of the entries of the chamber was connected to a syringe pump. The other entry was used to add CHMP4B-488, which was incubated for 25 min in ALIX buffer at a final concentration of 0.64 µM. The chamber was imaged every minute during the incubation with CHMP4B-488, and a 2-µm-thick volume stack (1 µm above and below the supported membrane) was acquired with a 100× 1.4-NA oil DIC Plan-Apochromat objective (Nikon) and SlideBook 6.0 imaging software (Intelligent Imaging Innovation). The stacks were converted into 2D images by maximum-intensity projection.

## EGF treatment and EGFR degradation analysis: Biochemistry and microscopy assays
HeLa cells stably expressing GFP-CHMP4B were transfected for 18 h with myc-ALIXΔPRR or an empty pCMV-Tag3C vector as a

control and then incubated for 3 h in serum-free medium, followed by 1-h incubation with 10 µg/ml cycloheximide in serum-free medium. Then, cells were treated for 1, 2, or 4 h with 100 ng/ml EGF and 10 µg/ml cycloheximide in serum-free medium. All the treatments were performed in a 37°C, 5% $CO_2$ incubator. Cells were harvested, lysed, and analyzed by SDS-PAGE and Western blot. EGFR degradation measured by immunofluorescence was performed in HeLa-MZ cells transfected for 18 h with myc-ALIXΔPRR-mCherry and then incubated for 3 h in serum-free medium, followed by 1-h incubation with 10 µg/ml cycloheximide in serum-free medium. Cells were then treated for 15, 60, 120, or 240 min with 100 ng/ml EGF and 10 µg/ml cycloheximide in serum-free medium. All the treatments were performed in a 37°C, 5% $CO_2$ incubator.

### Exosome isolation

Exosomes were isolated by differential centrifugation (Théry et al., 2006). Briefly, two square dishes (500 cm² square growth area) of 80% confluence HeLa Kyoto cells stably expressing CHMP4B-GFP were transfected with DNA (Myc-ALIXΔPRR, Myc-ALIXΔPRR-QQ, GFP, V5-CD9, or V5-CD9/3R) for 18 h or siRNA (ALIX or CHMP6) for 48 h. Cells were washed with PBS and incubated for 24 h with an exosome-free medium (Théry et al., 2006). The medium was collected and centrifuged at 300 $g$ for 10 min at 4°C. To remove cell debris, the supernatant was centrifuged two more times at 2,000 $g$ for 20 min and 10,000 $g$ for 30 min at 4°C. Exosomes were sedimented by high-speed centrifugation at 100,000 $g$ for 2 h at 4°C. The pellet was washed with PBS, resedimented by centrifugation at 100,000 $g$ for 2 h at 4°C, and finally resuspended in 120 µl of PBS.

### Subcellular fractionation

Subcellular fractions were prepared by flotation in sucrose gradients (Muriel et al., 2017). Briefly, cells grown in Petri dishes were gently scraped off the dish in ice-cold PBS using a flexible rubber policeman to obtain sheets of attached cells with minimal cell damage. Cells were sedimented at 175 $g$ for 5 min at 4°C, resuspended in HB (homogenization buffer: 250 mM sucrose, 3 mM imidazole, pH 7.4, containing 10 mM leupeptin, 1 mM pepstatin A, and 10 ng/ml aprotinin), and resedimented at 1,355 $g$ for 7 min. Cells were resuspended in 0.5–1 ml HB and homogenized by passage through the beveled tip of a 22-gauge needle using a 1-ml tuberculin syringe. The homogenate was centrifuged at 1,355 $g$ for 7 min. The supernatant was collected, adjusted to 40.6% sucrose, loaded at the bottom of an ultracentrifugation tube, and overlaid with 2.5 ml of 35% sucrose and 3 mM imidazole, pH 7.4, and then with HB. The gradient was centrifuged at 165,000 $g$ for 1 h at 4°C, and the interface between 35% sucrose and HB, containing endosomes and other light membranes, was collected and analyzed by SDS-PAGE.

### Recombinant protein purification

Human CHMP4B was expressed in bacteria and then purified (Mierzwa et al., 2017). Briefly, pMBP-HIS2-CHMP4B was expressed in *Escherichia coli* Rosetta cells. At $OD_{600}$ = 0.7, protein expression was induced with 0.5 mM IPTG for 3 h at 30°C. Bacteria were lysed and sonicated in lysis buffer containing

20 mM Hepes, pH 8, 100 mM NaCl, 1% Triton X-100, and cOmplete Protease Inhibitor Cocktail at 4°C. The fusion protein 6xHis-MBP-CHMP4B was purified by affinity chromatography using an MBPTrap HP 5-ml column. The column was first washed with 20 mM Hepes, pH 8, 250 mM NaCl, and 0.1% Triton X-100, followed by a second wash with 20 mM Hepes, pH 8.0, and proteins bound to the column were eluted in 20 mM Hepes, pH 8 and 10 mM maltose. The 6xHis-MBP region was removed by cleavage with TEV protease, followed by incubation with Ni-NTA agarose resin and dextrin Sepharose medium. CHMP4B fluorescent labeling was performed by incubating the protein with Alexa Fluor 488 tetrafluorophenyl ester in a 1:2 molar ratio (protein:dye) in the presence of 100 mM $NaHCO_3$, pH 9, for 1 h at room temperature. The free dye was removed by overnight dialysis against 20 mM Hepes, pH 8.0, at 4°C, using a membrane of 12–14-kD, followed by size-exclusion chromatography using 7-kD MWCO Zeba Spin Desalting Columns. Finally, the protein was centrifuged at 100,000 $g$ for 10 min at 4°C. The supernatant was aliquoted, snap frozen using liquid $N_2$, and stored at –80°C.

Human ALIX BRO1 domain, ALIX BRO1-I212D, and ALIX BRO1-QQ were expressed and purified from *E. coli* Rosetta cells transformed with the pGEX-ALIX BRO1 domain, pGEX-ALIX BRO1-I212D, or pGEX-ALIX BRO1-QQ. At $OD_{600}$ = 0.5, protein expression was induced with 0.4 mM IPTG for 18 h at 18°C. Bacteria were lysed and sonicated in lysis buffer containing PBS, 1 mM EDTA, 1 mM DTT, 1% Triton X-100, and cOmplete Protease Inhibitor Cocktail at 4°C. The lysate was centrifuged at 10,000 $g$ for 30 min at 4°C, and the supernatant was filtered through a 0.22-µm filter. The lysate was incubated with Glutathione Sepharose beads for 2 h at 4°C. The beads were first washed with PBS, followed by a second wash with 50 mM Tris, pH 7.5, 150 mM NaCl, 1 mM EDTA, and 1 mM DTT. ALIX BRO1 domain was release from the Glutathione Sepharose beads by the cleavage of GST-ALIX BRO1 domain using PreScission protease. The supernatant was loaded into a size-exclusion chromatography Zeba Spin Desalting Column (7-kD MWCO) for buffer exchange. The final buffer was 25 mM Hepes, pH 7.4, 0.3 mM BAPTA, 0.3 mM NTA, and 0.3 mM hydroxyethyl EDTA. After centrifugation at 100,000 $g$ for 10 min at 4°C, the supernatant was aliquoted, snap frozen using liquid $N_2$, and stored at –80°C.

### FRAP

HeLa GFP-CHMP4B cells were transfected with ALIXΔPRR-mCherry for 8 h, treated with 10 µM nocodazole for 2 h (to depolymerize microtubules and avoid endosome movement), and rinsed using FluoroBrite DMEM Medium (Thermo Fisher Scientific) before imaging. Endosomes were imaged using a 100× 1.4-NA oil DIC Plan-Apochromat VC objective (Nikon) with a Nikon A1 scanning confocal microscope and Nis-elements software (Nikon). Photobleaching was performed in circular regions with four iterations of 488 nm at 100% laser intensity.

### Lysosomal pH analysis with Lysosensor

To analyze the pH of endosomes containing ALIXΔPRR, HeLa MZ cells transfected with ALIXΔPRR-mCherry for 24 h were incubated for 30 min with EGF-Alexa Fluor 647 at 37°C and then chased for 90 min in marker-free medium to label late endocytic

compartments. Acidic endolysosomal compartments were then labeled using 1 µM Lysosensor in Leibovitz L-15 medium (Thermo Fisher Scientific) for 5 min at room temperature. Cells were then imaged using a 100× 1.4-NA oil DIC Plan-Apochromat VC objective (Nikon) with a Nikon A1 scanning confocal microscope and Nis-elements software.

### Image analysis

CHMP4B clusters were quantified in ALIXΔPRR-mCherry–expressing cells using CellProfiler (Kamentsky et al., 2011). Briefly, the cell periphery was selected manually, and the images were processed with the enhanced edges module. The object identification module was used to identify and quantify the CHMP4B clusters per cell. Coloc 2 plugin from ImageJ (National Institutes of Health) was used for colocalization analysis.

High-throughput automated image analysis was performed using custom module editor MetaXpress software, from Molecular Devices. Briefly, cells were segmented and the nucleus, cytoplasm, and the granules corresponding to LAMP1 were identified, as well as the granules corresponding to each marker (endosomal EGFR, CD9, CD81, CD63, syntenin, flotillin-1, GM130, and calnexin). In cells expressing ALIXΔPRR-mCherry or mCherry, the intensity of the granules per cell was quantified and compared with nonexpressing cell. In cells depleted of ALIX with siRNAs, the intensity of the granules per cell was quantified and compared with mock-treated control cells. For quantification, a mask of the LAMP1 staining pattern was created by segmentation, and this LAMP1 mask was applied onto the staining of the given marker (CD81, CD63, EGFR, syntenin, flotillin-1, calnexin, and GM130), so that the integrated intensity per cell of the marker present in LAMP1-positive endolysosomes could be quantified in the mask. To illustrate this image analysis process, the presence of each marker in the LAMP1 mask only was visualized on the computer screen (reflecting exactly what is being quantified), and the corresponding image of the mask was captured. It should be noted that such screen captures are not original image files, since image processing does not allow the export of each image modified during the process.

For FRAP analysis, background fluorescence was subtracted from region of interest intensity values using ImageJ. Those values were subsequently normalized by a nonbleached area, and the value of intensity of the first frame after bleaching was subtracted. Finally, FRAP curves were normalized with the prebleach fluorescence intensity value.

ImageJ was used to quantify and compare the pH of endolysosomes with or without ALIXΔPRR, and Lysosensor mean fluorescence intensity was measured in endolysosomes that contained EGF (using the EGF mask) and ALIXΔPRR (ΔPRR endosomes) or not (non-ΔPRR endosomes). This software was also used to quantify CHMP4B-488 intensity in the in vitro supported lipid bilayer experiments.

### Other experimental procedures

Cell lysis was performed with 50 mM Tris, pH 7.4, 1% NP-40, 0.25% sodium deoxycholate, 150 mM NaCl, 1 mM EDTA, 1 mM PMSF, 1 µg/ml aprotinin, 1 µg/ml leupeptin, and 1 µg/ml pepstatin. Western blot analysis was preformed using WesternBright ECL from Advansta.

### Statistical analysis

For the comparison of the mean values of two groups, statistical significance was determined by a two-tailed $t$ test for unpaired samples. Before $t$ test, normal distribution and equal variance were determined by Shapiro–Wilk test and Levene's mean test, respectively. The P value to be rejected was 0.05. For nonnormal distribution, a nonparametric test (Mann–Whitney $U$ test) was used to determined statistical significance. SigmaPlot 11.0 (Systat Software) was used for all the statistical analysis.

### Online supplemental material

Fig. S1 shows that the stable expression of GFP-CHMP4B in HeLa Kyoto cells is low, similar to endogenous levels. Fig. S1 also shows that CHMP4B is recruited onto membrane upon expression of full-length ALIX and ALIXΔPRR. Fig. S2 shows SDS gels that illustrate the steps in the purification of recombinant proteins (CHMP4B, ALIX-BRO1 domain, ALIX BRO1-I212D, and ALIX BRO1-QQ) used in the in vitro assay shown in Fig. 4. Fig. S3 shows Western blots of cell lysates and exosomal fractions, which illustrate the distribution of CD63, ALIXΔPRR, and CHMP4B, compared with the controls calnexin and GFP. Fig. S3 also shows the distribution of CD63, CD9, CD81, integrin β3, integrin α6, flotillin-1, syntenin, and EGFR in cell lysates and exosomal fractions prepared from cells expressing ALIXΔPRR or ALIXΔPRR-QQ. Fig. S4 shows representative micrographs of EGFR, syntenin, and flotillin-1 by automated fluorescence microscopy in cells expressing ALIXΔPRR. Data are processed as in Fig. 9 A and quantified in Fig. 9 B. Fig. S5 shows representative micrographs of CD63, CD81, EGFR, syntenin, flotillin-1, calnexin, and GM130 by automated fluorescence microscopy after ALIX depletion. Data are quantified in Fig. 9 C.

## Acknowledgments

The authors thank Dimitri Moreau from the high content and high throughput screening facility ACCESS Geneva and Christoph Bauer from the Geneva Bioimaging Center for constant support and help with imaging. We thank Guillaume van Niel for sharing insights about tetraspanin ubiquitination. We are grateful to members of the Gruenberg and Roux laboratories for fruitful discussion. We also thank Marie-Claire Velluz, Marie Hélène Beuchat, Brigitte Bernadets, and Frédéric Humbert for technical support.

Support to J. Gruenberg was from the Swiss National Science Foundation, the Swiss Sinergia program, the Polish-Swiss Research Program (PSPB-094/2010), the NCCR in Chemical Biology, and LipidX from the Swiss SystemsX.ch initiative, evaluated by the Swiss National Science Foundation. A. Roux acknowledges funding from Human Frontier Science Program Young Investigator Grant RGY0076/2009-C; the Swiss National Fund for Research grants 31003A_130520, 31003A_149975, and 31003A_173087; and the European Research Council Consolidator Grant 311536.

The authors declare no competing financial interests.

Author contributions: J. Larios carried out all experiments and analyses. V. Mercier performed some experiments with J. Larios, in particular the FRAP, the endosomal pH sensor, and the high-throughput imaging experiment in which ALIXΔPRR was expressed. J. Larios, J. Gruenberg, and A. Roux wrote the paper, with corrections from V. Mercier.

Submitted: 17 April 2019

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

# Supplemental material

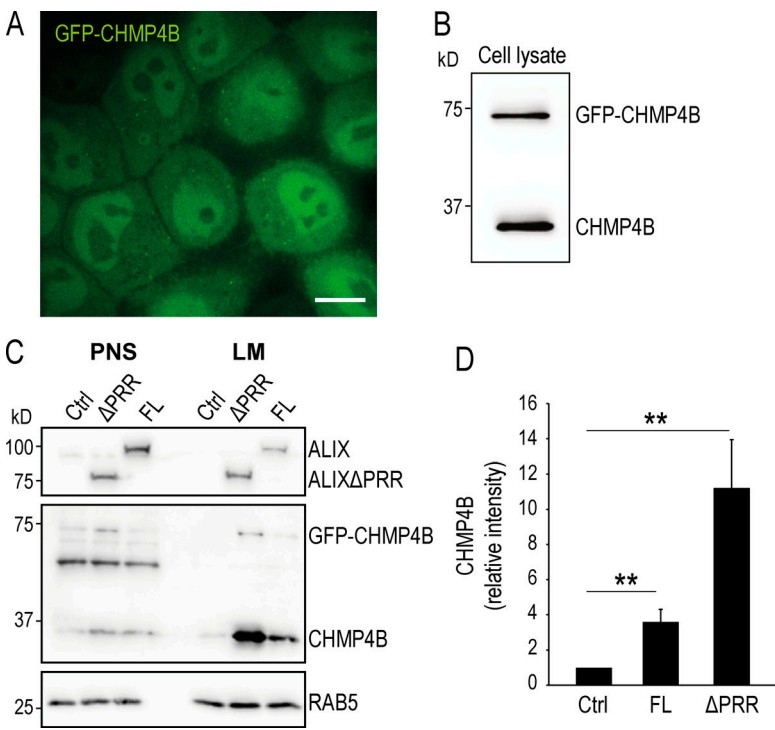

Figure S1.  **Stable cell line expressing GFP-CHMP4B and analysis of full-length ALIX versus ALIXΔPRR. (A and B)** HeLa Kyoto cells stably expressing GFP-CHMP4B (HeLa GFP-CHMP4B cells) were imaged (A) by confocal microscopy and analyzed by Western blotting (B) using anti-CHMP4B antibody. Scale bar in A: 10 µm. **(C and D)** HeLa GFP-CHMP4B cells were transfected for 18 h with full-length ALIX (FL) or ALIXΔPRR (ΔPRR). The postnuclear supernatant (PNS) and the light membranes (LM) were analyzed by Western blotting using antibodies against CHMP4B and ALIX (C), as well as RAB5 (equal loading control). The relative amounts of CHMP4B in LM fractions was quantified by densitometry (D), using RAB5 intensity for the normalization of CHMP4B signal. Boxes, mean; error bars, ±SD (n = 3, from three independent experiments); **, P < 0.01.

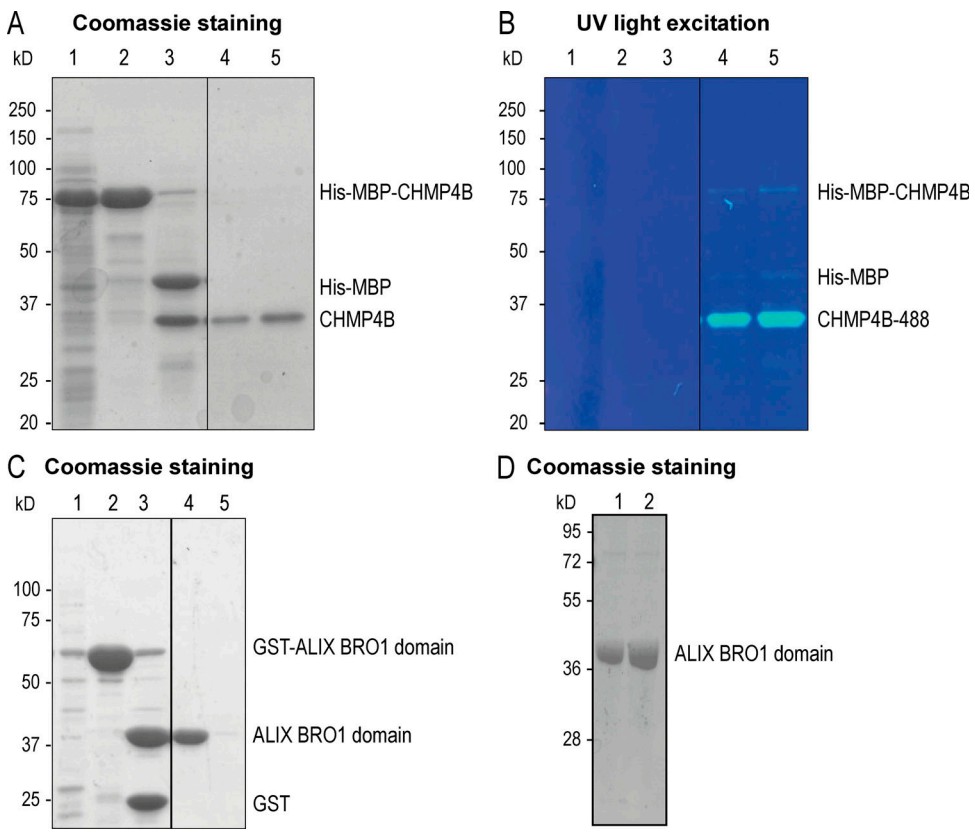

Figure S2. **Recombinant CHMP4B and ALIX BRO1 domain protein purification. (A and B)** A typical SDS-PAGE gel of recombinant human CHMP4B purification was stained with Coomassie brilliant blue (A). The gel compares aliquots taken at sequential steps of the purification: (1) Bacterial lysate expressing His-MBP-CHMP4B. (2) Protein elution from an MBPTrap column showing the His-MBP-CHMP4B purified protein. (3) Protein fragments obtained after incubation with TEV protease (His-MBP and CHMP4B). During the final step of the protein purification procedure, the protein was centrifuged to remove protein aggregates. Aliquots of the supernatant (4) and resuspended pellet (5) are shown. The same gel was exposed to UV light at 320-nm wavelength (B). Fluorescently labeled CHMP4B-488 is visible in wells 4 and 5. **(C)** A typical SDS-PAGE gel of the purification of human ALIX BRO1 domain was stained with Coomassie brilliant blue. The gel compares aliquots taken at sequential steps of the purification: (1) Bacterial lysate expressing GST-ALIX BRO1 domain. (2) GST-ALIX BRO1 domain purified with Glutathione Sepharose beads. (3) Protein fragments obtained after the incubation with PreScission protease (ALIX BRO1 domain and GST). During the final step of the protein purification procedure, the protein was centrifuged to remove protein aggregates. An aliquot of the supernatant (4) and the resuspended pellet (5) are shown. **(D)** A typical SDS-PAGE gel after purification of the human ALIX BRO1 I212D (lane 1) and QQ (lane 2) mutants, stained with Coomassie brilliant blue.

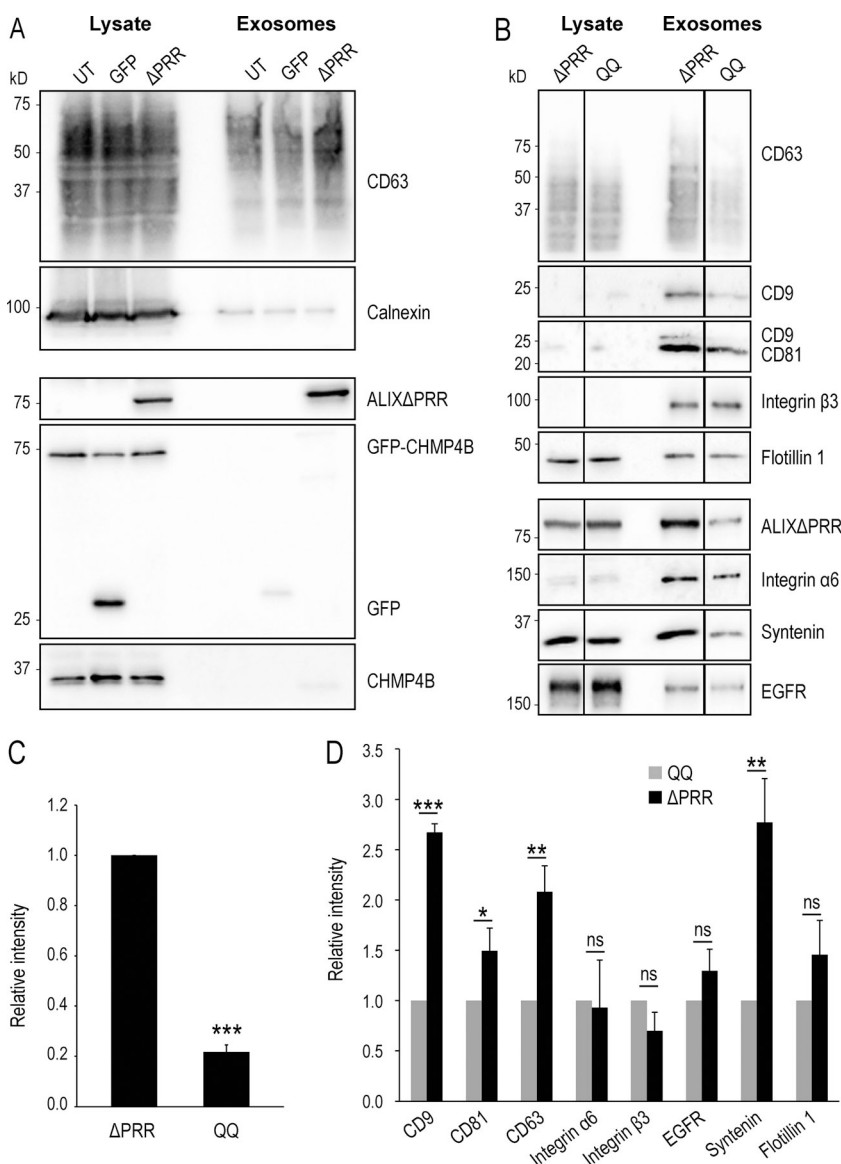

Figure S3. **The QQ mutant of ALIXΔPRR does not stimulate the release of exosomes containing tetraspanins. (A)** HeLa GFP-CHMP4B cells transfected for 18 h with ALIXΔPRR (ΔPRR) or GFP as a control and incubated with exosome-free medium for 24 h. The cell medium was collected, and exosomes were isolated by differential centrifugation. The cell lysates and the exosome fractions were analyzed by Western blotting using antibodies against CD63, calnexin ALIX, CHMP4, and GFP. Calnexin was not found in the exosome fractions prepared from cells expressing ALIXΔPRR; neither were GFP-CHMP4B and GFP. The same amount of protein was loaded for each condition. UT, untransfected cells. **(B–D)** HeLa GFP-CHMP4B cells were transfected for 18 h with ALIXΔPRR (ΔPRR) or with the QQ mutant of ALIXΔPRR (QQ). After processing as in A, the cell lysates and exosome fractions were analyzed by Western blotting using antibodies against the indicated proteins (B). The relative amount of protein in exosome fractions was quantified by densitometry (C and D). C shows the intensity of the QQ mutant relative to ALIXΔPRR, and D shows the relative intensities of the indicated markers in the ALIXΔPRR fractions relative to the fractions prepared from cells expressing the QQ mutant. Boxes, mean; error bars, ±SD ($n$ = 3, from three independent experiments); $t$ test; *, P < 0.05, **, P < 0.01, ***, P < 0.001; ns, not significant.

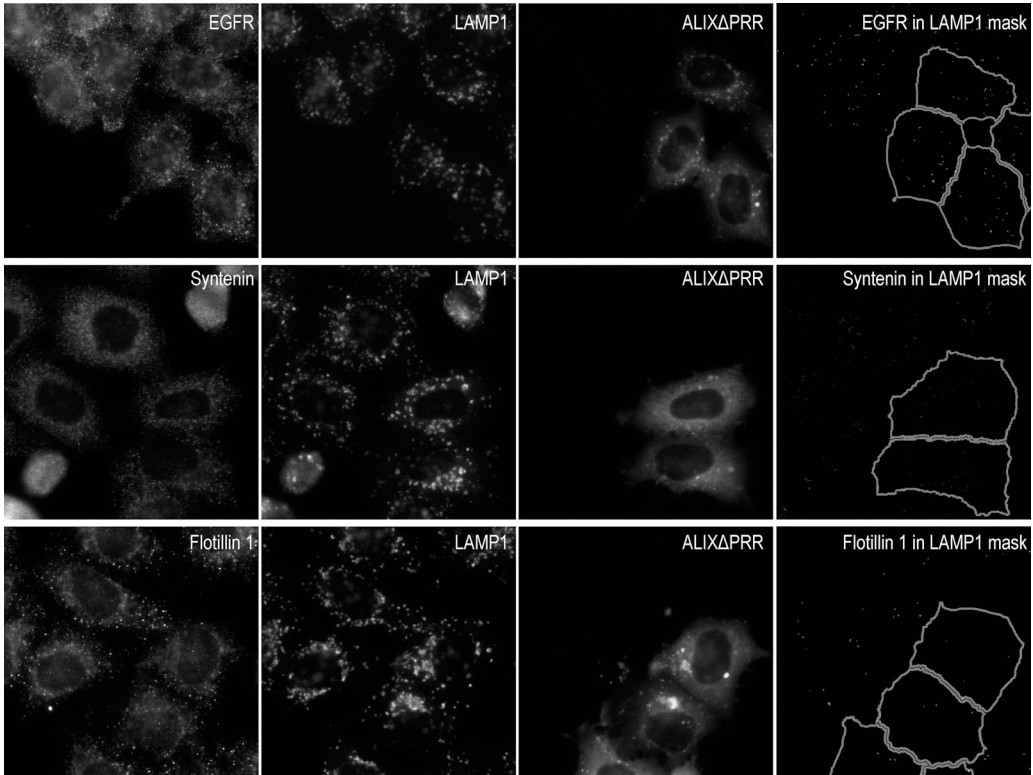

Figure S4.  **EGFR, syntenin, and flotillin-1 in endosomes of cells expressing ALIXΔPRR.** HeLa-MZ cells were transfected for 18 h with ALIXΔPRR-mCherry (A) or mCherry (not shown) as a control; only 30–40% of the cells were transfected to facilitate the compared analysis of transfected versus untransfected cells. After fixation, cells were labeled with antibodies against LAMP1 as well as antibodies against EGFR, syntenin, or flotillin-1 (panels showing CD81 or CD63, GM130, and calnexin are shown in Fig. 9). Samples were then processed for high-throughput automated triple-channel fluorescence microscopy. Cells were segmented using the ALIXΔPRR-mCherry or the mCherry signal to identify transfected and untransfected cells. To quantify the distribution of each marker within LAMP1-positive endolysosomes, a mask of the LAMP1 staining pattern was created by segmentation, and this mask was applied onto the staining of the given marker (EGFR, syntenin, flotillin-1) so that the integrated intensity per cell of the marker present in LAMP1-positive endolysosomes could be quantified in the mask (Fig. 9 B). To illustrate the image analysis process, the presence of each marker in the LAMP1 mask only was visualized on the computer screen (reflecting exactly what was being quantified), and the corresponding image of the mask was captured. The righthand panels show these screen captures illustrating the staining of each marker within LAMP1-endosomes only (LAMP1 mask). Scale bar: 10 µm.

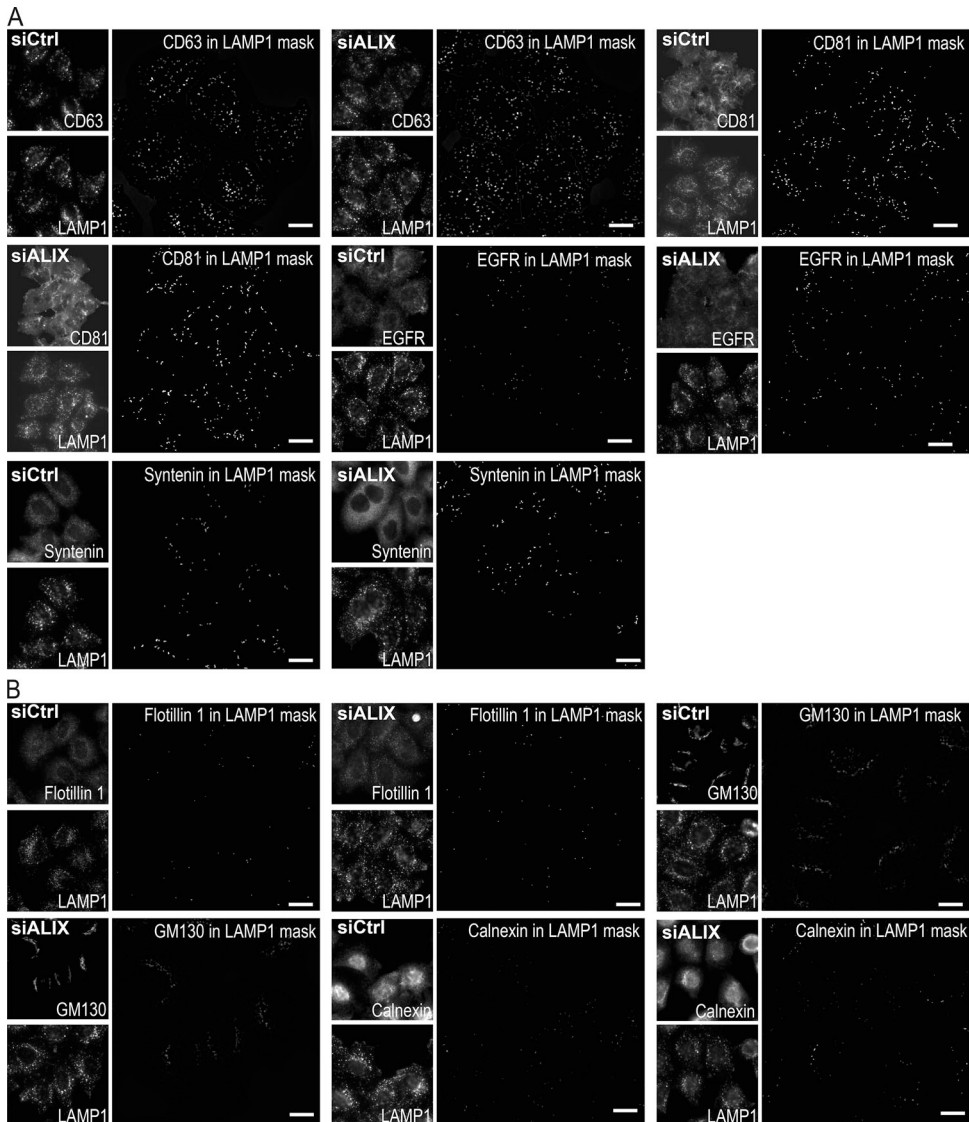

Figure S5. **ALIX depletion does not affect the endosomal distribution of the various markers. (A and B)** HeLa-MZ cells were treated with siRNA against ALIX or nontarget siRNAs (siCtrl) and processed for automated fluorescence microscopy using the same antibodies as in Fig. 9, A and B; and Fig. S4 to detect CD63, CD81, EGFR, and syntenin (A), as well as flotillin-1, GM130, and calnexin (B). In cells depleted of ALIX with siRNAs, the intensity of the granules per cell was quantified and compared with mock-treated control cells. To quantify the distribution of each marker within LAMP1-positive endolysosomes, a mask of the LAMP1 staining pattern was created by segmentation, and this mask was applied onto the staining of the given marker (A: CD63, CD81, EGFR, and syntenin; B: flotillin-1, GM130, and calnexin) so that the integrated intensity per cell of the marker present in LAMP1-positive endolysosomes could be quantified (Fig. 9 C). To illustrate the image analysis process, the presence of each marker in the LAMP1 mask only was visualized on the computer screen (reflecting exactly what was being quantified), and the corresponding image of the mask was captured. These screen captures illustrate the staining of each marker within LAMP1-endosomes only (LAMP1 mask). Scale bar: 10 µm.

