## [Peer Review File · The Journal of Cell Biology]

ALIX- and ESCRT-III-dependent sorting of tetraspanins to exosomes

Jorge Larios, Vincent Mercier, Aurelien Roux, and Jean Gruenberg

Corresponding Author(s): Jean Gruenberg, University of Geneva and Aurelien Roux, University of Geneva

Review Timeline:

Submission Date:	2019-04-17
Editorial Decision:	2019-05-15
Revision Received:	2019-10-31
Editorial Decision:	2019-11-27
Revision Received:	2019-12-06

Monitoring Editor: Marino Zerial

Scientific Editor: Tim Spencer

Transaction Report:

DOI: <https://doi.org/10.1083/jcb.201904113>

May 15, 2019

Re: JCB manuscript #201904113

Prof. Jean Gruenberg
University of Geneva
Department of Biochemistry
30 quai Ernest Ansermet
Geneva 4 1211
Switzerland

Dear Jean and Aurélien,

Thank you for submitting your manuscript entitled "ALIX- and ESCRT-III-dependent sorting of tetraspanins to exosomes". The manuscript was assessed by expert reviewers, whose comments are appended to this letter. We invite you to submit a revision if you can address the reviewers' key concerns, as outlined here.

You will see that all three reviewers agree that the underlying premise of the study is of interesting but they also concur that more work is necessary to support the main conclusions. In general, they request some control experiments and edits to take into consideration other concepts in the literature. It seems to us that the requested experiments are doable. We hope that you will be able to address these reviewer comments in a revised manuscript.

You will note that reviewer #1 feels that substantially more evidence is needed to adequately assess the role of this pathway in tetraspanin sorting and accumulation in exosomes (her/his point #1). While we agree with the reviewer that this is an interesting issue, we do not feel that it need be experimentally addressed in the revision.

GENERAL GUIDELINES:

Text limits: Character count for an Article is < 40,000, not including spaces. Count includes title page, abstract, introduction, results, discussion, acknowledgments, and figure legends. Count does not include materials and methods, references, tables, or supplemental legends.

Figures: Articles may have up to 10 main text figures. Figures must be prepared according to the policies outlined in our Instructions to Authors, under Data Presentation, <http://jcb.rupress.org/site/misc/ifora.xhtml>. All figures in accepted manuscripts will be screened prior to publication.

***IMPORTANT: It is JCB policy that if requested, original data images must be made available. Failure to provide original images upon request will result in unavoidable delays in publication. Please ensure that you have access to all original microscopy and blot data images before

submitting your revision.***

Supplemental information: There are strict limits on the allowable amount of supplemental data. Articles may have up to 5 supplemental figures. Up to 10 supplemental videos or flash animations are allowed. A summary of all supplemental material should appear at the end of the Materials and methods section.

The typical timeframe for revisions is three months; if submitted within this timeframe, novelty will not be reassessed at the final decision. Please note that papers are generally considered through only one revision cycle, so any revised manuscript will likely be either accepted or rejected.

Thank you for this interesting contribution to Journal of Cell Biology. You can contact us at the journal office with any questions, cellbio@rockefeller.edu or call (212) 327-8588.

Sincerely,

Marino Zerial, PhD
Monitoring Editor
JCB

Tim Spencer, PhD
Deputy Editor
Journal of Cell Biology

Reviewer #1 (Comments to the Authors (Required)):

In this manuscript, Larios et al document that the Bro domain-containing protein, ALIX, recruits ESCRT-III subunits to endosomes and facilitates the selective recruitment of tetraspanins to internal vesicles of multivesicular endosomes. ALIX is well known to interact with ESCRT-I and ESCRT-III subunits, to regulate intraluminal vesicle (ILV) formation by the classical ESCRT pathway in yeast, and to facilitate the budding of certain retroviruses by recruiting their Gag proteins, but its role in unconventional ILV formation is not understood. Because ALIX appears to be largely autoinhibited by an interaction of its proline-rich domain (PRR) with the Bro domain, the authors here employ a mutant ALIX that lacks the PRR to overexpress fully active ALIX. They show convincingly that this results in the recruitment of the CHMP4 family ESCRT-III subunits - and, to a lesser extent, a few other ESCRT-III components - to late endosomes both in vivo and, for CHMP4, in vitro. Mutations that disrupt the Bro domain or its interaction with the late endosome/lysosome-restricted lipid, LBPA, block this recruitment, suggesting a pathway from LBPA to ALIX to ESCRT-III. A consequence of this recruitment is the accumulation of polyubiquitin on the endosomes and a partial depletion of tetraspanins CD63, CD9 and CD81 from late endosomes, together with enhanced tetraspanin incorporation into exosomes. The authors conclude that ALIX functions in a

non-canonical ESCRT-III-dependent pathway to sort tetraspanins and other cargoes into ILVs for ultimate secretion as exosomes.

In dissecting the role of ALIX in ILV formation, the paper addresses a question that is of interest to the field and that is still incompletely answered. The data in the first four figures are pristine and clearly document a direct role for ALIX in recruiting several ESCRT-III subunits (and curiously, the ESCRT-II subunit VPS22, which is not really discussed) to endosomes independently of other ESCRT subunits. The *in vitro* data in Figure 3 using purified components are particularly convincing. The data in the last three figures are suggestive of an important role in protein sorting. Thus, in principle this paper would be of interest to the readers of JCB.

However, in its current form the paper has several deficits, as detailed in the comments described below. There are two major concerns. First, at the "why does this matter" level, the authors fail to describe the solid physiological context within which an alternative pathway for ILV formation might function. Such a context exists and is begging for an explanation, but is neither discussed nor experimentally addressed in this paper (points # 1, 2 and 5). This should be remedied. Second, the data supporting a role for this pathway in sorting ubiquitylated cargo (point #3) and tetraspanins for exosome incorporation (point #4) are weak and require additional controls, and the involvement of CHMP4B and LBPA binding in tetraspanin sorting needs to be confirmed (point #4). Several additional minor concerns also should be addressed (points #6-10). If the manuscript is appropriately modified to address these weaknesses, it should make a solid contribution to the field.

Specific concerns:

1. In the Introduction citing alternative fates of ILVs, the authors ignore additional fates of ILVs other than degradation or exosomal release. For example, in antigen-presenting cells MHC class II molecules on internal vesicles are loaded with peptides for presentation at the plasma membrane (Peters et al., 1991, *Nature* 349: 669; Kleijmeer et al., 2001, *J. Cell Biol.* 155: 53; Zwart et al, 2005, *Immunity* 22: 221), and internal vesicles of multivesicular endosomes in melanocytes seed the formation of PMEL-containing amyloid fibrils upon which melanins deposit during melanosome maturation (Berson et al., 2001, *Mol. Biol. Cell* 12: 3451; Hurbain et al., 2008 *Proc. Natl. Acad. Sci. USA* 105: 19726). These are clear examples whereby alternative forms of ILVs are preserved from degradation and should be cited.

2. The authors also fail to cite evidence supporting ESCRT- and VPS4-independent mechanisms for ILV formation that have been well-documented and assessed mechanistically (Theos et al., 2006, *Dev. Cell* 10: 343; Trajkovic et al., 2008, *Science* 319: 1244-1247; van Niel et al., 2011, *Dev. Cell* 21: 708; van Niel et al., 2013, *Cell Rep.* 13: 43), and evidence that CD63 is present on a distinct population of ILVs from ESCRT-dependent cargoes (Edgar et al., 2014, *Traffic* 15: 197). These are directly relevant to the proposed mechanism described in this paper. Moreover, the involvement of Apolipoprotein E in the formation of PMEL- and CD63-containing ILVs (Van Niel et al., 2013) already provided a potential mechanism for formation of that class of ILVs. The authors need to present and clearly distinguish this class of ILV formation from the well-characterized ESCRT-dependent process; in the document as it stands, only the latter is considered. For example, paragraph 2 of the Introduction needs to be qualified for the types of proteins that are incorporated into ILVs or specified for ESCRT-0/I-dependency. The role of the proposed pathway in these alternative fates should also be addressed experimentally (see point # 5).

3. The data in Figure 5A-C and Figure S5B are convincing of some accumulation of ubiquitylated proteins in late endosomes of cells overexpressing ALIXdeltaPRR, but the conclusion that these

proteins are cargoes (on line 242) is not supported - it is not clear if the polyubiquitin is conjugated to cargo, released free in the endosome, or associated with ubiquitylated effector proteins on the cytosolic face. Identification of some of the polyubiquitylated proteins (such as by purification and mass spectrometry analyses of polyubiquitylated proteins from ALIXdeltaPRR-expressing cells relative to ALIXdeltaPRR-I212D-expressing cells) would be helpful in addressing this question. Are specific polyubiquitylated bands detected by SDS-PAGE of overexpressing cells relative to controls? Note, the authors may wish to discuss their results in light of evidence that tetraspanins in yeast are sorted to MVBs via the classical ESCRT-dependent pathway following ubiquitylation (MacDonald et al., 2015, Dev. Cell 33: 328-342).

4. The quantification of the experiments in Figures 6 and 7 is interesting but not terribly dramatic, and the effects shown could potentially be due to indirect effects of ALIX depletion or overexpression. They are also not completely consistent. For example, while ALIX overexpression has no consistent effect on syntenin accumulation in exosomes, ALIX depletion does. These experiments would be much more convincing with additional controls. First, quantification of ALL markers - including EGFR, flotillin, and integrin components, as well as a solid negative control like calnexin - in both exosome fractions (relative to cell lysates by blotting) and LAMP1+ endosomes (by immunofluorescence microscopy) should be done in both ALIXdeltaPRR overexpressers and ALIX KD cells; if the results are interpreted correctly, then opposite results should be seen for most/all markers in the knockdowns and the overexpressers (and exceptions such as syntenin distinguished as either outliers or the rule). Second, a more robust control for ALIX specificity should be included. A good one to use for the overexpression would be the QQ mutant described in Figure 3 - if ALIX functions by the mechanism described in previous figures, then overexpression of the QQ mutant should not impact tetraspanin accumulation in exosomes. Similarly, an ESCRT-I or -II component kd should impact incorporation into exosomes of EGFR but not tetraspanins. Third, it would be helpful to link these observations to the recruitment of CHMP4B or other ESCRT-III components by assessing the impact of CHMP4B kd or VPS4 dominant mutant overexpression. These experiments are needed to test whether the observations on exosomes and cargo recruitment to endosomal ILVs are linked to the effector recruitment phenotypes observed in the previous figures.

5. Given that classical ESCRT-independent pathways for ILV formation have been previously described, it would be of great benefit to test if the ALIX-CHMP4 pathway described here plays a role in these pathways. For example, is the sorting of exogenous PMEL (Theos et al., 2006 and van Niel et al., 2011) or PLP (Trajkovic et al., 2008) to ILVs blocked by ALIX depletion? These could easily be tested by expressing these proteins by transfection in the HeLa cells described here, and would go a long way in providing a solid physiological context for the described pathway.

Minor concerns:

6. In the text description of Figure 2, VPS22 is mistakenly referred to as an accessory ESCRT-III component; it is an ESCRT-II subunit. Rather, CHMP1A and CHMP1B are more considered accessory ESCRT-III components and homologs of the yeast Did2 protein.

7. In the text describing Figure 3A,B, it is misleading in the text to interpret this as a failure of ALIXdeltaPRR-I212D to recruit GFP-CHMP4B to endosomes, since it is not itself localized to endosomes. Taken at face value, these data support the conclusion that CHMP4B recruits ALIX to endosomes, which then recruits more CHMP4B, and directly contradicts the statement on line 187: "Since ALIX binds late endosomes via LBPA and then recruits ESCRT-III in vivo" - the "then" is misleading in the context of these results.

8. In Figure 6A, it is not clear how the authors come up with their number of a 20X increased yield of ALIX Δ PRR in the exosome fraction relative to endogenous ALIX; the image shown certainly does not support this (maybe at most a 2-3X increase in the exosome fraction relative to the amount in the cell lysate). If the number is correct, then a more representative image should be shown. Similarly, in Figure 6B it is not clear that the recovery of tetraspanins and other proteins was normalized to the level in the cell lysates - was it?

9. The authors' earlier papers showed that LBPA was limited to the interior of late endosomes/lysosomes. How is it available for binding to ALIX on the cytoplasmic face? Or does ALIX bind to some other ligand on the cytoplasmic face that is induced by LBPA accumulation on the interior? This should be addressed in the text of the introduction or discussion.

10. Figures S3 and S6 need a scale bar.

Reviewer #2 (Comments to the Authors (Required)):

The Gruenberg and Roux labs present experimental evidence that suggests that ALIX and ESCRT-III somehow sequester and package a specific subset of cargo molecules (tetraspanins) for exosomal secretion. As such the results provide an interesting starting point to begin to better understand on a mechanistic level (finally!), how cargo molecules are selectively packaged into exosomes. A few control experiments should be added to strengthen the conclusions:

Major points:

1. In the majority of experiments expression of ALIX Δ PRR is used to recruit ESCRT-III to late endosomes. In these experiments, overexpression of wt-ALIX appears to be the appropriate control (e.g. Fig.1e-j and others, particular exosome experiments in Fig. 6/7).
2. The finding that interaction of ALIX Δ PRR with ESCRT-III leads to a mutual reinforcement of endosomal localization is interesting. Do the two proteins already interact in the cytosol and 'land' together on LBPA containing endosomes as preformed complexes?
3. The in vitro recruitment assays should be performed by comparing the capability of ALIX Δ PRR / ALIX Δ 212D to recruit Chmp4b.
4. Fig.5a: The localization of ubiquitin in cells expressing ALIX or ALIX Δ PRR should be shown.
5. The differences for the ubiquitinated proteins in Fig. S5A (LM fraction) are very difficult to evaluate due to the usual smearing of the ubiquitinated proteins. The quantification suggests 20-30% more ubiquitinated proteins on LM fractions in cells expressing ALIX Δ PRR - this quantification appears to be at odds with the immunofluorescence experiments in Fig. 5A/B showing ubiquitin conjugates almost only on endosomes in cells overexpressing ALIX Δ PRR. Also, the combination of light-blue for ubiquitin and deep-blue for Hoechst (Fig.5b) is not helpful in this respect.
6. The statement that ALIX Δ PRR only affects a subset of cargoes and not endo/lysosomes in general appears to be a bit premature (line 250-252). After all, there is massive ESCRT-III recruitment and ubiquitin accumulation on these endosomes. Either rephrase the sentence, or demonstrate this more clearly and perhaps check cathepsin D trafficking, pH etc. in combination with ultrastructural analysis
7. Fig. 6C: Un-transfected control cells are not visible on the images for CD81 staining, in which the CD81 staining looks any different from the transfected cells. In any case it would be helpful to show fields with more than 2 or 3 cells.
8. Are ALIX Δ PRR expressing cells producing additional exosomes or the same number with just

different cargo? Are ALIX and ALIX Δ PRR on the same MVB and but produce each different subsets of exosomes or are these mixed exosomes (one exosomes contains ALIX and ALIX Δ PRR and hence mixed cargo?)

9. Fig.7A - the absence of CD9 in cells expressing ALIX Δ PRR is curious and does not really fit to Fig. 6/C, where total CD9 protein levels are not affected by expression of ALIX Δ PRR. please clarify.

10. Is ALIX binding (directly) to these proteins sufficient for sorting? Could you force proteins into exosomes by linking them to ALIX?

Minor points:

1. Introduction: A number of manuscripts have recently shown that ESCRT-III needs Vps4 to do work. I would recommend to rephrase the introduction accordingly (e.g. line 65)

2. Fig. 1I, please include PNS

3. siRNA depletion for STAM1 is not strong -makes it difficult to interpret the result. Maybe try HRS instead?

Reviewer #3 (Comments to the Authors (Required)):

Larios et al., present a paper examining the role of ALIX in recruiting ESCRT-III to endosomes. For cytokinesis and viral release, an alternate ESCRT-I-independent and ALIX-dependent mechanism of recruiting ESCRT-III to sites of membrane remodelling exists through direct interaction of ALIX's Bro1 domain with CHMP4 proteins. Current models suggest that the C-terminal PRR of ALIX provides an auto-inhibitory function by folding back against the N-terminal half of the protein. Here, the authors employ the overexpression of a PRR deletion of ALIX to demonstrate that it can assemble on endosomal membranes and recruit downstream ESCRT-III components. In an elegant in-vitro binding assay, the authors showed that LBPA present in flattened GUVs could recruit ALIX which could recruit CHMP4B. In cells, when overexpressing ALIX Δ PRR, whilst no perturbation of EGFR degradation was observed, the authors found that ALIX Δ PRR was more efficiently incorporated onto extracellular and aided incorporation of tetraspanins including CD9, CD63 and CD81 into these structures, whereas depletion of ALIX reduced exosomal incorporation of these tetraspanins.

I think the manuscript is interesting, but have some concerns. The manuscript is heavily reliant upon overexpression of truncated ALIX proteins (without examination of the effect of overexpression of WT ALIX protein), and whilst this truncation clearly localises more strongly to intracellular structures than full length ALIX, given the well described interactions between ALIX's Bro1 domain and CHMP4's C-terminal helix, the subsequent endosomal recruitment/assembly of CHMP4 in this context is not surprising. Whilst there is more ALIX Δ PRR on secreted EVs from cells overexpressing ALIX Δ PRR, is this enhanced presentation simply a consequence of more ALIX being on endosomes in the 1st place? I think the manuscript is missing loss-of-function data to uncover how essential this pathway is in the context of endosomal sorting and exosome biogenesis and molecular data to explain how tetraspanins are sorted by ALIX.

Additionally, ALIX has well described roles in exosome production and sorting of cargo to exosomes, which I feel could have been more broadly incorporated into this manuscript. The involvement in tetraspanin incorporation is interesting, but I feel a little too preliminary and requiring more molecular detail and integration with the existing literature on ALIX. In addition, the (lack of) parallels with ESCRT-dependent viral release, where ALIX's PRR was essential (Fisher et al., 2007, Usami et al,

2007) could be more fully discussed. Tetraspanins have also been proposed to drive cargo incorporation into ILV in an ESCRT-independent manner (e.g., van Neil et al., 2012) - it is not clear how this would work if they themselves are subject to ESCRT-dependent inclusion.

Some slightly more specific points:

There are instances of overinterpretation in the text. E.g., L207 states that 'the results show that CHMP4B recruitment onto late endosomal membranes strictly depends on LBPA and ALIX' and L226 states 'LBPA and ALIX are necessary and sufficient for recruitment of ESCRT-III onto late endosomal membranes'. The authors haven't shown this, they have only shown that the endosomal localisation of GFP-CHMP4B that is driven by overexpression of ALIX-dPRR-mCherry depends upon [an interaction surface in ALIX that can bind] LBPA. Their subsequent argument that exosomes from ALIX depleted cells are both produced and contain less tetraspanins, indicates that ALIX/LBPA is dispensable for ILV generation, rather than being necessary and sufficient for ESCRT-III activity at this organelle.

For the EGFR degradation experiments, as this is a transient overexpression experiment, untransfected cells will be in the majority and will likely swamp any effect on EGFR degradation observed in the transfected cells. Why not stably express the mutant and remove endogenous ALIX? Related to this, although the western blot in 5F makes it appear you are at near endogenous levels, the transfection efficiency will mean that you are expressing much more of the deletion mutant in the successfully expressing cells.

I find Figure 7 hard to interpret - the images appear very pixelated and the resolution seems insufficient to accurately assess colocalization. It isn't clear why the authors have chosen LAMP1 as a marker as this will decorate lysosomes as well as MVBs. CD9 and CD81 don't appear to show any vesicular staining, apart from the CD81 in the dPRR transfected cells, which is the opposite of what the quantification reports. The CD63 was dotted and appeared to colocalise with the LAMP1 in all cells, whether or not they were expressing mCherry or mCherry-ALIXdPRR, which again, isn't what the quantification reports. I think what you are trying to say in the text is that in cells overexpressing ALIX dPRR, the tetraspanins are hyper-secreted, possibly on exosomes, but I am afraid that don't think the data supports this conclusion. In 7C, in the absence of ALIX, I can't really see any reduced CD63 incorporation into EVs on the western blot. I think that this part of the manuscript is quite weak and needs a lot more characterisation of the extracellular vesicle fraction to ensure that they are exosomes, rescue experiments and a molecular explanation of how ALIX controls tetraspanin incorporation into ILVs, and then EVs.

Discussion of the dispensability of ALIX for EGFR degradation is missing a key reference from the Woodman lab (Doyette et al., 2008) and the cited Schmidt paper claims ALIX has an active role in suppressing EGFR degradation, which is contrary to their argument.

Figure 1G, very few of the LBPA+ structures are ALIX or CHMP4B positive.

Minor:

Some of methodology could be made clearer - e.g., ALIX was described to be inserted in the XhoI site of pCMV-tag3, and then mCherry was then inserted downstream using the XhoI and ApaI sites. I'm not sure that this could happen as the XhoI digestion would liberate the ALIX. The methods section doesn't describe the untagged constructs - presumably these were used these in the paper too?

The resolution of the blots prevents me from being certain, but the Rab5 blot in the LM fraction of

S5A and 5D appear duplicated, but slightly differently cropped. I will grant that they are from the same type of experiment, but the accompanying ALIXdPRR blots are different and the legends claim that Rab5 was used in normalisation. I may be wrong, but if these were unintentionally duplicated, it may be best to exchange them.

Related to this, the degree of pixilation in all of the supplied images is poor, to the extent that it is hard to meaningfully assess the data in the IF/westerns. This may just be the result of downsampling during the submission process, but I would recommend improving it.

ALIX depletion in 7C seems to stabilise CD81 - is this consistent?

Reviewer #1:

In this manuscript, Larios et al document that the Bro domain-containing protein, ALIX, recruits ESCRT-III subunits to endosomes and facilitates the selective recruitment of tetraspanins to internal vesicles of multivesicular endosomes. ALIX is well known to interact with ESCRT-I and ESCRT-III subunits, to regulate intraluminal vesicle (ILV) formation by the classical ESCRT pathway in yeast, and to facilitate the budding of certain retroviruses by recruiting their Gag proteins, but its role in unconventional ILV formation is not understood. Because ALIX appears to be largely autoinhibited by an interaction of its proline-rich domain (PRR) with the Bro domain, the authors here employ a mutant ALIX that lacks the PRR to overexpress fully active ALIX. They show convincingly that this results in the recruitment of the CHMP4 family ESCRT-III subunits - and, to a lesser extent, a few other ESCRT-III components - to late endosomes both in vivo and, for CHMP4, in vitro. Mutations that disrupt the Bro domain or its interaction with the late endosome/ lysosome-restricted lipid, LBPA, block this recruitment, suggesting a pathway from LBPA to ALIX to ESCRT-III. A consequence of this recruitment is the accumulation of polyubiquitin on the endosomes and a partial depletion of tetraspanins CD63, CD9 and CD81 from late endosomes, together with enhanced tetraspanin incorporation into exosomes. The authors conclude that ALIX functions in a non-canonical ESCRT-III-dependent pathway to sort tetraspanins and other cargoes into ILVs for ultimate secretion as exosomes.

In dissecting the role of ALIX in ILV formation, the paper addresses a question that is of interest to the field and that is still incompletely answered. The data in the first four figures are pristine and clearly document a direct role for ALIX in recruiting several ESCRT-III subunits (and curiously, the ESCRT-II subunit VPS22, which is not really discussed) to endosomes independently of other ESCRT subunits. The in vitro data in Figure 3 using purified components are particularly convincing. The data in the last three figures are suggestive of an important role in protein sorting. Thus, in principle this paper would be of interest to the readers of JCB.

We thank the reviewer for her/his constructive comments and support to the publication of our article. We hope that we address all issues raised by this reviewer in the following.

However, in its current form the paper has several deficits, as detailed in the comments described below. There are two major concerns. First, at the "why does this matter" level, the authors fail to describe the solid physiological context within which an alternative pathway for ILV formation might function. Such a context exists and is begging for an explanation, but is neither discussed nor experimentally addressed in this paper (points # 1, 2 and 5). This should be remedied.

Concerning the "why does it matter" issue, we are sorry if we gave the impression to this reviewer that we did not provide the solid physiological context that supports the existence of an alternative pathway for ILV. As requested by this reviewer, we now discuss the alternative fates of ILVs and alternative mechanisms of ILV formation (points 1-2), including the necessary references. However, we did not provide more experimental evidence to assess in more detail the role of the pathway in tetraspanin sorting and accumulation in exosomes, following the recommendation of the editor that this point does not need to be experimentally addressed in the revision.

Second, the data supporting a role for this pathway in sorting ubiquitylated cargo (point #3) and tetraspanins for exosome incorporation (point #4) are weak and require additional controls, and the involvement of CHMP4B and LBPA binding in tetraspanin sorting needs to be confirmed (point #4). Several additional minor concerns also should be addressed (points #6-10). If the manuscript is appropriately modified to address these weaknesses, it should make a solid contribution to the field.

As requested by this reviewer, we have addressed the points 3-4 and 6-10 mentioned above, and our detailed reply is below.

Specific concerns:

1. In the Introduction citing alternative fates of ILVs, the authors ignore additional fates of ILVs other than degradation or exosomal release. For example, in antigen-presenting cells MHC class II molecules on internal vesicles are loaded with peptides for presentation at the plasma membrane (Peters et al., 1991, *Nature* 349: 669; Kleijmeer et al., 2001, *J. Cell Biol.* 155: 53; Zwart et al, 2005, *Immunity* 22: 221), and internal vesicles of multivesicular endosomes in melanocytes seed the formation of PMEL-containing amyloid fibrils upon which melanins deposit during melanosome maturation (Berson et al., 2001, *Mol. Biol. Cell* 12: 3451; Hurbain et al., 2008 *Proc. Natl. Acad. Sci. USA* 105: 19726). These are clear examples whereby alternative forms of ILVs are preserved from degradation and should be cited.

We apologize for having failed to cite these important papers that report two additional fates of ILVs, peptide loading onto MHC-II molecules (Peters et al. 1991; Kleijmeer et al 2001; Zwart et al, 2005) and backfusion with the limiting membrane (Kleijmeer et al 2001) in antigen-presenting cells, and formation of melanosomes in melanocytes (Berson et al 2001, Hurbain et al 2008). Although we feel that the fate of ILVs in specialized cell types raise mechanistic issues that extend beyond the scope of the present study, we have modified the text to clarify this issue and we have added these references as well as reviews that cover these topics.

2. The authors also fail to cite evidence supporting ESCRT- and VPS4-independent mechanisms for ILV formation that have been well-documented and assessed mechanistically (Theos et al., 2006, *Dev. Cell* 10: 343; Trajkovic et al., 2008, *Science* 319: 1244-1247; van Niel et al., 2011, *Dev. Cell* 21: 708; van Niel et al., 2013, *Cell Rep.* 13: 43), and evidence that CD63 is present on a distinct population of ILVs from ESCRT-dependent cargoes (Edgar et al., 2014, *Traffic* 15: 197). These are directly relevant to the proposed mechanism described in this paper. Moreover, the involvement of Apolipoprotein E in the formation of PMEL- and CD63-containing ILVs (Van Niel et al., 2013) already provided a potential mechanism for formation of that class of ILVs. The authors need to present and clearly distinguish this class of ILV formation from the well-characterized ESCRT-dependent process; in the document as it stands, only the latter is considered. For example, paragraph 2 of the Introduction needs to be qualified for the types of proteins that are incorporated into ILVs or specified for ESCRT-0/I-dependency. The role of the proposed pathway in these alternative fates should also be addressed experimentally (see point # 5).

As already mentioned above, we are sorry that we did not mention the exciting papers quoted by this reviewer, which conclusively show the existence of alternative mechanisms responsible for the biogenesis of ILVs in specialized cell types. We feel that a detailed discussion of the ApoE-, PMEL- and CD63-dependent formation of melanosomes in melanocytes or the ceramide-dependent formation of proteolipid-containing exosomes in oligodendrocytes extends beyond the scope of this paper. However, as requested by this reviewer, we have modified the text to clarify this issue and we have added the Trajkovic 2008, Theos 2006, van Niel 2011 and 2015 and Edgar 2014 references mentioned by this reviewer in the Introduction and Discussion sections of the paper.

3. The data in Figure 5A-C and Figure S5B are convincing of some accumulation of ubiquitylated proteins in late endosomes of cells overexpressing ALIXdeltaPRR, but the conclusion that these proteins are cargoes (on line 242) is not supported - it is not clear if the polyubiquitin is conjugated to cargo, released free in the endosome, or associated with ubiquitylated effector proteins on the cytosolic face. Identification of some of the polyubiquitylated proteins (such as by purification and mass spectrometry analyses of polyubiquitylated proteins from ALIXdeltaPRR-expressing cells relative to ALIXdeltaPRR-I212D-expressing cells) would be helpful in addressing

this question. Are specific polyubiquitylated bands detected by SDS-PAGE of overexpressing cells relative to controls? Note, the authors may wish to discuss their results in light of evidence that tetraspanins in yeast are sorted to MVBs via the classical ESCRT-dependent pathway following ubiquitylation (MacDonald et al., 2015, Dev. Cell 33: 328-342).

This reviewer mentions that the conclusion that ubiquitylated proteins are cargoes is not supported, since it is not clear if polyubiquitin is conjugated to cargo, released free in the endosome, or associated with ubiquitylated effector proteins on the cytosolic face. We are sorry if the text was not clear. To the best of our knowledge, free polyubiquitin does not exist in cells. Also, the monoclonal antibody FK2 (Enzo Life Sciences) that we used only recognizes K29-, K48-, and K63-linked mono- and polyubiquitylated proteins (and not free ubiquitin) – see: <http://www.enzolifesciences.com/BML-PW0755/mono-and-polyubiquitylated-conjugates-monoclonal-antibody-fk2-biotin-conjugate/>. Finally, ubiquitination remains the hallmark of ILV formation on endosomes, whether ubiquitin is conjugated to cargo molecules or effector proteins during ILV formation. Indeed, ESCRTs themselves are ubiquitinated and the network of more or less weak interactions between ubiquitin and ubiquitin-binding domains contributes to cargo sorting and ILV formation (see for example Mageswaran et al. 2014 Traffic. 15: 212–29, from the Babst group).

To address this issue more directly, we have generated a CD9 mutant that cannot be ubiquitinated after replacement of all three cytoplasmic Lys residues with Arg (CD9/3R). We now show that CD9/3R levels in exosomes are significantly reduced when compared to WT (new Fig 8G-H), supporting the notion that ubiquitination plays a role in sorting to exosomes. We also find that this defective incorporation could be partially rescued by ALIX Δ PRR overexpression, presumably because incorporation of the CD9/3R mutant into exosome was facilitated by interactions with other tetraspanins, upon stimulation of exosome production.

We thank this reviewer for suggesting the MacDonald et al paper, which we now discuss in the revised version of the Discussion.

As suggested by the reviewer, we purified endosomes from cells expressing ALIX Δ PRR by subcellular fractionation, and analyzed the proteome of these fractions by mass spectrometry. These preliminary new data indeed suggest that ubiquitylated proteins accumulate in the endosomes of ALIX Δ PRR expressing cells. Unfortunately, we were unable to repeat these experiments because of technical problems with the equipment, have therefore not added the data to the revised paper. However, we can of course show the data to this reviewer if he/she wishes to.

4.1 The quantification of the experiments in Figures 6 and 7 is interesting but not terribly dramatic, and the effects shown could potentially be due to indirect effects of ALIX depletion or overexpression. They are also not completely consistent. For example, while ALIX overexpression has no consistent effect on syntenin accumulation in exosomes, ALIX depletion does. These experiments would be much more convincing with additional controls. First, quantification of ALL markers - including EGFR, flotillin, and integrin components, as well as a solid negative control like calnexin - in both exosome fractions (relative to cell lysates by blotting) and LAMP1+ endosomes (by immunofluorescence microscopy) should be done in both ALIX Δ PRR overexpressers and ALIX KD cells; if the results are interpreted correctly, then opposite results should be seen for most/ all markers in the knockdowns and the overexpressers (and exceptions such as syntenin distinguished as either outliers or the rule).

We disagree with this reviewer that effects observed after ALIX overexpression and depletion are not consistent. Our observations that ALIX depletion significantly reduces exosome formation is comforting as it significantly strengthens our conclusions. But, these observations are not inconsistent with the effect of ALIX overexpression. The relationship between gain-of-function and loss-of-function is not linear, and

may depend on protein abundance, stoichiometry/affinity of interactions with partners and number of partners (e.g. Schmid and McMahon Nature 2007).

To address this reviewer's concerns and as suggested by the reviewer, all proteins tested after ALIX Δ PRR expression (old Fig 6A) were also tested after ALIX depletion including EGFR (new Fig 7), integrin β 3 and integrin α 6 (new Fig 8C-D). Moreover, and as requested we have also included the immunofluorescence analysis of EGFR (new Fig7), syntenin and flotillin, as well as their quantification within LAMP1 endosomes (new Fig 9). In the new Fig 9, we have also added micrographs that illustrate the distribution of GM130 and calnexin, which are not detected in LAMP1 endo-lysosomes. Consistent with these observations, GM130 and calnexin are not detected in exosomes (new Fig 8A and Fig S3A). Neither is free GFP, when using GFP-expressing cells — or CHMP4-GFP, as expected (new Fig S3A).

We also wish to mention that we do not necessarily expect, in contrast to the statement made by this reviewer, that opposite results should be seen for most/ all markers after knockdown or overexpression by light microscopy, depending on their steady state distributions. For example, CD63 is a permanent resident of endo-lysosomes, while CD81 is primarily present at the plasma membrane, and thus ALIX depletion/overexpression does not affect both proteins equally. In any case, and as requested by this reviewer, we now compare in this revised version all markers (including EGFR, flotillin, and syntenin), and negative control (calnexin, GM130) both in exosome fractions (relative to cell lysates by blotting) and LAMP1-positive endosomes, and in both in ALIX Δ PRR overexpressers and ALIX KD cells. These data further confirm our conclusions. They show that ALIX Δ PRR expression decreases the amounts of tetraspanin in the LAMP1-positive endo-lysosomes – consistent with our findings that ALIX Δ PRR expression stimulates tetraspanin exosomal secretion (new Fig 9). Conversely, ALIX knockdown does not affect the tetraspanin endosomal distribution, in agreement with our observations that the treatment inhibits exosomal secretion (new Fig S4 and Fig S5). To the best of our knowledge this is the first time that the distribution of tetraspanin is compared under conditions that do or do not promote exosome secretion.

4.2 Second, a more robust control for ALIX specificity should be included. A good one to use for the overexpression would be the QQ mutant described in Figure 3 - if ALIX functions by the mechanism described in previous figures, then overexpression of the QQ mutant should not impact tetraspanin accumulation in exosomes. Similarly, an ESCRT-I or -II component kd should impact incorporation into exosomes of EGFR but not tetraspanins. Third, it would be helpful to link these observations to the recruitment of CHMP4B or other ESCRT-III components by assessing the impact of CHMP4B kd or VPS4 dominant mutant overexpression. These experiments are needed to test whether the observations on exosomes and cargo recruitment to endosomal ILVs are linked to the effector recruitment phenotypes observed in the previous figures.

We thank this reviewer for the excellent suggestion to use the QQ mutant, with a defective LBPA binding site, as control. To address this issue, we have carried out two series of experiments to better illustrate ALIX specificity. First, in the original version, we could show that purified recombinant CHMP4B was recruited onto LBPA-containing bilayers by purified recombinant ALIX. We now show that CHMP4B recruitment is abolished when using the QQ mutant (new Fig 4I-J). Second, and as requested by this reviewer, we also show that syntenin, which had been shown to mediate ALIX-dependent exosome production (Baietti et al. Nat Cell Biol. 2012), and tetraspanins did not accumulate in exosomes after overexpression of the QQ mutant (new Fig S3B, D), much like after ALIX KD (new Fig 8C, D), consistent with observations that this mutant is unable to recruit CHMP4 in vivo and in vitro. By contrast, the QQ mutant had little if any effect on the presence of EGFR in exosomes (new Fig S3B, D).

As requested by this reviewer, we also investigated the possible role of other ESCRT components. We tried to test the impact of CHMP4B depletion or overexpression of the VPS4 dominant-negative mutant. Unfortunately, these treatments were lethal in our cells, and thus the data cannot be interpreted. We then selected CHMP6, because of its well-established function as ESCRT-III nucleation factor – in addition to ALIX as proposed in this manuscript. CHMP6 is not recruited to endosomes upon ALIX Δ PRR expression (old Fig 2F, new Fig 3F) and not required for ALIX-dependent CHMP4B membrane association (old Fig 4, new Fig 5), but we now find that CHMP6 depletion decreases the secretion of exosomes containing tetraspanins and syntenin (new Fig 8E, F), much like ALIX, supporting the view that both CHMP6- and ALIX-dependent ESCRT-III nucleation mechanisms are involved in exosome biogenesis.

5. Given that classical ESCRT-independent pathways for ILV formation have been previously described, it would be of great benefit to test if the ALIX-CHMP4 pathway described here plays a role in these pathways. For example, is the sorting of exogenous PMEL (Theos et al., 2006 and van Niel et al., 2011) or PLP (Trajkovic et al., 2008) to ILVs blocked by ALIX depletion? These could easily be tested by expressing these proteins by transfection in the HeLa cells described here, and would go a long way in providing a solid physiological context for the described pathway.

As mentioned above, we did not provide additional experimental evidence to assess in more detail the role of the pathway in tetraspanin sorting and accumulation in exosomes, following the recommendation of the editor that this point does not need to be experimentally addressed in the revision.

Minor concerns:

6. In the text description of Figure 2, VPS22 is mistakenly referred to as an accessory ESCRT-III component; it is an ESCRT-II subunit. Rather, CHMP1A and CHMP1B are more considered accessory ESCRT-III components and homologs of the yeast Did2 protein.

Sorry for the typo. This was corrected.

7. In the text describing Figure 3A,B, it is misleading in the text to interpret this as a failure of ALIX Δ PRR-I212D to recruit GFP-CHMP4B to endosomes, since it is not itself localized to endosomes. Taken at face value, these data support the conclusion that CHMP4B recruits ALIX to endosomes, which then recruits more CHMP4B, and directly contradicts the statement on line 187: "Since ALIX binds late endosomes via LBPA and then recruits ESCRT-III in vivo" - the "then" is misleading in the context of these results.

The text was corrected as requested.

8. In Figure 6A, it is not clear how the authors come up with their number of a 20X increased yield of ALIX Δ PRR in the exosome fraction relative to endogenous ALIX; the image shown certainly does not support this (maybe at most a 2-3X increase in the exosome fraction relative to the amount in the cell lysate). If the number is correct, then a more representative image should be shown. Similarly, in Figure 6B it is not clear that the recovery of tetraspanins and other proteins was normalized to the level in the cell lysates - was it?

We thank the reviewer for making us aware about the incorrect fold increase in ALIX Δ PRR yield in the text. The appropriate value is 6X and this has been corrected. We did not normalize tetraspanins and

other proteins to cell lysates. To facilitate comparison, all proteins are normalized relative to free GFP, as a control. This is now clarified in the legend of new Fig 8.

9. The authors' earlier papers showed that LBPA was limited to the interior of late endosomes/lysosomes. How is it available for binding to ALIX on the cytoplasmic face? Or does ALIX bind to some other ligand on the cytoplasmic face that is induced by LBPA accumulation on the interior? This should be addressed in the text of the introduction or discussion.

LBPA is by no means limited to the lumen of endo-lysosomes: it is abundant within, but not restricted to, intra-luminal membranes. This is consistent with our observations that LBPA binds ALIX on the endosomal membrane. For the reviewers' eyes only, here is a preliminary observation, showing the micrograph of endosomes labeled with the RFP-tagged single chain variable fragment of our anti-LBPA mAb antibody 6C4 expressed in the cell cytoplasm (in collaboration with Sandrine Moutel & Frank Perez, Curie Institute). This clearly shows that LBPA is accessible to proteins in the cytosol.

10. Figures S3 and S6 need a scale bar.

Figure S3 and S6 were modified (new Fig 2D and Fig 7E).

Reviewer #2:

The Gruenberg and Roux labs present experimental evidence that suggests that ALIX and ESCRT-III somehow sequester and package a specific subset of cargo molecules (tetraspanins) for exosomal secretion. As such the results provide an interesting starting point to begin to better understand on a mechanistic level (finally!), how cargo molecules are selectively packaged into exosomes. A few control experiments should be added to strengthen the conclusions:

Major points:

1. In the majority of experiments expression of ALIX Δ PRR is used to recruit ESCRT-III to late endosomes. In these experiments, overexpression of wt-ALIX appears to be the appropriate control (e.g. Fig.1e-j and others, particular exosome experiments in Fig. 6/7).

We thank the reviewer for this comment. When expressed in HeLa cells, both full-length ALIX and ALIX Δ PRR show both a cytosolic and punctate (endosomal) distribution (Fig 1B). In this revised version we now show that full-length ALIX, much like ALIX Δ PRR, exhibits the capacity to recruit CHMP4B to endosomes (new Fig S1C-D). However, earlier data indicate that overexpression of full-length ALIX can be detrimental, because of ALIX pro-apoptotic activity (see Trioulier et al. JBC. 2004; Wu et al. Oncogene 2002), due to interactions of the PRR domain with the protein ALG-2 (Vito et al. J Biol Chem 1999; Missotten et al. Cell Death Differ 1999), which is necessary for cell death (Vito et al. Science. 1996). Since apoptotic cells release cell fragments and apoptotic bodies – which can contaminate exosomal

fractions – we decided to use the version of ALIX deleted of the auto-inhibitory domain (PRR) but still dimerization-competent (Pires et al. Structure 2010), to avoid problems in our exosome experiments. This has been clarified in the paper.

2. The finding that interaction of ALIX Δ PRR with ESCRT-III leads to a mutual reinforcement of endosomal localization is interesting. Do the two proteins already interact in the cytosol and 'land' together on LBPA containing endosomes as preformed complexes?

This is a very interesting question, but not easily addressed — unless putative ALIX-ESCRT-III interactions are strong. Our view, however, is that ALIX and ESCRT-III are not associated in the cytosol. The Weissenhorn lab has shown that ALIX dimers target ESCRT-III filaments and are involved in HIV budding (Pires et al Structure 2009), while we found that ALIX dimerization depends on LBPA-dependent endosome association: we proposed that ALIX dimerization and activation is triggered by interactions with LBPA on endosomes (Bissig et al., Dev Cell 2013).

3. The in vitro recruitment assays should be performed by comparing the capability of ALIX Δ PRR / ALIXI212D to recruit Chmp4b.

We thank the reviewer for this helpful suggestion, and we carried out these experiments. The in vitro assay was carried out using not only the I212D mutant but also the QQ mutant, which does not bind LBPA. Consistent with our in vivo observations, our new data show that both mutants fail to recruit CHMP4 onto LBPA-containing membranes (new Fig 4I-J). In addition, following the recommendation of reviewer 1, we also show now that overexpression of the QQ mutant fails to cause the accumulation tetraspanins in exosomes (new Fig S3B-D), much like after ALIX KD (Fig 7C-D, new Fig 8C-D), consistent with observations that this mutant is unable to recruit CHMP4 in vivo and in vitro.

4. Fig.5a: The localization of ubiquitin in cells expressing ALIX or ALIX Δ PRR should be shown.

As requested, we now show the localization of conjugated ubiquitin in cells expressing ALIX Δ PRR (new Fig 6B).

5. The differences for the ubiquitinated proteins in Fig. S5A (LM fraction) are very difficult to evaluate due to the usual smearing of the ubiquitinated proteins. The quantification suggests 20-30% more ubiquitinated proteins on LM fractions in cells expressing ALIX Δ PRR - this quantification appears to be at odds with the immunofluorescence experiments in Fig. 5A/B showing ubiquitin conjugates almost only on endosomes in cells overexpressing ALIX Δ PRR. Also, the combination of light-blue for ubiquitin and deep-blue for Hoechst (Fig.5b) is not helpful in this respect.

We are sorry if this point was not clear in the original manuscript. First, it should be stressed that our immunofluorescence data showing the accumulation of ubiquitin conjugates in endosomes after overexpression of ALIX Δ PRR (old Fig 5A-B, new Fig 6C-D) agree nicely with earlier studies reporting similar observations after overexpression of other ESCRTs (Bishop et al. J. Cell Biol 2002). We believe that two main reasons account for the difference between light microscopy (old Fig 5A-B, new Fig 6C-D) and biochemistry (old Fig S5, new Fig 6E-F): i) The background staining of the antibody against conjugated ubiquitin is high in blots, and thus the control staining was overestimated in light membranes (old Fig S5A, LM). This has been corrected in the new version (new Fig 6E). ii) in immunofluorescence experiments, cells were permeabilized with saponin prior to fixation in order to remove cytosolic proteins and thus reduce the cytosol staining. This point was clarified in the paper.

The colors of figure 5B (new Fig 6C) were modified to obtain a better contrast between channels.

6. The statement that ALIX Δ PRR only affects a subset of cargoes and not endo/lysosomes in general appears to be a bit premature (line 250-252). After all, there is massive ESCRT-III recruitment and ubiquitin accumulation on these endosomes. Either rephrase the sentence, or demonstrate this more clearly and perhaps check cathepsin D trafficking, pH etc. in combination with ultrastructural analysis

We have carried out the experiments suggested by this reviewer. We now also show that the pH of endosomes and lysosomes is not affected by ALIX Δ PRR, after quantification by automated microscopy using a pH-sensitive probe (new Fig 7I-J). In addition, we also show that the levels of mature cathepsin D are not affected by ALIX Δ PRR overexpression, when compared to mock-transfected or untransfected cells (new Fig 7G-H). However, we agree with this reviewer that our previous conclusion was a bit overstated. We have rephrased the sentence and now conclude: "These data, together with our findings that ALIX Δ PRR does not affect ligand-induced EGFR degradation (Fig 7C-F) indicate that both traffic to the lysosomes and the degradation capacity of endo-lysosomes are not affected by ALIX Δ PRR, and thus suggest that the expression of ALIX Δ PRR does not cause a general traffic jam in the late endosomal pathway, but rather results in the selective retention of a subset of ubiquitinated cargoes."

7. Fig. 6C: Un-transfected control cells are not visible on the images for CD81 staining, in which the CD81 staining looks any different from the transfected cells. In any case it would be helpful to show fields with more than 2 or 3 cells.

We apologize if this was not clear. To clarify this issue, we now quantify CD81 (and CD63) in LAMP1-endosomes by automated microscopy with or without ALIX Δ PRR expression (new Fig 9A) or ALIX depletion by RNAi (new Fig S4) and we show the corresponding micrographs. In addition, to better illustrate the endosomal distribution of CD81 (and CD63), we also show the CD81 (or CD63) staining within LAMP1-positive endo-lysosomes only (the mask used in automated microscopy), after subtraction of the staining present elsewhere. Finally, and as requested, we also show views with both transfected and untransfected cells (new Figure 9A).

8. Are ALIX Δ PRR expressing cells producing additional exosomes or the same number with just different cargo? Are ALIX and ALIX Δ PRR on the same MVB and but produce each different subsets of exosomes or are these mixed exosomes (one exosomes contains ALIX and ALIX Δ PRR and hence mixed cargo?)

The reviewer raises a fundamental question (in subcellular trafficking in general), which is not easily answered in the exosome pathway – or in any other pathway. EGF addition triggers the endocytosis of EGFR: are more EGFR molecules packaged per endocytic vesicle or are more endocytic vesicles formed? Today it seems that both are probably true depending on the EGF dose and the cell type. It is tempting to speculate that exosome production is a regulated pathway, and that increased ALIX Δ PRR levels, by stimulating ESCRT-III membrane association, stimulate exosome production. Similarly, since ALIX Δ PRR likely dimerizes with ALIX, there is no reason to believe that ALIX and ALIX Δ PRR are present on different MVBs and produce different exosomes.

9. Fig.7A - the absence of CD9 in cells expressing ALIX Δ PRR is curious and does not really fit to Fig. 6/C, where total CD9 protein levels are not affected by expression of ALIX Δ PRR. please clarify.

This point is well-taken. And we agree with this reviewer that the data with CD9 were not very convincing. In fact, CD9 turned out to be quite difficult to monitor adequately by immuno-fluorescence with the antibodies we have available. We have therefore removed the analysis of CD9, but we carried out a far more complete analysis of both CD81 and CD63 by immuno-fluorescence.

We have now quantified CD63 and CD81 in LAMP1-endosomes by automated microscopy after ALIX Δ PRR expression and ALIX depletion by RNAi. To better illustrate the endosomal distribution of CD63 and CD81, we also show now the staining intensity of each tetraspanin within LAMP1-positive endo-lysosomes only (the mask used in automated microscopy), after subtraction of the staining present elsewhere (new Fig 9A). These data confirm that ALIX Δ PRR expression decreases the amounts of tetraspanin in the LAMP1-positive endo-lysosomes – consistent with our findings that ALIX Δ PRR expression stimulates tetraspanin exosomal secretion. Conversely ALIX knockdown does not affect the alter the tetraspanin endosomal content, in agreement with our observations that the treatment inhibits exosomal secretion (new Fig S4 and new Fig S5).

10. Is ALIX binding (directly) to these proteins sufficient for sorting? Could you force proteins into exosomes by linking them to ALIX?

Again, the reviewer raises a really interesting issue concerning the mechanism of sorting into exosomes. By analogy with ILV formation, direct binding to ALIX is likely to be sufficient (e.g. see the work of JoAnne Trejo on GPCR sorting). We believe that it is in principle possible to force proteins into exosomes by linking them to ALIX. However, the experiment is not easy and may be difficult to interpret, since such forced interactions should not interfere with ALIX functions, and in particular with ALIX dimerization capacity.

Minor points:

1. Introduction: A number of manuscripts have recently shown that ESCRT-III needs Vps4 to do work. I would recommend to rephrase the introduction accordingly (e.g. line 65)

As requested, the introduction was rephrased and the corresponding references were added (Adell et al., 2014; Adell et al., 2017; Mierzwa et al., 2017).

2. Fig. 1I, please include PNS

As requested, Figure 1I was modified (new Fig 1G).

3. siRNA depletion for STAM1 is not strong -makes it difficult to interpret the result. Maybe try HRS instead?

We share the view of the reviewer, and initially the experiments in Fig 4 (new Fig 5) were performed after HRS depletion. However, HRS KD caused massive cell death, and results were not interpretable. For this reason, we decided to deplete STAM proteins (STAM1 and STAM2). Although incomplete, STAM1 was still decreased 60% after KD, and this is clearly indicated in the text.

Reviewer #3:

Larios et al., present a paper examining the role of ALIX in recruiting ESCRT-III to endosomes. For cytokinesis and viral release, an alternate ESCRT-I-independent and ALIX-dependent mechanism

of recruiting ESCRT-III to sites of membrane remodelling exists through direct interaction of ALIX's Bro1 domain with CHMP4 proteins. Current models suggest that the C-terminal PRR of ALIX provides an auto-inhibitory function by folding back against the N-terminal half of the protein. Here, the authors employ the overexpression of a PRR deletion of ALIX to demonstrate that it can assemble on endosomal membranes and recruit downstream ESCRT-III components. In an elegant in-vitro binding assay, the authors showed that LBPA present in flattened GUVs could recruit ALIX which could recruit CHMP4B. In cells, when overexpressing ALIXdPRR, whilst no perturbation of EGFR degradation was observed, the authors found that ALIXdPRR was more efficiently incorporated onto extracellular and aided incorporation of tetraspanins including CD9, CD63 and CD81 into these structures, whereas depletion of ALIX reduced exosomal incorporation of these tetraspanins.

I think the manuscript is interesting, but have some concerns. The manuscript is heavily reliant upon overexpression of truncated ALIX proteins (without examination of the effect of overexpression of WT ALIX protein), and whilst this truncation clearly localises more strongly to intracellular structures than full length ALIX, given the well described interactions between ALIX's Bro1 domain and CHMP4's C-terminal helix, the subsequent endosomal recruitment/assembly of CHMP4 in this context is not surprising. Whilst there is more ALIXdPRR on secreted EVs from cells overexpressing ALIXdPRR, is this enhanced presentation simply a consequence of more ALIX being on endosomes in the 1st place? I think the manuscript is missing loss-of-function data to uncover how essential this pathway is in the context of endosomal sorting and exosome biogenesis and molecular data to explain how tetraspanins are sorted by ALIX.

We thank the reviewer for this comment and we are sorry if this was not clear in the original manuscript. When expressed in HeLa cells, both full-length ALIX and ALIX Δ PRR show both a cytosolic and punctate (endosomal) distribution (Fig 1B). In this revised version we now show that full-length ALIX, much like ALIX Δ PRR, exhibits the capacity to recruit CHMP4B to endosomes (new Fig S1C-D). However, earlier data indicate that overexpression of full-length ALIX can be detrimental, because of its pro-apoptotic activity (see Trioulier et al. JBC. 2004; Wu et al. Oncogene 2002), due to interactions of ALIX PRR with the protein ALG-2 (Vito et al. J Biol Chem 1999; Missotten et al. Cell Death Differ 1999), which is necessary for cell death (Vito et al. Science. 1996). Since apoptotic cells release cell fragments and apoptotic bodies, we decided to avoid problems in our exosome experiments, and thus used the version of ALIX deleted of the auto-inhibitory domain (PRR) but still dimerization-competent (Pires et al. Structure 2010). This has been clarified in the paper.

Much like this reviewer, we are also convinced that enhanced amounts of secreted ALIX is a direct consequence of more ALIX being on endosomes in the 1st place. The interesting point is that this increase is accompanied by a selective increase in the incorporation of tetraspanins into exosomes and secretion. To address this reviewer's concerns, we have carried out a number of additional experiments:

1) Specificity of LBPA-dependent ALIX binding and formation of exosomes containing tetraspanins. We have carried out two series of experiments to better illustrate the specificity of ALIX. First, in the original version, we could show that purified recombinant CHMP4B was recruited onto LBPA-containing bilayers by purified recombinant ALIX. We now show that CHMP4B recruitment was abolished when using the QQ mutant (new Fig 4I-J), which fails to bind endosomes and to recruit CHMP4B in vivo (old Fig 3A-E, new Fig 4A-E). In addition, we also show that tetraspanins do not accumulate in exosomes after overexpression of the QQ mutant (new Fig S3B, D), much like after ALIX KD (old Fig 7C-D, new Fig 8C-D), consistent with observations that this mutant is unable to recruit CHMP4 in vivo and in vitro. By contrast, the QQ mutant did not affect the sorting of EGFR into exosomes (new Fig S3B, D).

2) Loss of function. We show that ALIX depletion by RNAi decreases the secretion of exosomes containing tetraspanins and syntenin, which had been previously shown to mediate ALIX-dependent

exosome production (Baietti et al. Nat Cell Biol. 2012). We also investigated whether the ESCRT subunit CHMP6, a well-established ESCRT-III nucleation factor, was involved in exosome production. Although CHMP6 is not recruited to endosomes upon ALIX Δ PRR expression (old Fig 2F, new Fig 3F) and is not required for ALIX-dependent CHMP4B membrane association (old Fig 4A-C, new Fig 5A-C), we now find that CHMP6 depletion with siRNAs decreases the secretion of exosomes containing tetraspanins and syntenin (new Fig 8E-F), much like after ALIX depletion. These experiments support the view that parallel pathways, involving CHMP6- and ALIX-dependent ESCRT-III nucleation, are involved in exosome biogenesis.

Additionally, ALIX has well described roles in exosome production and sorting of cargo to exosomes, which I feel could have been more broadly incorporated into this manuscript. The involvement in tetraspanin incorporation is interesting, but I feel a little too preliminary and requiring more molecular detail and integration with the existing literature on ALIX. In addition, the (lack of) parallels with ESCRT-dependent viral release, where ALIX's PRR was essential (Fisher et al., 2007, Usami et al, 2007) could be more fully discussed. Tetraspanins have also been proposed to drive cargo incorporation into ILV in an ESCRT-independent manner (e.g., van Neil et al., 2012) - it is not clear how this would work if they themselves are subject to ESCRT-dependent inclusion.

As requested, we discuss the role of ALIX in exosome production and cargo sorting, including in particular the studies of Guido David and Pascale Zimmerman (Baietti et al. Nat Cell Biol 2012) and of Gisou van der Goot (Abrami et al. Cell Reports 2013), as well as the work of Mika Simons, which challenged the role of ALIX and ESCRTs in exosome production in oligodendrocytes (Trajkovic et al. Science 2008).

This reviewer acknowledged the fact that our work on tetraspanins is interesting, but he/she was also concerned that the work was a little too preliminary and requiring more molecular detail. To address this issue, we have further investigated the role of protein ubiquitination in this process. Our data indicate that ubiquitination may be involved in protein sorting into exosomes, since ALIX Δ PRR causes the accumulation of conjugated ubiquitin in endosomes. We have now generated a CD9 mutant that cannot be ubiquitinated after replacement of all three cytoplasmic Lys residues with Arg (CD9/3R). We show that CD9/3R levels in exosomes are significantly reduced when compared to WT (new Fig 8G-H), supporting the notion that ubiquitination plays a role in sorting to exosomes. We also find that this defective incorporation could be partially rescued by ALIX Δ PRR overexpression, presumably because incorporation of the CD9/3R mutant into exosome was facilitated by interactions with other tetraspanins, upon stimulation of exosome production.

We now discuss in more detail the role of tetraspanins in the biogenesis of intraluminal membranes, including the ApoE-, PMEL- and CD63-dependent formation of melanosomes in melanocytes (Theos et al Dev Cell 2006; van Niel et al. Dev Cell 2011) and the possible involvement of CD63 in the formation of ILVs in HeLa cells (Edgar et al Traffic 2014). We also mention the ceramide-dependent formation of proteolipid-containing exosomes in oligodendrocytes (Trajkovic et al. Science 2008). This reviewer also wonders how tetraspanins may regulate exosome formation if they themselves are subject to ESCRT-dependent inclusion. However, it is CD63 (and not tetraspanins in general) that has been proposed to regulate ILV formation, and its precise role in non-specialized cell-types is not clear. Edgar et al. show that CD63 depletion by RNAi does not affect ILV formation: the authors propose that competitive relationships exist between ESCRT-dependent and -independent mechanisms of ILV formation. Now, in melanocytes, van Niel et al. show that CD63, but not the related tetraspanin CD81, is involved in melanogenesis. They also show that, while PMEL sorting into melanosomes is ESCRT-independent and CD63-dependent, sorting into ILVs destined for lysosomes is ESCRT-dependent in the same cells. Hence, it is not known whether lysosomal targeting of CD63 itself is ESCRT-dependent in the same cell type.

Finally, and as requested by this reviewer, we now discuss the role of the PRR in viral release vs ILV formation. The reason for the differential requirement of the PRR in these two processes is not clear. Since an ALIX Δ PRR dimerization mutant no longer inhibits viral budding, ALIX Δ PRR may inhibit HIV budding because its dimerization capacity is no longer regulated (Pires et al. Structure 2009). However, the same dimerization mutant also inhibits ALIX functions at the endosome (Bissig et al Dev Cell 2013). Hence, some factors necessary for HIV release, but not ILV formation, may interact with the PRR of ALIX. Interestingly, ALIX Δ PRR also inhibits cytokinesis, because the PRR contains CEP55 binding site. During cytokinesis, CEP55 recruits ALIX at the midbody, and ALIX in turn recruits other ESCRTs, including CHMP4B (Carlton et al. PNAS 2008) — ALIX localization to the midbody is independent of LBPA (Bissig Dev. Cell 2013). Similarly, ALIX Δ PRR may support exosome biogenesis because it contains intact binding sites in the BRO1 domain for both the membrane (LBPA) and ESCRT-III (CHMP4B). This has been clarified in the revised version.

Some slightly more specific points:

There are instances of overinterpretation in the text. E.g., L207 states that 'the results show that CHMP4B recruitment onto late endosomal membranes strictly depends on LBPA and ALIX' and L226 states 'LBPA and ALIX are necessary and sufficient for recruitment of ESCRT-III onto late endosomal membranes'. The authors haven't shown this, they have only shown that the endosomal localisation of GFP-CHMP4B that is driven by overexpression of ALIX-dPRR-mCherry depends upon [an interaction surface in ALIX that can bind] LBPA. Their subsequent argument that exosomes from ALIX depleted cells are both produced and contain less tetraspanins, indicates that ALIX/LBPA is dispensable for ILV generation, rather than being necessary and sufficient for ESCRT-III activity at this organelle.

We are sorry if this reviewer felt that our conclusions were overstated. We have toned down the text in both statements (ex-lines 207 and 226). However, we have addressed this concern, as mentioned above. In the original version, we could show that purified recombinant CHMP4B was recruited onto LBPA-containing bilayers by purified recombinant ALIX. We now show that, much like in vivo (old Fig 3A-E, new Fig 4A-E), CHMP4B recruitment in vitro is abolished when using the QQ mutant that does not bind LBPA, or the I212D mutant (new Fig 4I-J) that does not recruit CHMP4B. In addition, we also show that tetraspanins did not accumulate in exosomes after overexpression of the QQ mutant (new Fig S3B, D), much like after ALIX KD (old Fig 7C-D, new Fig 8C-D), consistent with observations that this mutant is unable to recruit CHMP4 in vivo and in vitro.

For the EGFR degradation experiments, as this is a transient overexpression experiment, untransfected cells will be in the majority and will likely swamp any effect on EGFR degradation observed in the transfected cells. Why not stably express the mutant and remove endogenous ALIX? Related to this, although the western blot in 5F makes it appear you are at near endogenous levels, the transfection efficiency will mean that you are expressing much more of the deletion mutant in the successfully expressing cells.

We are sorry if this was not clear, but both statements are incorrect. In our experiments, the majority of the cells were transfected (\approx 70%). Hence, untransfected cells did not swamp the effects observed in transfected cells, and ALIX Δ PRR is indeed expressed at near endogenous levels. This has been clarified in Legends and in the text. However, and to better address the concern of this reviewer, cells transfected with ALIX Δ PRR-mCherry were treated with EGF, and after different incubation times, analyzed by automated fluorescence microscopy using antibody against EGFR. This analysis confirms that ALIX Δ PRR expression did not affect EGFR degradation (new Fig 7E-F).

I find Figure 7 hard to interpret - the images appear very pixelated and the resolution seems

insufficient to accurately assess colocalization. It isn't clear why the authors have chosen LAMP1 as a marker as this will decorate lysosomes as well as MVBs. CD9 and CD81 don't appear to show any vesicular staining, apart from the CD81 in the dPRR transfected cells, which is the opposite of what the quantification reports. The CD63 was dotted and appeared to colocalise with the LAMP1 in all cells, whether or not they were expressing mCherry or mCherry-ALIXdPRR, which again, isn't what the quantification reports. I think what you are trying to say in the text is that in cells overexpressing ALIX dPRR, the tetraspanins are hyper-secreted, possibly on exosomes, but I am afraid that don't think the data supports this conclusion. In 7C, in the absence of ALIX, I can't really see any reduced CD63 incorporation into EVs on the western blot. I think that this part of the manuscript is quite weak and needs a lot more characterisation of the extracellular vesicle fraction to ensure that they are exosomes, rescue experiments and a molecular explanation of how ALIX controls tetraspanin incorporation into ILVs, and then EVs.

We do not understand the comment of this reviewer concerning LAMP1, since this protein is present at the same density (copies/squared micron) in late endosomes (endo-lysosomes) and lysosomes (see Griffiths et al Cell 1980). In fact, all late endosomal proteins/lipids, including LBPA, CD63 and Rab7 are mostly found in LAMP1-positive late endosomes or endo-lysosomes.

This point concerning Fig 7A is well-taken, and we agree with this reviewer that the data were not very convincing. In fact, CD9 turned out to be quite difficult to monitor adequately by immuno-fluorescence with the antibodies we have available. We have therefore removed the analysis of CD9, but we carried out a far more complete analysis of CD81 and CD63 by immuno-fluorescence.

We have now quantified CD63 and CD81 in LAMP1-endosomes by automated microscopy after ALIX Δ PRR expression and ALIX depletion by RNAi. To better illustrate the endosomal distribution of CD63 and CD81, we also show now the staining intensity of each tetraspanin within LAMP1-positive endo-lysosomes only (the mask used in automated microscopy), after subtraction of the staining present elsewhere (new Fig 9A). These data show that ALIX Δ PRR expression decreases the amounts of tetraspanin in the LAMP1-positive endo-lysosomes – consistent with our findings that ALIX Δ PRR expression stimulates tetraspanin exosomal secretion. Conversely ALIX knockdown does not affect the tetraspanin endosomal distribution, in agreement with our observations that the treatment inhibits exosomal secretion (new Fig S4 and new Fig S5). Altogether, our data support our conclusion that ALIX Δ PRR expression increases tetraspanin secretion, which is accompanied by a concomitant reduction in endosomes. To the best of our knowledge this is the first time that the distribution of tetraspanin is compared under conditions that do or do not promote exosome secretion.

To better address the concerns of this reviewer concerning the characterization of exosome formation, we have performed new experiments along three lines (as detailed also above):

1) Specificity. We have carried out two series of experiments to better illustrate the specificity of ALIX in this pathway. First, in the original version, we could show that purified recombinant CHMP4B was recruited onto LBPA-containing bilayers by purified recombinant ALIX. We now show that CHMP4B recruitment was abolished when using the QQ mutant (new Fig 4I-J), which fails to bind endosomes and to recruit CHMP4B in vivo (old Fig 3A-E, new Fig 4A-E). In addition, we also show that CD9, CD63 and to some extent CD81 did not accumulate in exosomes after overexpression of the QQ mutant (new Fig S3B, D), much like after ALIX KD (old Fig 7C-D, new Fig 8C-D), consistent with observations that this mutant is unable to recruit CHMP4 in vivo and in vitro. By contrast, the QQ mutant did not affect the sorting of EGFR into exosomes (new Fig S3B, D).

2) Loss of function. We show that ALIX depletion by RNAi decreases the secretion of exosomes containing tetraspanins and syntenin, which had been previously shown to mediate ALIX-dependent exosome production (Baietti et al. Nat Cell Biol. 2012). We also investigated whether the ESCRT subunit CHMP6, a well-established ESCRT-III nucleation factor, was involved in exosome production. Although CHMP6 is not recruited to endosomes upon ALIX Δ PRR expression (old Fig 2F, new Fig 3F) and is not required for ALIX-dependent CHMP4B membrane association (old Fig 4, new Fig 5), we now find that

CHMP6 depletion decreases the secretion of exosomes containing tetraspanins and syntenin (new Fig 8E-F), much like after ALIX depletion. These experiments support the notion that both CHMP6- and ALIX-dependent ESCRT-III nucleation mechanisms are involved in exosome biogenesis.

3) Ubiquitination. We have further investigated the role of protein ubiquitination in this process. Our data indicate that ubiquitination may be involved in protein sorting into exosomes, since ALIX Δ PRR causes the accumulation of conjugated ubiquitin in endosomes. We have now generated a CD9 mutant that cannot be ubiquitinated after replacement of all three cytoplasmic Lys residues with Arg (CD9/3R). We show that CD9/3R levels in exosomes are significantly reduced when compared to WT (new Fig 8G-H), supporting the notion that ubiquitination plays a role in sorting to exosomes. We also find that this defective incorporation could be partially rescued by ALIX Δ PRR overexpression, presumably because incorporation of the CD9/3R mutant into exosome was facilitated by interactions with other tetraspanins, upon stimulation of exosome production.

Discussion of the dispensability of ALIX for EGFR degradation is missing a key reference from the Woodman lab (Doyette et al., 2008) and the cited Schmidt paper claims ALIX has an active role in suppressing EGFR degradation, which is contrary to their argument.

The Doyotte reference from the Woodman group was added, and we corrected the text concerning the Schmidt reference.

Figure 1G, very few of the LBPA+ structures are ALIX or CHMP4B positive.

Figure 1G (new Fig 2A) was modified and a better image was selected to show more precisely the colocalization of ALIX and CHMP4B with late endosomes.

Minor:

Some of methodology could be made clearer - e.g., ALIX was described to be inserted in the XhoI site of pCMV-tag3, and then mCherry was then inserted downstream using the XhoI and ApaI sites. I'm not sure that this could happen as the XhoI digestion would liberate the ALIX. The methods section doesn't describe the untagged constructs - presumably these were used these in the paper too?

The methods section was revised and the issues raised by this reviewer were clarified.

The resolution of the blots prevents me from being certain, but the Rab5 blot in the LM fraction of S5A and 5D appear duplicated, but slightly differently cropped. I will grant that they are from the same type of experiment, but the accompanying ALIXdPRR blots are different and the legends claim that Rab5 was used in normalisation. I may be wrong, but if these were unintentionally duplicated, it may be best to exchange them.

We thank the reviewer for making us aware of this problem. In the blots shown in old Fig 5D and S5A (new Fig 6E and new Fig 7A), the same membranes were used for the incubation with antibodies against EGFR and ubiquitin. However, a mistake had been made when selecting the ALIXdPRR panels. The correct panels are now used in the new version, and Fig 7A Legends clearly state that the RAB5 and ALIXPRR blots are the same as in Fig 6E.

Related to this, the degree of pixilation in all of the supplied images is poor, to the extent that it is hard to meaningfully assess the data in the IF/westerns. This may just be the result of downsampling during the submission process, but I would recommend improving it.

The reviewer is correct to conclude that the low quality of some figures was due to the decreased resolution of the submitted files. As requested, we did our best to improve the micrographs and to increase the resolution.

ALIX depletion in 7C seems to stabilise CD81 - is this consistent?

This is an interesting point, but we have no evidence that CD81 is stabilized after ALIX depletion.

November 27, 2019

RE: JCB Manuscript #201904113R

Prof. Jean Gruenberg
University of Geneva
Department of Biochemistry
30 quai Ernest Ansermet
Geneva 4 1211
Switzerland

Dear Jean and Aurélien,

Thank you for submitting your revised manuscript entitled "ALIX- and ESCRT-III-dependent sorting of tetraspanins to exosomes". The manuscript was returned to the expert reviewers who replied generally favorably but two of which also remained critical with respect to some mechanistic details in the study. We appreciate that there is a substantial amount of work that went in this study already and we feel that the mechanistic message of the discovery of a new ILV pathway for the delivery of tetraspanins to exosomes is interesting enough to justify publication in JCB. Therefore, we will overrule the further requests for experiments by rev#1 and 3, and invite you to submit a revision where you address the minor issues and the criticism raised by these reviewers via text changes/clarifications.

One more technical issue I have pertains to the method used to isolate exosomes. Hadi Valadi asserts the importance of adding a filtration step after the differential centrifugation followed by another sedimentation step (see e.g. Maugeri et al. Nat Comm. 2019). If this is a gold standard in the exosome field I would encourage you to adopt it in your procedures, something that you can easily do I believe, in order to validate the key conclusions (mind that I do not ask you to repeat all experiments).

****Please be sure to include a point-by-point rebuttal document along with your revised manuscript.****

Once these issues are addressed, we would be happy to publish your paper in JCB pending final revisions necessary to meet our formatting guidelines (see details below).

A. MANUSCRIPT ORGANIZATION AND FORMATTING:

Full guidelines are available on our Instructions for Authors page, <http://jcb.rupress.org/submission-guidelines#revised>. ****Submission of a paper that does not conform to JCB guidelines will delay the acceptance of your manuscript.****

1) Text limits: Character count for Articles and Tools is < 40,000, not including spaces. Count includes title page, abstract, introduction, results, discussion, and acknowledgments. Count does not include materials and methods, figure legends, references, tables, or supplemental legends. You

are slightly over this limit at the moment. We should be able to give you the extra room in this case but please do your best to be as concise as possible when adding new text to the paper.

2) Figure formatting: Scale bars must be present on all microscopy images, including inset magnifications. Molecular weight or nucleic acid size markers must be included on all gel electrophoresis.

3) Statistical analysis: Error bars on graphic representations of numerical data must be clearly described in the figure legend. The number of independent data points (n) represented in a graph must be indicated in the legend. Statistical methods should be explained in full in the materials and methods. For figures presenting pooled data the statistical measure should be defined in the figure legends. Please also be sure to indicate the statistical tests used in each of your experiments (both in the figure legend itself and in a separate methods section) as well as the parameters of the test (for example, if you ran a t-test, please indicate if it was one- or two-sided, etc.). Also, since you used parametric tests in your study (e.g. t-tests, ANOVA, etc.), you should have first determined whether the data was normally distributed before selecting that test. In the stats section of the methods, please indicate how you tested for normality. If you did not test for normality, you must state something to the effect that "Data distribution was assumed to be normal but this was not formally tested."

4) Materials and methods: Should be comprehensive and not simply reference a previous publication for details on how an experiment was performed. Please provide full descriptions (at least in brief) in the text for readers who may not have access to referenced manuscripts. The text should not refer to methods "...as previously described."

5) Please be sure to provide the sequences for all of your primers/oligos and RNAi constructs in the materials and methods. You must also indicate in the methods the source, species, and catalog numbers (where appropriate) for all of your antibodies.

6) Microscope image acquisition: The following information must be provided about the acquisition and processing of images:

- a. Make and model of microscope
- b. Type, magnification, and numerical aperture of the objective lenses
- c. Temperature
- d. imaging medium
- e. Fluorochromes
- f. Camera make and model
- g. Acquisition software
- h. Any software used for image processing subsequent to data acquisition. Please include details and types of operations involved (e.g., type of deconvolution, 3D reconstitutions, surface or volume rendering, gamma adjustments, etc.).

7) References: There is no limit to the number of references cited in a manuscript. References should be cited parenthetically in the text by author and year of publication. Abbreviate the names of journals according to PubMed.

8) Supplemental materials: There are strict limits on the allowable amount of supplemental data. Articles/Tools may have up to 5 supplemental figures. At the moment, you are below this limit but please bear it in mind when revising. Please also note that tables, like figures, should be provided as individual, editable files. A summary

of all supplemental material should appear at the end of the Materials and methods section.

9) eTOC summary: A ~40-50 word summary that describes the context and significance of the findings for a general readership should be included on the title page. The statement should be written in the present tense and refer to the work in the third person.

10) Conflict of interest statement: JCB requires inclusion of a statement in the acknowledgements regarding competing financial interests. If no competing financial interests exist, please include the following statement: "The authors declare no competing financial interests." If competing interests are declared, please follow your statement of these competing interests with the following statement: "The authors declare no further competing financial interests."

11) ORCID IDs: ORCID IDs are unique identifiers allowing researchers to create a record of their various scholarly contributions in a single place. At resubmission of your final files, please consider providing an ORCID ID for as many contributing authors as possible.

B. FINAL FILES:

-- High-resolution figure and video files: See our detailed guidelines for preparing your production-ready images, <http://jcb.rupress.org/fig-vid-guidelines>.

Thank you for this interesting contribution, we look forward to publishing your paper in Journal of Cell Biology.

Sincerely,

Marino Zerial, PhD
Monitoring Editor
Journal of Cell Biology

Tim Spencer, PhD
Executive Editor
Journal of Cell Biology

Reviewer #1 (Comments to the Authors (Required)):

The revised manuscript by ... et al is substantially improved from the original version. Most of the concerns have been addressed satisfactorily.

The new added data and the associated text have raised a few additional minor concerns that should be addressed before publication by changes to the text or replacement of image panels with others that should be available. They are as follows:

1. In the new added data in Suppl. Fig. 3B-D showing the effect of overexpressing the QQ mutant on exosome content, while it is true that there is a decrease in the content of CD9, CD81 and syntenin as described in the text, there is actually an increase in the content of two integrins (alpha6 and beta3) in the image shown in panel B. The data in panel D would suggest that this increase was not consistently observed. If this is correct, then a more representative blot should be shown in its place.
2. It is disappointing that the authors were unable to directly link the recruitment of ESCRT-III factors in the first half of the paper to the effect of ALIXdeltaPRR on exosome content in Figure 8; the additional data on CHMP6 depletion are interesting, but not really on point given that CHMP6 recruitment was not affected by ALIXdeltaPRR expression in Figure 3. Given this, the authors should discuss these results more extensively on page 15; with the data provided, the conclusion that ALIX functions in exosome secretion by recruitment of CHMP4B is highly speculative, and should be indicated as such.
3. In the new data in Fig. 8G, H, the increased association of CD9-3R with exosomes upon overexpression of deltaPRR is attributed to the association of CD9-3R with endogenous ubiquitylated CD9, but this is pure speculation with no supporting data. It is fine to speculate, but please save it for the discussion (perhaps together with the new added text on page 16) and make it clear that this is speculation and not a solid conclusion. As a potential alternative explanation for the result, could it also be that the "exosome" preparation includes other extracellular vesicles, and that ALIXdeltaPRR might also support release of extracellular vesicles from tetraspanin-enriched domains of the plasma membrane?
4. Some of the labels in Figure 9 are very confusing. In panel A, the fourth column is labeled "Protein X in LAMP1"; this is not clear. What is meant is "Protein X overlapping LAMP1". In panels B and C, the label for the Y axis is very confusing. In panel B what appears to be meant (according to the Figure Legend) is the signal intensity overlapping LAMP1 per cell, like in panel A - is that correct (it looks from panel A like the numbers should be more dramatic)? Or is this really the total intensity of

the marker in the cells? The former would be much more informative for this paper. Also, why in this graph is there a signal for GM130, calnexin, flotillin 1, syntenin, or EGFR if these proteins did not overlap LAMP1? And what is indicated by UT? In panel C, it looks from the Figure Legend as if this should be the total integrated intensity per cell, and not the signal overlapping LAMP1 like expected for panel B; is this correct? What does the gray bar represent in this graph?

Reviewer #2 (Comments to the Authors (Required)):

The authors have done a great job during the revision and addressed the major points. The data of the new experiments significantly strengthen their original conclusion and support the model that they put forward: (1) there is more than one way to nucleate ESCRT-III on endosomes (2) and ALIX dependent ESCRT-III complexes sort tetraspanins into ILVs destined to become exosomes. This is an important model and I recommend the publication of the paper in JCB.

Reviewer #3 (Comments to the Authors (Required)):

Larios et al., have prepared a revised version of their manuscript documenting a role for ALIX and ESCRT-III in the sorting of tetraspanins to the internal vesicles of MVBs. The authors have made extensive revisions to the text to incorporate suggested references and discussion points. Many of the new data are clear and convincing (eg the QQ mutant in the in-vitro assays), but I'm afraid that other elements leave me a bit confused. I couldn't see much new loss of function data to explore how essential this pathway was for endosomal sorting of tetraspanins/exosome biogenesis - the loss of exosomal incorporation of tetraspanins in the absence of ALIX is minimal.

The authors discuss the involvement of ALIX in apoptosis, which prevented them over expressing WT ALIX as a control in many of their assays. However, the authors are using HeLa cells, which have an impaired p53 response and in which ALIX expression has been well tolerated in a variety of previous reports (e.g., Sun, Dev Cell, 2016; Lee et al., Science 2008; Carlton et al., PNAS 2008) - can the authors show that ALIX overexpression actually causes apoptosis in this context? That the paper (barring Fig 1B and S1) is based upon the results of overexpressing a deletion mutant vs empty vector is a concern for me.

Regarding new molecular understanding, while data relating to the QQ mutant are a nice addition, the data relating to ubiquitination are a little confusing. The authors use a ubiquitination-defective version of CD9 and show that exosomal CD9 is decreased when this mutant is expressed, suggesting ubiquitination is required for some element of sorting to ILVs, but then show that overexpression of the mutant ALIX causes exosomal incorporation of this protein at levels above controls; so ubiquitination now appears dispensable for the exosomal incorporation. I appreciate that the authors have caveated their findings with a 'may', but I'm afraid that I am still no wiser to the role of ubiquitin (or any other molecular details) in this process.

The in-vitro data suggest that the link between LBPA and the CHMP4 subunits of ESCRT-III can be performed by the Bro1 domain alone. Is this similar in cells - can you recruit ESCRT-III to endosomes by just overexpressing the Bro1 domain? This molecular ordering is nice, but I am afraid leaves me confused by Fig 4A - that mCherry-ALIXdPRR QQ cannot be recruited to endosomes under conditions of overexpression fits with the hypothesis that LBPA is doing the recruiting via QQ. However, mCherry-ALIXdPRR I212D also fails to be recruited to endosomes. As this site is the binding site for CHMP4, this suggests that ALIX is being recruited to CHMP4 already present on the

endosome. I can't see how CHMP4 can both be needed for recruiting ALIXdPRR through this defined site and also be recruited downstream of ALIXdPRR. Can you clarify?

Also, CHMP4/ESCRT-III assemblies on endosomes are highly dynamic (e.g., Migliano et al., eLIFE 2017; Quinney et al., PNAS 2019) and whilst the message of the paper is that ALIX recruits ESCRTs to endosomes, I wonder if an alternate explanation for persistent endosomal CHMP4 when you overexpress ALIX dPRR is that ALIX dPRR prevents the dynamic turnover of CHMP4? I appreciate that you've done a FRAP curve in 2D/E, but this recovery rate seems really slow (e.g., T half for full recovery of CHMP4 at midbody <60s (Mierzwa, NCB); you get to 35% recovery after 5 mins) and again absent of any controls (eg VPS4 mutant, ESCRT-II nucleator overexpression). Also, this was done in the presence of Noc - the curves in the absence of Noc would be useful to retain, as per the original submission.

More loss of function data is included in Fig 8, which show that depletion of the canonical ESCRT-III coordinating machinery on endosomes (CHMP6) has a stronger effect than this ALIX-dependent pathway on exosomal tetraspanin incorporation - while this is interesting, it doesn't address my original concerns on the molecular mechanism of how endogenous ALIX controls tetraspanin sorting to ILV. This and the following figure look to examine the relative distribution of tetraspanins (and others) between the exosomal pool and a LAMP-1-positive endosomal pool to gain appreciation of the relative distribution of these cargos. The message being that in the dPRR overexpressing cells, there is less CD63 in endosomes and more in the exosomal fraction. While less tetraspanin appears in the LAMP-1-positive mask of the cells overexpressing the ALIXdPRR mutant, this doesn't really parallel the biochemical data obtained in Fig 8; CD63 levels in exosomes are barely affected by ALIXdPRR overexpression (Fig 8, v. slight increase, maybe) whereas there's a 50% reduction in the LAMP+ endosomal signal (9B). Would biochemical fraction not be a more robust way of examining intracellular distribution of these exosomal cargos? Also, in Fig 9, ALIXdPRR has become largely cytosolic (compare Fig 9 to, say Fig 1C) - why is this and which is the correct distribution of this protein? Is CHMP4 still recruited to endosomes if mCherry-ALIX dPRR is cytosolic?

Reviewer #1:

The revised manuscript by ... et al is substantially improved from the original version. Most of the concerns have been addressed satisfactorily.

The new added data and the associated text have raised a few additional minor concerns that should be addressed before publication by changes to the text or replacement of image panels with others that should be available. They are as follows:

1. In the new added data in Suppl. Fig. 3B-D showing the effect of overexpressing the QQ mutant on exosome content, while it is true that there is a decrease in the content of CD9, CD81 and syntenin as described in the text, there is actually an increase in the content of two integrins (alpha6 and beta3) in the image shown in panel B. The data in panel D would suggest that this increase was not consistently observed. If this is correct, then a more representative blot should be shown in its place.

As requested by this reviewer, panel D in Fig S3 was replaced by a more representative blot.

2. It is disappointing that the authors were unable to directly link the recruitment of ESCRT-III factors in the first half of the paper to the effect of ALIX Δ PRR on exosome content in Figure 8; the additional data on CHMP6 depletion are interesting, but not really on point given that CHMP6 recruitment was not affected by ALIX Δ PRR expression in Figure 3. Given this, the authors should discuss these results more extensively on page 15; with the data provided, the conclusion that ALIX functions in exosome secretion by recruitment of CHMP4B is highly speculative, and should be indicated as such.

We are sorry if the text was not clear. Our data with CHMP6 agree well with the notion that two parallel pathways exist for ESCRT-III endosomal recruitment in yeast: one dependent on ESCRT-I/II (canonical pathway) and a second one dependent on ESCRT-0/BRO1 (Tang et al., 2016). Our results show that ALIX Δ PRR-dependent CHMP4B recruitment to endosomes fit nicely with the recent findings that ALIX can nucleate ESCRT-III on endosomal membranes (Skowyra et al Science 2018; Radulovic et al EMBOJ 2018). In addition, our observations that CHMP6/VPS20 is not recruited to endosomes upon ALIX Δ PRR expression, and not required for ALIX-dependent CHMP4B membrane association, but is involved in exosome biogenesis, also fit very nicely the role of CHMP6/VPS20 as CHMP4/SNF7 nucleation factor (Saksena et al., 2009; Tang et al., 2015; Teis et al., 2008). Altogether, these observations clearly support the emerging view that ESCRT-III recruitment and polymerization on endosomal membranes can be mediated alternatively via parallel pathways, dependent on ALIX and CHMP6, respectively. This issue was clarified in the Discussion.

We have carefully avoided the overinterpretation of our observations. In the Discussion, we state that precise functions of the ALIX-dependent pathway into late endosomes remain to be elucidated, but that some speculations based on our observations are possible. This was further clarified in the revised version.

3. In the new data in Fig. 8G, H, the increased association of CD9-3R with exosomes upon overexpression of deltaPRR is attributed to the association of CD9-3R with endogenous ubiquitylated CD9, but this is pure speculation with no supporting data. It is fine to speculate, but please save it for the discussion (perhaps together with the new added text on page 16) and make it clear that this is speculation and not a solid conclusion. As a potential alternative explanation for the result, could it also be that the "exosome" preparation includes other extracellular vesicles, and that ALIX Δ PRR might also support release of extracellular vesicles from tetraspanin-enriched domains of the plasma membrane?

As requested we have changed the text and now mention the alternative explanation mentioned by the reviewer. We decided to clarify this issue already in the Results section, to avoid any possible further confusion. Indeed, one cannot exclude the possibility that ALIX Δ PRR overexpression triggered the release of CD9/3R-containing microvesicles from the plasma membrane. However, this is unlikely,

given the fact that ALIX Δ PRR is not detected on the plasma membrane and that ALIX Δ PRR membrane association requires an intact binding site for the late endosome lipid LBPA. Alternatively, the incorporation of the CD9/3R mutant into exosome is facilitated by interactions of CD9/3R with WT CD9 and other tetraspanins upon stimulation of exosome production (Fig 8A). This issue was clarified in the revised version.

4. Some of the labels in Figure 9 are very confusing. In panel A, the fourth column is labeled "Protein X in LAMP1"; this is not clear. What is meant is "Protein X overlapping LAMP1". In panels B and C, the label for the Y axis is very confusing. In panel B what appears to be meant (according to the Figure Legend) is the signal intensity overlapping LAMP1 per cell, like in panel A - is that correct (it looks from panel A like the numbers should be more dramatic)? Or is this really the total intensity of the marker in the cells? The former would be much more informative for this paper. Also, why in this graph is there a signal for GM130, calnexin, flotillin 1, syntenin, or EGFR if these proteins did not overlap LAMP1? And what is indicated by UT? In panel C, it looks from the Figure Legend as if this should be the total integrated intensity per cell, and not the signal overlapping LAMP1 like expected for panel B; is this correct? What does the gray bar represent in this graph?

We are sorry if the captions in Fig 9 were confusing. However, it is not easy to find adequate and simple terms. This reviewer suggests: "Protein X overlapping LAMP1", but this is also confusing since it conveys the incorrect notion that the given marker (e.g. CD81) overlaps with LAMP1. We have replaced "Protein X in LAMP1" with "Protein X in LAMP1 mask" as the simplest and most concise manner to describe the image (the distribution of each marker within LAMP1-positive endo-lysosomes).

The labeling in panel (B) is clear. As stated in the Legends, the Y-axis corresponds to the integrated signal intensity of the corresponding marker in LAMP1 endosomes per cell, hence the integrated intensity per cell of the signal shown in the corresponding right panels.

We fully agree with this reviewer that the signals for proteins that do not overlap with LAMP1 (GM130, calnexin, flotillin 1, syntenin, EGFR) are meaningless. However, we did these experiments because they were specifically requested by this reviewer, and we did not feel that it was worth arguing with the reviewer. This reviewer wrote (capitals as in original review): "*First, quantification of ALL markers - including EGFR, flotillin, and integrin components, as well as a solid negative control like calnexin - in both exosome fractions (relative to cell lysates by blotting) and LAMP1+ endosomes (by immunofluorescence microscopy) should be done in both ALIX Δ PRR overexpressers and ALIX KD cells*". At this point, we leave the final decision to the editor, but we are happy to keep the micrographs as they are now.

UT refers to untransfected control cells. The Legend was corrected.

As shown in Fig S5A and S5B and as stated in the Legend, the Y-axis in panel C corresponds to the integrated signal intensity of the corresponding marker in LAMP1 endosomes per cell, as in panel B.

Reviewer #2:

The authors have done a great job during the revision and addressed the major points. The data of the new experiments significantly strengthen their original conclusion and support the model that they put forward: (1) there is more than one way to nucleate ESCRT-III on endosomes (2) and ALIX dependent ESCRT-III complexes sort tetraspanins into ILVs destined to become exosomes.

This is an important model and I recommend the publication of the paper in JCB.

We are very grateful to this reviewer for his positive comments.

Reviewer #3:

Larios et al., have prepared a revised version of their manuscript documenting a role for ALIX and ESCRT-III in the sorting of tetraspanins to the internal vesicles of MVBs. The authors have made extensive revisions to the text to incorporate suggested references and discussion points. Many of the new data are clear and convincing (eg the QQ mutant in the in-vitro assays), but I'm afraid that other elements leave me a bit confused. I couldn't see much new loss of function data to explore how essential this pathway was for endosomal sorting of tetraspanins/exosome biogenesis - the loss of exosomal incorporation of tetraspanins in the absence of ALIX is minimal.

This reviewer states that the decrease in the incorporation of tetraspanins in exosomes without ALIX is minimal. There is no doubt that the effects of ALIX siRNAs are only partial. However, our data are not significantly different from similar data by others. For example, Baietti et al (Nature Cell Biology, 2012) observed approx. 50% decrease in syntenin incorporation after ALIX KO (Fig 2, Baietti et al), much like us (Fig 8C). The apparent limited effects of the siRNAs on CD63 most likely reflects the use of different antibodies in our and in the Baietti studies. Baietti's antibody stained a few CD63 chains only, while ours showed mostly the smear (due to differential glycosylation), causing an overestimation of the background. But, then the CD63 values in Baietti are comparable to CD81 (another tetraspanin) in our work, which shows a staining pattern on blots similar to CD63 in Baietti et al. In addition, our work suggests that the biogenesis of exosome containing tetraspanins can be driven by ALIX or CHMP6. Presumably, CHMP6 accounts for partial tetraspanin secretion when ALIX is depleted.

The authors discuss the involvement of ALIX in apoptosis, which prevented them over expressing WT ALIX as a control in many of their assays. However, the authors are using HeLa cells, which have an impaired p53 response and in which ALIX expression has been well tolerated in a variety of previous reports (e.g., Sun, Dev Cell, 2016; Lee et al., Science 2008; Carlton et al., PNAS 2008) - can the authors show that ALIX overexpression actually causes apoptosis in this context? That the paper (barring Fig 1B and S1) is based upon the results of overexpressing a deletion mutant vs empty vector is a concern for me.

We thank the reviewer for this comment and the clarification. We feel that it is not necessary for us to demonstrate whether ALIX triggers apoptosis, since we are using full length ALIX in a few experiments only and since the issue extends much beyond the present study. However, we share this reviewer's concern that protein overexpression can be problematic. In particular, and as this reviewer is probably aware, overexpression of ALIX or several other ESCRT subunits is not well-tolerated by many cell types. In this context, we believe that deletion of the PRR limits the danger that problems potentially associated with the overexpression of the full-length protein occur.

Regarding new molecular understanding, while data relating to the QQ mutant are a nice addition, the data relating to ubiquitination are a little confusing. The authors use a ubiquitination-defective version of CD9 and show that exosomal CD9 is decreased when this mutant is expressed, suggesting ubiquitination is required for some element of sorting to ILVs, but then show that overexpression of the mutant ALIX causes exosomal incorporation of this protein at levels above controls; so ubiquitination now appears dispensable for the exosomal incorporation. I appreciate that the authors have caveated their findings with a 'may', but I'm afraid that I am still no wiser to the role of ubiquitin (or any other molecular details) in this process.

We thank this reviewer for his insightful comment. However, we do not feel that these data question the role of ubiquitination in the process. We feel that our observations that CD9 incorporation into exosomes is significantly decreased after mutagenesis of all cytoplasmically-exposed Lys residues (CD9/3R mutant) strongly supports the notion that ubiquitination is involved in the sorting process.

Is it so surprising that the process can be rescued by overexpression of ALIXdeltaPRD? ALIXdeltaPRD presumably dimerizes (Pires et al Structure 2010; Bissig et al Dev Cell 2013) and hetero-dimerizes with endogenous ALIX. In addition, tetraspanins interact with each other forming homo- and hetero-multimers (Berditchevski and Odintsova, 2007; Perez-Hernandez et al., 2013; van Deventer et al., 2017). Presumably, CD9/3R incorporation into exosome is facilitated by interactions of CD9/3R with WT CD9 and other tetraspanins, upon stimulation of exosome production by ALIXdeltaPRD .

The in-vitro data suggest that the link between LBPA and the CHMP4 subunits of ESCRT-III can be performed by the Bro1 domain alone. Is this similar in cells - can you recruit ESCRT-III to endosomes by just overexpressing the Bro1 domain? This molecular ordering is nice, but I am afraid leaves me confused by Fig 4A - that mCherry-ALIXdPRR QQ cannot be recruited to endosomes under conditions of overexpression fits with the hypothesis that LBPA is doing the recruiting via QQ. However, mCherry-ALIXdPRR I212D also fails to be recruited to endosomes. As this site is the binding site for CHMP4, this suggests that ALIX is being recruited to CHMP4 already present on the endosome. I can't see how CHMP4 can both be needed for recruiting ALIXdPRR through this defined site and also be recruited downstream of ALIXdPRR. Can you clarify?

Our previous data in an ALIX knockdown background showed that ALIX functions on endosomes could not be restored by expression of the Bro1 domain alone or by ALIX dimerization mutants (Bissig et al., Dev Cell 2013). We speculated that ALIX dimerizes upon membrane interactions. If so, the Bro1 domain alone may not be able to nucleate CHMP4B polymerization on membranes.

Similarly, the ALIXΔPRR-I212D mutant is unable to recruit GFP-CHMP4B onto endosomal membranes, and is itself strictly cytosolic. These observations indicate not only that ALIX recruits ESCRT-III highly specifically, but also that interactions with ESCRT-III and the polymerization of ESCRT-III filaments are necessary to stabilize ALIX onto endosomal membranes. Without ESCRT-III, ALIX is released and no longer remains membrane-associated.

Also, CHMP4/ESCRT-III assemblies on endosomes are highly dynamic (e.g., Migliano et al., eLIFE 2017; Quinney et al., PNAS 2019) and whilst the message of the paper is that ALIX recruits ESCRTs to endosomes, I wonder if an alternate explanation for persistent endosomal CHMP4 when you overexpress ALIX dPRR is that ALIX dPRR prevents the dynamic turnover of CHMP4? I appreciate that you've done a FRAP curve in 2D/E, but this recovery rate seems really slow (e.g., T half for full recovery of CHMP4 at midbody <60s (Mierzwa, NCB); you get to 35% recovery after 5 mins) and again absent of any controls (eg VPS4 mutant, ESCRT-II nucleator overexpression). Also, this was done in the presence of Noc - the curves in the absence of Noc would be useful to retain, as per the original submission.

We thank the reviewer for this insightful comment. We cannot exclude the possibility that overexpression of ALIX dPRR interferes with ESCRT-III dynamic turnover. However, even if it were the case, it would not challenge our main conclusions that ALIX interactions with LBPA and CHMP4B are necessary for ESCRT-III nucleation and polymerization and in turn stimulate the secretion of exosomes containing tetraspanins. In addition, one should probably also bear in mind that these structures, endosomes and midbody, are different and thus that the kinetics of recovery are not easily compared. Finally, recovery by FRAP at the midbody is not easily discriminated from the recruitment of additional subunits.

More loss of function data is included in Fig 8, which show that depletion of the canonical ESCRT-III coordinating machinery on endosomes (CHMP6) has a stronger effect than this ALIX-dependent pathway on exosomal tetraspanin incorporation - while this is interesting, it doesn't address my original concerns on the molecular mechanism of how endogenous ALIX controls tetraspanin sorting to ILV. This and the following figure look to examine the relative distribution of tetraspanins (and others) between the exosomal pool and a LAMP-1-positive endosomal pool to gain appreciation of the relative distribution of these cargos. The message being that in the dPRR overexpressing cells, there is less CD63 in endosomes and more in the exosomal fraction. While less tetraspanin appears in the LAMP-1-positive mask of the cells overexpressing the ALIXdPRR mutant, this doesn't really parallel the biochemical data obtained in Fig 8; CD63 levels

in exosomes are barely affected by ALIXdPRR overexpression (Fig 8, v. slight increase, maybe) whereas there's a 50% reduction in the LAMP+ endosomal signal (9B). Would biochemical fraction not be a more robust way of examining intracellular distribution of these exosomal cargos? Also, in Fig 9, ALIXdPRR has become largely cytosolic (compare Fig 9 to, say Fig 1C) - why is this and which is the correct distribution of this protein? Is CHMP4 still recruited to endosomes if mCherry-ALIX dPRR is cytosolic?

We agree with this reviewer that our work has not uncovered the mechanisms of ALIX-dependent tetraspanin sorting into ILV. However, we feel that this issue extends much beyond the scope of the present study and will have to be addressed in the future.

The point made by this reviewer with CD63 detection is correct, and is essentially identical to the first point made above by this reviewer. Our antibody against CD63 shows mostly a smear in gels (due to differential glycosylation), causing an overestimation of the background, and therefore fractionation would unfortunately not provide an alternative issue, as detection also relies on blots.

The distribution of ALIXdPRR in Fig 1 and 9 is not different, but the microscopes are. Fig 1C is a high resolution, confocal image and the cytosolic staining appears significantly weaker. By contrast, pictures in Fig 9 were captured by automated microscopy and the low-resolution micrographs shows the presence of ALIXdPRR in the cytosol and in punctae, corresponding to LAMP1-positive endosomes.